# Disentangling the architectural and non-architectural functions of CTCF and cohesin in gene regulation

Takeo Narita [1], Sinan Kilic [1], Yoshiki Higashijima [1,2], Natalie M. Scherer [1], Georgios Pappas[1], Elina Maskey[1] & Chunaram Choudhary [1] ✉

Cohesin- and CTCF-mediated chromatin loops facilitate enhancer–promoter and promoter–promoter interactions, but their impact on global gene regulation remains debated. Here we show that acute removal of cohesin or CTCF in mouse cells dysregulates hundreds of genes. Cohesin depletion primarily downregulates CBP/p300-dependent putative enhancer targets, whereas CTCF loss both up- and downregulates enhancer targets. Beyond loop anchoring, CTCF directly modulates transcription, acting as an activator or repressor depending on its binding position and orientation at promoters. Mechanistically, when activating, CTCF increases DNA accessibility and promotes RNA polymerase II recruitment; when repressing, it prevents RNA polymerase II binding without altering chromatin accessibility. Promoter-bound CTCF activates housekeeping genes essential for cell proliferation. CTCF's transcriptional activation function—but not its loop anchoring role—is shared with its vertebrate-specific paralog, CTCFL. These findings reconcile architectural and non-architectural roles of cohesin and CTCF, offering a unified model for their functions in enhancer-dependent and enhancer-independent transcription control.

Cohesin and CTCF fold vertebrate genomes into loops and topologically associating domains[1–5]. Cohesin-driven looping promotes enhancer–promoter (E–P) and promoter–promoter interactions, while CTCF anchors these loops to ensure proper enhancer targeting and prevent misactivation[6–8]. Disrupting or inverting CTCF sites rewires E–P loops and alters gene expression[5,9–12]. Promoter-proximal CTCF binding has been proposed to facilitate long-range enhancer contacts[13–15] and select target promoters within topologically associating domains[16]. Genome-wide chromatin conformation analyses have mapped thousands of E–P loops, implying a broad role of three-dimensional genome folding on transcription regulation[3,14,17–20].

Although CTCF was originally described as a transcriptional repressor[21] and activator[22], its gene regulatory roles are now largely interpreted in the context of its architectural function in genome organization[23,24]. Acute removal of CTCF or cohesin globally disrupts loop domains, but the resulting transcriptional changes are unexpectedly limited[25,26]. Several key questions remain unresolved: (1) Why do cohesin and CTCF depletion regulate few genes[25,26], and why do the regulated genes show limited overlap[20,27]? (2) If and to what extent does CTCF impact transcription by non-architectural mechanisms? (3) CTCF is conserved across bilaterians[28]; why is it essential for the proliferation of mammalian but not *Drosophila* cells[29–33]? (4) What functional similarities exist between CTCF and its vertebrate-specific paralog CTCFL, which cannot anchor cohesin loops[34]?

We previously showed that CBP/p300-mediated H2B N-terminal acetylation (H2BNTac) marks active enhancers[35,36], that CBP/p300 is

[1]Proteomic Program, The Novo Nordisk Foundation Center for Protein Research, Department of Cellular and Molecular Medicine, Faculty of Health and Medical Sciences, University of Copenhagen, Copenhagen, Denmark. [2]Institute for Promotion of Tenure Track, University of Miyazaki, Miyazaki, Japan. ✉e-mail: chuna.choudhary@cpr.ku.dk

essential for recruiting RNA polymerase II (Pol II) to enhancers[37] and that it plays a central role in regulating cell-type-specific genes[38], and that most CRISPR-interference-identified enhancers act through CBP/p300 (ref. 39).

Here, we show that cohesin and CTCF regulate hundreds of genes, with cohesin primarily activating CBP/p300-dependent enhancer targets. By contrast, CTCF functions through both enhancer-dependent and enhancer-independent mechanisms, with or without cohesin. These distinct roles explain the limited overlap in transcriptional changes upon cohesin or CTCF loss, clarify CTCF's essentiality for mammalian cell proliferation and reveal how it regulates genes independently of chromatin looping.

## Results

### Cohesin regulates many genes, albeit subtly

Multiple enhancer types are reported in metazoans[40–44]. Earlier studies assessed transcription changes after acute RAD21 or CTCF depletion[20,25,26], but did not resolve what enhancer type(s) drives the observed changes.

A close association between CBP/p300 and enhancers[35–39] motivated us to examine the involvement of cohesin and CTCF in activating genes through CBP/p300-dependent enhancer type. We used mouse embryonic stem cells (mESCs) expressing degron-tagged *Rad21-mAid-Gfp* (RAD21[AID])[26] (Extended Data Fig. 1a). Nascent transcription was profiled by nascent RNA labeling by 5-ethynyl uridine (5-EU) and next-generation sequencing (EU-seq) at 4 h post depletion ($n = 2$). CBP/p300-regulated genes were previously identified[37] after treating (0.5–2 h) cells with the CBP/p300 inhibitor A-485 (ref. 45).

A-485 significantly ($P_{adj} < 0.05$, ≥2-fold change) downregulated >1,000 genes, whereas RAD21[AID] depletion altered only one (Extended Data Fig. 1b). Similar results were obtained previously for RAD21[AID] depletion[20]. To capture weakly regulated genes, we used a fold-change (FC)-based criterion, categorizing genes as highly down- or upregulated (HD or HU; ≥2-FC), intermediate down- or upregulated (ID or IU; 1.5–2-FC), slightly down- or upregulated (SD or SU; 1.3–1.5-FC) and not changed (NC; <1.2-FC). After filtering low-expressed genes (transcripts per million (TPM) < 15), >300 showed 1.3–3-fold downregulation (Extended Data Fig. 1c).

To test the robustness of the FC-based changes, we performed 12 additional EU-seq replicates. While more genes passed the statistical threshold ($P_{adj} < 0.05$), the total remained low (20 down, 1 up) (Extended Data Fig. 2a). Notably, despite the limited number of regulated genes and modest effect sizes, transcriptional changes showed strong correlation between initial and replicate measurements (Pearson's correlation coefficient (PCC) = 0.65; Extended Data Fig. 2b). Over 90% of strongly regulated genes (≥1.5-fold) from initial experiments were consistently downregulated in ≥6 new replicates, compared with only 1.3% among initially unchanged genes (Extended Data Fig. 2c,d).

This, together with the analyses presented below, supports the overall reproducibility of the measured transcription changes (Supplementary Note 1). For further analyses, RAD21-regulated genes were defined as those with ≥1.3-FCs in initial experiments and reproducible regulation in at least half of the new replicates (Extended Data Fig. 2e). Downregulated genes (HD + ID + SD) were grouped as RAD21[down], upregulated genes (HU + IU + SD) as RAD21[up] and unchanged genes as RAD21[NC].

### RAD21-regulated genes are biased for CBP/p300 regulation

Next, we examined CBP/p300's involvement in activating cohesin-dependent genes. A-485-regulated genes were categorized as very highly downregulated (HD2, ≥4-fold), highly downregulated (HD1, ≥2-fold), intermediate downregulated (ID, 1.5–2-fold) or not downregulated (ND, <1.2-fold). The combined downregulated categories (HD2 + HD1 + ID) are labeled A-485[down], and unaffected genes are labeled A-485[ND].

RAD21[AID] depletion downregulated 2% ($n = 216$) and upregulated 0.5% ($n = 44$) of genes by ≥1.3-fold, whereas A-485 treatment downregulated 21% ($n = 2,032$) by ≥1.5-fold (Extended Data Fig. 2e). RAD21[down] genes are biased for A-485[down] genes, whereas RAD21[up] genes are not (Fig. 1a). Genes most strongly downregulated by RAD21[AID] depletion were also most downregulated by A-485 (Fig. 1b). Among RAD21[down] genes, the fraction of A-485[down] genes increased progressively from SD (83%) to ID (90%) to HD (100%) (Fig. 1c). Among A-485-regulated gene categories, RAD21[down] genes were 127-fold more frequent in HD2 versus ND (16.6% versus 0.13%; odds ratio (OR) = 199, $P < 2.2 \times 10^{-16}$, Fisher's exact test) (Fig. 1d).

Although cohesin also regulates Polycomb interactions[46], acute RAD21[AID] depletion-induced gene downregulation appears most consistent with impaired enhancer interaction rather than increased Polycomb repression (Extended Data Fig. 2f and Supplementary Note 2).

### Broader regulatory scope of cohesin in differentiated cells

To assess cohesin's scope in differentiated cells, RAD21[AID] mESCs were differentiated into neuronal progenitor cells (NPCs). Nascent transcriptomes were profiled after A-485 (1 h) and RAD21[AID] depletion (1–4 h). Number of downregulated genes increased with depletion duration, with 89–90% overlap across time points (Extended Data Fig. 3a–c). As in mESCs, RAD21[down] genes in NPCs were enriched for A-485[down] genes, while RAD21[up] showed little bias (Fig. 1e).

Among genes strongly downregulated (HD + ID) after 2–4 h of RAD21[AID] depletion, 89–94% are also downregulated by A-485 (2 h: OR = 93.8; 4 h: OR = 53.9; $P < 2.2 \times 10^{-16}$, Fisher's exact test) (Fig. 1f). Indeed, among RAD21[down] HD genes, 92–97% are downregulated >2-fold by A-485 (2 h: OR = 141.2; 4 h: OR = 367.1; $P < 2.2 \times 10^{-16}$) and 80–82% are downregulated >4-fold (2 h: OR = 180.9; 4 h: OR = 167.1; $P < 2.2 \times 10^{-16}$) (Fig. 1f). Overlap was weaker in the SD category, especially at 4 h, likely reflecting accumulation of secondary transcriptional changes and more false positives among weakly regulated genes.

Genes most strongly regulated by CBP/p300 were also most sensitive to RAD21[AID] depletion: among genes downregulated >4-fold by A-485, 42–51% of them were downregulated by RAD21[AID] depletion (Fig. 1g). In contrast, A-485[ND] genes remained largely unaffected by RAD21[AID] depletion.

Regulated NPC genes included known and putative enhancer targets. Some, such as *Myc*, *Kitl*, *Enc1*, *Tnfrsf21* and *Fbn2*, were downregulated by both RAD21[AID] depletion and A-485, whereas others (*Gli3*, *Zfp608*) were only affected by A-485 (Supplementary Fig. 1a). The *Gli3* proximal gene, *Inhba*, was RAD21-sensitive. *Zfp608*, where a newly described scaffolding element promotes looping[47], is activated by CBP/p300, but not by RAD21[AID] depletion (Supplementary Fig. 1b). *Fbn2*, silent in mESCs but looped by cohesin[5,48], became expressed in NPCs and was downregulated by both RAD21[AID] depletion and A-485.

Collectively, this demonstrates a broad role of cohesin in promoting CBP/p300-dependent enhancer target activation in differentiated cells.

### RAD21- and CTCF-regulated genes differ in CBP/p300 dependency

Next, we checked the dependence of CTCF-regulated genes on CBP/p300. CTCF-regulated genes were quantified in mESCs after acute depletion of *Ctcf-mAid-Gfp* (CTCF[AID])[26] (Extended Data Fig. 4a). CTCF[AID] depletion (6 h) significantly regulated 37 genes (32 down, 5 up) (Extended Data Fig. 4b), consistent with earlier reports[20]. For further analyses, CTCF-regulated genes were classified using the above-described FC and expression (TPM > 15) criteria (Extended Data Fig. 4c,d). Downregulated, upregulated and not changed genes are denoted as CTCF[down], CTCF[up] and CTCF[NC], respectively.

Supporting the accuracy of our measurements, nascent transcript changes at 6 h correlated with published messenger RNA changes at 24 h (ref. 49) (PCC = 0.53; Extended Data Fig. 4e). FC-based nascent

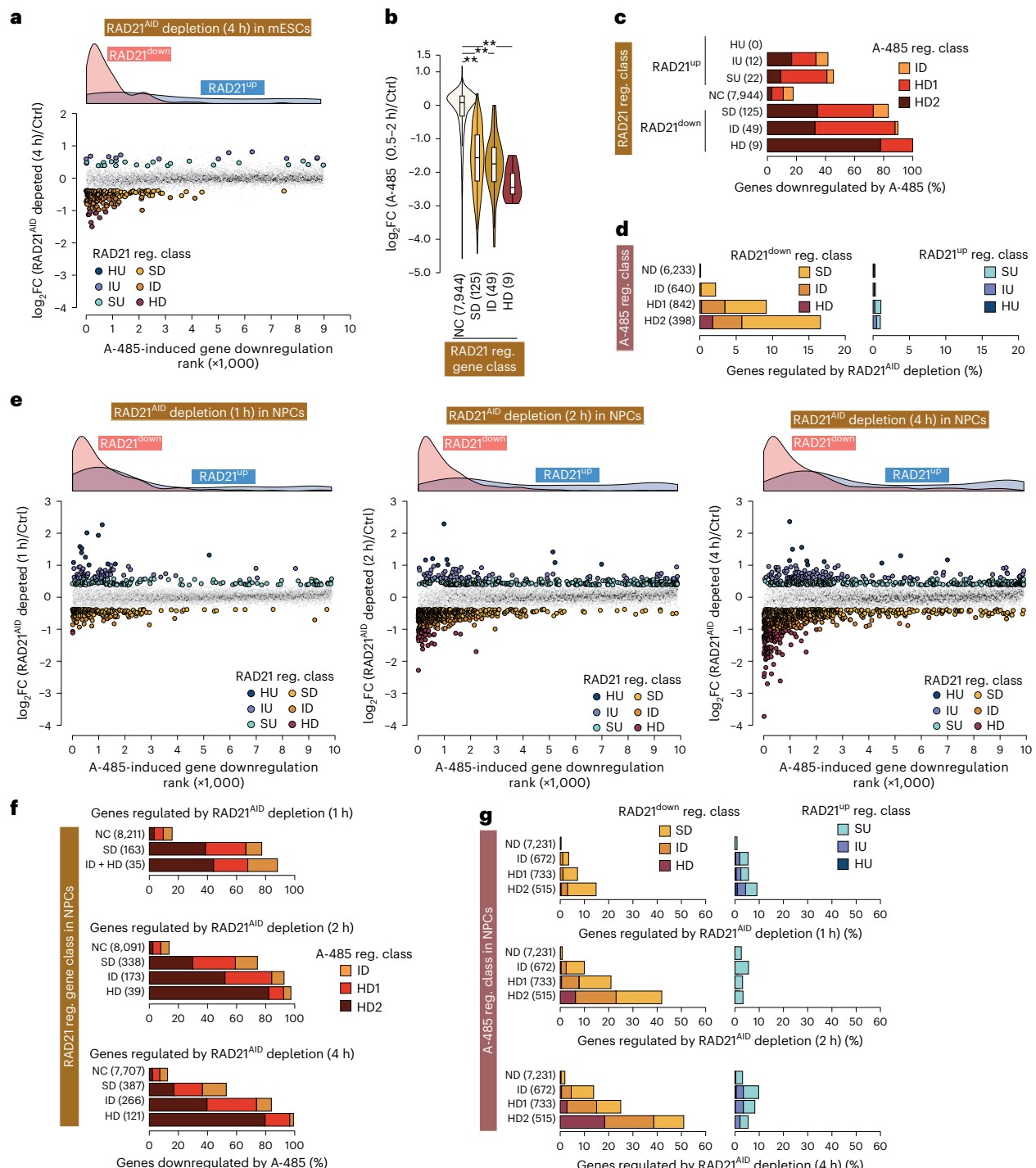

**Fig. 1 | RAD21^AID depletion downregulates hundreds of genes, with a strong bias toward CBP/p300-regulated targets. a**, Genes downregulated by RAD21^AID depletion in mESCs are biased for regulation by CBP/p300. Genes are ranked based on the degree of downregulation after A-485 treatment. Density plots (top) show the distribution of genes up- and downregulated upon RAD21^AID depletion. Classification of RAD21-regulated genes (RAD21 reg. class) is defined in Extended Data Fig. 2e. **b**, A-485-induced gene regulation across different groups of RAD21^AID-regulated genes in mESCs. Box plots show the median and upper and lower quartiles; whiskers represent 1.5 × interquartile range (IQR). Statistical significance was determined by two-sided Mann–Whitney U-test, followed by correction for multiple comparisons with the Benjamini–Hochberg method; **$P_{adj}$ < 0.001, *$P_{adj}$ < 0.05. $P_{adj}$: NC versus HD, 7.4 × 10⁻⁷; NC versus ID, 1.8 × 10⁻²⁵; NC versus SD, 4.7 × 10⁻⁵⁴. **c**, Proportion of A-485-downregulated genes within RAD21^AID depletion-regulated gene classes in mESCs. Within each category, the fraction of genes upregulated or downregulated after A-485 treatment

is depicted. **d**, Fraction of RAD21^AID depletion up- and downregulated genes within the indicated A-485-regulated gene groups in mESCs. Genes regulated after A-485 treatment are grouped into the indicated classes. Within each class, the fraction of genes regulated by RAD21^AID depletion is shown. **e**, Genes downregulated by RAD21^AID depletion in NPCs are biased for regulation by CBP/ p300. Genes are ranked by A-485-induced downregulation. Density plots (top) show the distribution of genes up- and downregulated after RAD21^AID depletion (left 1 h, middle 2 h, right 4 h). **f**, Fraction of A-485-downregulated genes within the indicated groups of genes regulated by RAD21 in NPCs. RAD21-regulated genes are grouped into the indicated class; within each class, the fraction of genes downregulated by A-485 (1 h) treatment is depicted. **g**, Within the specified A-485-regulated gene groups, the fraction of genes up- and downregulated by RAD21^AID depletion in NPCs. A-485-regulated genes are grouped as specified, and the proportion affected by RAD21^AID depletion is shown. Ctrl, control.

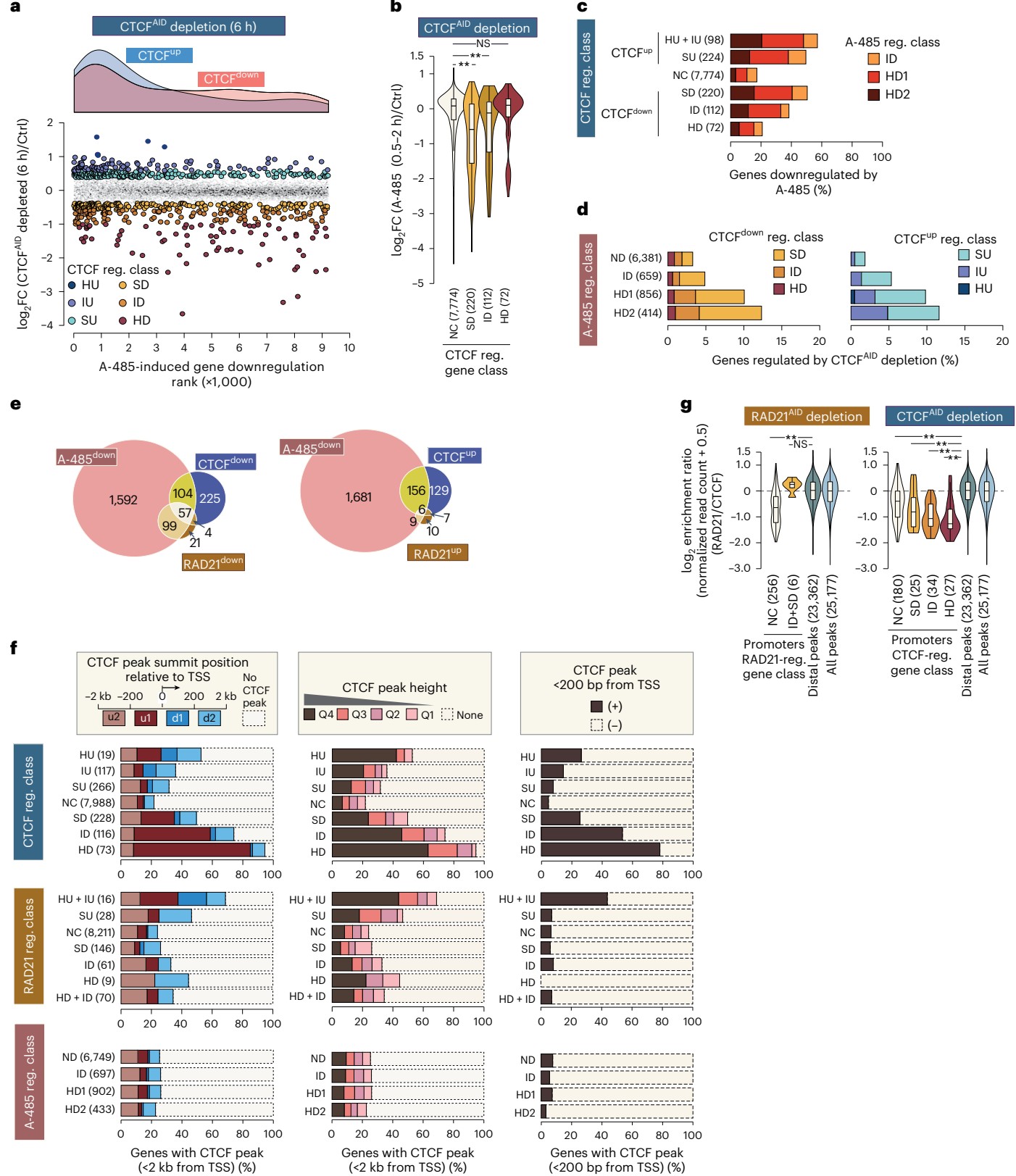

transcript up- and downregulation show significant consistency in mRNA FC expression ($P_{adj}$; HU versus NC $7.1 \times 10^{-11}$, IU versus NC $2.3 \times 10^{-40}$, SU versus NC $2.8 \times 10^{-52}$, SD versus NC $9.2 \times 10^{-62}$, ID versus NC $1.5 \times 10^{-46}$, HD versus NC $8.4 \times 10^{-44}$, two-sided Mann–Whitney $U$-test) (Extended Data Fig. 4f).

Within CTCF[down] genes, weakly regulated genes were more sensitive to A-485, opposite to the trend seen in RAD21[down] genes (Figs. 1a,b

and 2a,b). Among CTCF[down] categories, overlap with A-485[down] genes decreased as CTCF[AID] depletion-induced gene downregulation increased (SD: 50%, ID: 38%, HD: 21%) (Fig. 2c). In contrast, CTCF[up] genes showed the opposite pattern, with overlap increasing as upregulation became stronger.

Thus, only a subset of CTCF-regulated genes depends on CBP/p300, with the strongest CTCF targets being least CBP/

**Fig. 2 | Genes downregulated by CTCF^AID depletion are enriched for promoter-proximal CTCF binding, and most of them activated independently of cohesin and CBP/p300. a**, Genes up- and downregulated by CTCF^AID depletion in mESCs are biased for regulation by CBP/p300. Genes are rank-ordered by the extent of their downregulation after A-485 treatment, and the density plots (top) show the distribution of genes up- and downregulated after CTCF^AID depletion. CTCF-regulated gene class is defined based on gene expression FC thresholds defined in Extended Data Fig. 4c,d. **b**, A-485-induced regulation within the indicated CTCF-regulated gene classes. Box plots show the median and upper and lower quartiles; whiskers represent 1.5 × IQR. Statistical significance was determined by two-sided Mann–Whitney *U*-test, followed by correction for multiple comparisons with the Benjamini–Hochberg method; **$P_{adj}$ < 0.001, *$P_{adj}$ < 0.05. $P_{adj}$; NC versus HD, 0.83; NC versus ID, 1.1 × 10$^{-4}$; NC versus SD, 1.8 × 10$^{-27}$. **c**, Fraction of A-485-downregulated genes within the indicated CTCF-regulated gene groups. Genes regulated after CTCF^AID depletion are grouped into the indicated classes, and within each class, the fraction of genes regulated by A-485 is shown. **d**, Fraction of CTCF up- and downregulated genes within the specified groups of A-485-regulated genes. A-485-regulated gene class is defined

based on gene expression FC thresholds defined in Extended Data Fig. 1c. **e**, Overlap of CTCF^down and RAD21^down genes (left panel) and CTCF^up and RAD21^up genes (right panel) with A-485-downregulated genes. **f**, Fraction of genes bound by CTCF in promoter-proximal (±2 kb of TSS) regions (left panels) and strength of CTCF peaks (middle panels) within the indicated classes of genes regulated by CTCF^AID depletion, RAD21^AID depletion and A-485 treatment (EU-seq TPM ≥ 15). The right panels show the fraction of genes with CTCF peaks within ±200 bp from their TSS. **g**, Shown is the RAD21/CTCF enrichment ratio in the indicated classes of RAD21- and CTCF-regulated genes. RAD21/CTCF enrichment ratio in distal (>200 bp from TSS) and CTCF + RAD21 peaks are shown for comparison. Promoters lacking CTCF + RAD21 binding within ±200 bp of their TSS are excluded from this analysis. The box plots display the median and upper and lower quartiles; whiskers show 1.5 × IQR. Statistical significance was determined by two-sided Mann–Whitney *U*-test, followed by correction for multiple comparisons with the Benjamini–Hochberg method; **$P_{adj}$ < 0.001, *$P_{adj}$ < 0.05. RAD21-regulated gene class versus distal peak $P_{adj}$; NC, 5.0 × 10$^{-55}$; ID + HD, 0.23. CTCF-regulated gene class versus distal peak $P_{adj}$; NC, 1.6 × 10$^{-19}$; SD, 1.7 × 10$^{-6}$; ID, 5.9 × 10$^{-16}$; HD, 3.3 × 10$^{-11}$.

p300-dependent. Importantly, both CTCF^down and CTCF^up genes show similar CBP/p300 bias (Fig. 2d), suggesting CTCF facilitates and blocks E–P interactions with comparable prominence.

### Limited overlap between RAD21- and CTCF-regulated genes

RAD21^AID and CTCF^AID depletion-induced transcriptional changes correlate weakly (PCC = 0.18; Extended Data Fig. 4g), and the regulated genes show limited gene overlap (Fig. 2e). Still, RAD21^down genes are clearly biased for CTCF regulation: although RAD21^AID depletion downregulates only ~2% of expressed genes, 32% of these overlap with CTCF^down genes (OR = 17). Notably, 93% of genes downregulated by both RAD21^AID and CTCF^AID are also A-485^down (OR = 71). Overall, 86% of RAD21^down genes overlap with A-485^down (OR = 80, *P* < 2.2 × 10$^{-16}$), compared with 44% of RAD21^up (OR = 3.8). In contrast, 42% of CTCF^down and 52% of CTCF^up overlap with A-485^down (OR = 3.4 and 6.0, respectively). Thus, CTCF- and RAD21-regulated genes overlap only modestly, but their shared targets are strongly CBP/p300-dependent.

### RAD21- and CTCF-regulated genes differ in enhancer proximity

To examine enhancer prevalence near CTCF- and RAD21-regulated genes, we mapped candidate enhancers marked by H3K27ac + H2BK20ac and stratified them by H2BK20ac signal strength. Enhancers were prominently enriched within 50 kilobases (kb) of RAD21^down genes compared with random controls (Extended Data Fig. 5a,b). They were also prevalent near genes downregulated by both CTCF^AID depletion and A-485 (CTCF^down A-485^down), but far less frequent near genes downregulated only by CTCF^AID (CTCF^down A-485^ND). Consistent with enhancer involvement and cohesin-dependent looping, known enhancer targets such as *Sox2*, *Sik1*, *Dlk1-Dio3*, *Epha4*, *Klf4* and *Prdm14* were downregulated by RAD21^AID and/or CTCF^AID depletion (Supplementary Fig. 2a). At the *Prdm14* locus, mutation and inversion of the CTCF binding site redirects

enhancer activity to *Slco5a1* (refs. 50–52), explaining its upregulation after CTCF^AID depletion. Micro-C confirmed cohesin-dependent E–P loops at several of these loci (Supplementary Fig. 2b). Together, these results highlight cohesin–CTCF looping in enhancer-driven activation and reveal distinct enhancer prevalence near cohesin- versus CTCF-regulated genes.

### CTCF-regulated genes exhibit strong CTCF promoter binding

CTCF binding near promoters is thought to facilitate enhancer interactions[13–15]. We examined CTCF occupancy in promoter-proximal regions (±2 kb from transcription start site (TSS)) and near core promoters (<200 base pairs (bp) from TSS). CTCF binds near a large fraction of CTCF^down promoters, with both frequency and enrichment scaling with the degree of downregulation (Fig. 2f). In both CTCF^down and CTCF^up genes, CTCF is mostly bound within 200 bp of the TSS. In contrast, TSS-proximal CTCF binding is much less frequent in RAD21^down and A-485^down genes, indicating that promoter CTCF binding is not a general mechanism for enhancer targeting.

To assess promoter-bound CTCF's role in cohesin loop anchoring, we compared RAD21/CTCF enrichment ratios at promoters versus distal CTCF + RAD21 peaks. At RAD21^down promoters, the ratio is similar to distal CTCF peaks, although the gene set is small (Fig. 2g and Extended Data Fig. 6a,b). In contrast, CTCF^down promoters show much lower ratios, which decline further with increasing CTCF dependence. This weak cohesin enrichment may reflect the low prevalence of enhancers near CTCF^down A-485^down genes (Extended Data Fig. 5a,b).

**Promoter-bound CTCF activates genes without cohesin.** CTCF Y226A/F228A mutations abolish CTCF-anchored cohesin loops[53,54]. To separate cohesin-dependent from independent functions, we engineered mESCs expressing endogenous CTCF mutated at Y226A/F228A and fused with dTAG (*Gfp-Fkbp12^V36F-Ctcf^Y226A/F228A*; hereafter

**Fig. 3 | The cohesin loop anchoring defective CTCF^Y226A/F228A mutant retains the ability to activate genes with CTCF-bound promoters. a**, Schematic for generating CTCF^YF-AA cells. Endogenous *Ctcf* is modified by fusing it with *Puro-P2A-Gfp-Fkbp12^F36V* at the N terminus and introducing Y226A and F228A mutations. The mutation of Y226A and F228A is confirmed by sequencing. **b**, Genes regulated after CTCF^YF-AA depletion (2 h, top panels; 4 h, bottom panels) are grouped into indicated classes. Within each class, the fraction of genes regulated by the depletion of wild-type CTCF^AID is shown. Wild-type CTCF^AID-regulated gene class (CTCF^down reg. class, and CTCF^up reg. class) is defined based on gene expression FC thresholds after depletion (6 h) of CTCF^AID, as defined in Extended Data Fig. 4c,d. **c**, The heatmap shows FC in gene expression after the indicated perturbations. The analysis includes genes (*n* = 91) that are strongly downregulated after CTCF^AID depletion (≥2-fold down, EU-seq expression

TPM > 15 in CTCF^AID cells and TPM > 5 in other treatment conditions). The dotted line indicates CTCF^AID-downregulated genes that are downregulated <1.5-fold after CTCF^YF-AA depletion. Note that genes that are downregulated by CTCF^AID depletion, but not by CTCF^YF-AA depletion, are all downregulated by A-485. **d,e**, FC in gene expression after CTCF^AID and CTCF^YF-AA depletion and enrichment of CTCF in their promoter-proximal (±2 kb from TSS) regions. Genes downregulated after CTCF^AID depletion are grouped into: CTCF^down A-485^ND (**d**) and CTCF^down A-485^Down (**e**). Within each group, heatmaps show FC in gene regulation after CTCF^AID (6 h) and CTCF^YF-AA (2 h, 4 h) depletion. Genes are sorted by their downregulation after CTCF^YF-AA depletion (2 h). For the genes with CTCF binding ±2 kb from TSS, the position of CTCF binding and height of CTCF ChIP–seq peak are displayed. Region ±200 bp from TSS is highlighted.

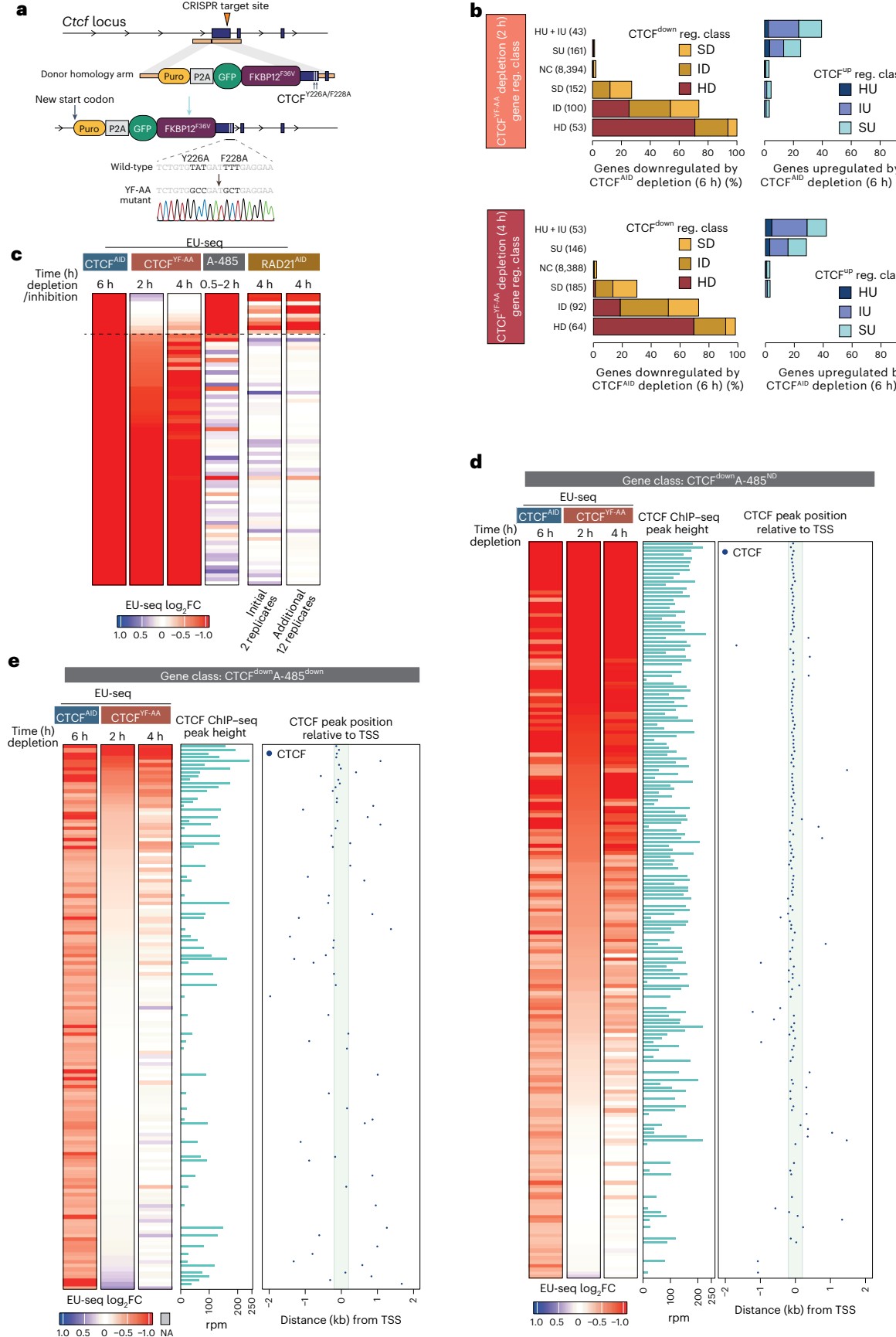

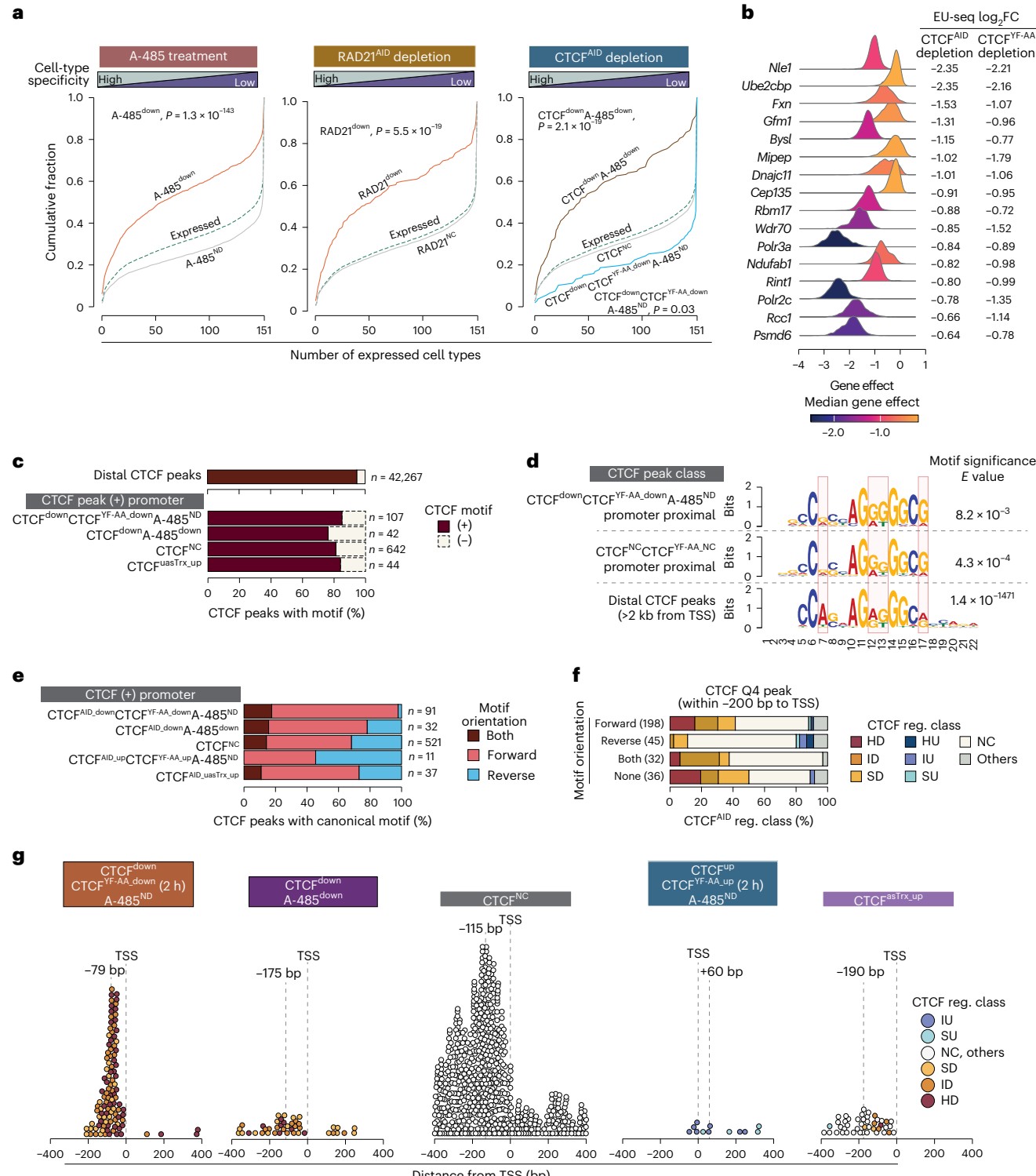

**Fig. 4 | CTCF functions as a transcription activator and repressor and regulates essential housekeeping genes. a**, Cell-type specificity of the indicated classes of A-485-, RAD21- and CTCF-regulated genes. For the indicated groups, the expression of genes is analyzed across 151 mouse tissues and developmental stages. 'Expressed' includes genes that are expressed in mESCs in the indicated conditions. *P* values were calculated using a two-sample Kolmogorov–Smirnov test. **b**, A list of common essential genes that are downregulated by both CTCF[AID] and CTCF[YY-AA] depletion but not by A-485. Common essential genes are defined by the DepMap database based on their essentiality for the proliferation of human cancer cell lines. **c**, Fraction of CTCF peaks with identifiable canonical CTCF binding motif. In the indicated classes of regulated genes, CTCF motifs are searched in ±100 bp from the CTCF peak summit (peak summit distance:

±400 bp from TSS). Distal (>2 kb away from TSS) CTCF peaks are included as a reference. **d**, Sequence motif enriched in CTCF peaks detected in distal and promoter (upstream 200 bp to downstream 50 bp from TSS) regions. **e**, CTCF motif orientation within CTCF peak regions in the promoters (±400 bp from TSS) of the indicated class of regulated genes. **f**, CTCF[AID] depletion-induced regulation of genes with strong CTCF (Q4 of CTCF peaks) binding in their promoters (upstream 200 bp from TSS). Genes bound by Q4 CTCF peaks are grouped by CTCF motif orientation, and within each group, the fraction of the CTCF-regulated gene class is depicted. **g**, Distribution of CTCF binding position in the promoter-proximal regions (±400 bp of TSS) of the indicated classes of regulated genes. In each group, the dotted lines indicate the TSS position and the median CTCF peak distance from TSS.

CTCF$^{YF-AA}$) (Fig. 3a). Cell growth was impaired upon CTCF$^{YF-AA}$ depletion (Extended Data Fig. 7a–c), indicating it retains essential functions.

Constitutive expression of CTCF$^{YF-AA}$ caused widespread transcriptional changes relative to CTCF$^{AID}$-expressing cells (Extended Data Fig. 7d–f), likely reflecting both primary and secondary effects. Importantly, acute (2–4 h) depletion of CTCF$^{YF-AA}$ downregulated hundreds of genes (Extended Data Fig. 7g,h), with the strongest targets largely overlapping those downregulated by wild-type CTCF$^{AID}$ depletion (Fig. 3b). Of the 72 genes reduced >2-fold upon CTCF$^{AID}$ depletion, 86% (62 of 72) were also downregulated ≥1.5-fold within 2 h of CTCF$^{YF-AA}$ depletion (Fig. 3c). These shared targets were largely insensitive to A-485 or RAD21$^{AID}$ depletion. In contrast, the minority of genes uniquely downregulated by CTCF$^{AID}$ depletion (10 of 72) were all sensitive to A-485, and many also to RAD21$^{AID}$ depletion.

Grouping CTCF$^{down}$ genes by A-485 response reinforced this distinction: CTCF$^{down}$A-485$^{ND}$ genes were mostly downregulated by CTCF$^{YF-AA}$ depletion (Fig. 3d), while CTCF$^{down}$A-485$^{down}$ genes were only modestly affected (Fig. 3e). In both cases, genes downregulated by CTCF$^{YF-AA}$ depletion were predominantly bound by CTCF at their promoters.

This suggests that the CTCF$^{YF-AA}$ mutant loses the ability to activate genes by cohesin loop anchoring but retains the ability to activate genes by promoter binding.

**Promoter-bound CTCF can repress transcription.** A small set of genes is commonly upregulated by both wild-type CTCF$^{AID}$ and CTCF$^{YF-AA}$ depletion. Unlike genes exclusively upregulated by CTCF$^{AID}$ depletion, these genes lack proximal enhancers but show frequent CTCF binding at or just downstream of the TSS (−50 to +400 bp) (Supplementary Fig. 3a–c). In contrast, genes upregulated only by wild-type CTCF$^{AID}$ depletion are enhancer-rich and A-485-sensitive, indicating that they may be upregulated by enhancer mistargeting. These results support reporter-based findings that promoter-bound CTCF can repress transcription[55,56], while indicating that this affects a small number of genes.

### CTCF activates essential housekeeping genes

Consistent with enhancer involvement in cell-type-specific regulation, RAD21$^{down}$, A-485$^{down}$ and CTCF$^{down}$A-485$^{down}$ genes show tissue-restricted expression (Fig. 4a). In contrast, genes downregulated by both CTCF$^{YF-AA}$ and CTCF$^{AID}$ depletion but not by A-485 (CTCF$^{down}$CTCF$^{YF-AA\_down}$ A-485$^{ND}$) are broadly expressed across mouse tissues. Promoter CTCF binding at these genes is conserved between mouse and human (Supplementary Fig. 3d), highlighting a conserved function. This group includes several DepMap-defined 'common essential' genes (that is, *Nle1*, *Bysl*, *Rbm17*, *Wdr70*, *Polr3a*, *Polr2c*, *Psmd6*) (Fig. 4b), suggesting that CTCF's essential role in cell proliferation likely stems from activating housekeeping genes.

### CTCF binding position and orientation dictate its functions

Our findings suggest that CTCF activates protein-coding genes, while previous work suggested it represses upstream antisense transcription (uasTrx) without affecting sense transcription[57]. To resolve this, we examined uasTrx regulation in our data. Despite the short-lived nature of uasTrx transcripts, computational analysis identified 80 promoters with >2-fold uasTrx upregulation (CTCF$^{uasTrx\_up}$) following CTCF$^{AID}$ depletion. About 60% of these showed CTCF binding and uasTrx upregulation in manual inspection, and were used for further analysis.

We then assessed CTCF motif features. Promoter-bound CTCF more often lacks a canonical motif than distal peaks (18.6% versus 5.3%; Fig. 4c), and de novo motif analysis shows sequence differences between promoter and distal sites (Fig. 4d). However, these differences do not distinguish CTCF-dependent from CTCF-independent promoters. Instead, motif orientation correlated with transcriptional outcome: only 2% of CTCF$^{down}$CTCF$^{YF-AA\_down}$A-485$^{ND}$ promoters had exclusively reverse motifs, versus 22–32% of CTCF$^{down}$A-485$^{ND}$, CTCF$^{uasTrx\_up}$ and CTCF$^{NC}$ promoters (Fig. 4e). Among promoters with strong CTCF binding near the TSS (±200 bp), 29% of forward/no-motif promoters were downregulated after CTCF$^{AID}$ depletion, compared with only 2% of exclusively reverse-motif promoters, which instead trended toward upregulation (Fig. 4f).

CTCF binding positions also varied with functional outcomes (Fig. 4g). In CTCF$^{down}$CTCF$^{YF-AA\_down}$A-485$^{ND}$ genes, CTCF localized tightly upstream of the TSS (median ~79 bp). In CTCF$^{up}$ genes, CTCF occupied core promoters and slightly downstream, while in CTCF$^{down}$A-485$^{down}$ and CTCF$^{NC}$ genes, binding was more dispersed and shifted upstream. Notably, in CTCF$^{uasTrx\_up}$ genes, CTCF bound further upstream in genes showing only uasTrx upregulation compared with genes showing both uasTrx upregulation and concomitant downregulation of sense transcript.

These results confirm a forward-orientation bias for promoter CTCF in CTCF$^{down}$ genes[14,26] and show that this bias is strongest in CTCF$^{down}$A-485$^{ND}$ genes, which are activated independently of cohesin and CBP/p300.

### CTCF acts as a canonical transcription activator

To test whether CTCF directly activates promoters, we performed luciferase assays on eight CTCF$^{down}$ and two CTCF$^{up}$ promoters bound by CTCF (Supplementary Fig. 4a). All eight CTCF$^{down}$ promoters showed reduced activity upon acute CTCF$^{YF-AA}$ depletion, and in six, mutation of the CTCF motif strongly diminished activity (Fig. 5a). The remaining two promoters (*Dusp12* and *Tab1*) lack canonical CTCF motifs, and manually predicted binding sites may not account for CTCF occupancy. Control promoters lacking CTCF sites (*Psmd4*, *Actn4*) remained unaffected, while CTCF$^{up}$ promoters (*Lsm11*, *Ocel1*) were derepressed upon CTCF$^{YF-AA}$ depletion, with motif mutations further enhancing activity (Fig. 5a and Supplementary Fig. 4b).

These results indicate that CTCF can function as both a direct activator and repressor of promoters, with motif orientation influencing its activator function.

### BORIS substitutes for CTCF in promoter activation

CTCF and its paralog BORIS (CTCFL) share highly similar DNA-binding zinc fingers but differ in their N- and C-terminal regions[58]. Unlike CTCF, BORIS cannot anchor cohesin loops but binds many CTCF-occupied sites, showing a preference for promoters over distal regions[59,60].

Using the same CTCF$^{AID}$ mESC system used here, previous work measured mRNA expression changes after swapping CTCF$^{AID}$ with BORIS or chimeric constructs: CTCF-N-terminus_BORIS-ZF_CTCF-C-terminus (CBC) or BORIS-N-terminus_CTCF-ZF_BORIS-C-terminus (BCB)[49].

Re-analysis of these RNA sequencing (RNA-seq) data revealed that BORIS restored expression of enhancer-independent CTCF targets (CTCF$^{down}$CTCF$^{YF-AA\_down}$A-485$^{ND}$), but not enhancer-dependent targets (CTCF$^{down}$A-485$^{down}$) (Fig. 5b). The BCB chimera partially restored enhancer-independent targets, likely due to weaker transgene induction (Supplementary Fig. 4c), while CBC partially restored both target types, consistent with partial cohesin loop restoration by chimeric CTCF-N-terminus-BORIS[34].

These results show that BORIS, similar to CTCF, is a competent transcription activator, explaining its promoter-biased binding.

### CTCF controls chromatin accessibility

To assess how CTCF affects promoter function, we analyzed chromatin accessibility using an independently generated degron line (*Gfp-Fkbp12$^{F36V}$-Ctcf*; $^{dTAG}$CTCF). After 3 h of $^{dTAG}$CTCF depletion, accessibility decreased >2-fold at 6.4% of regions but increased at only 0.03% (Fig. 6a), contrasting with the widespread bidirectional changes observed after 24 h of CTCF$^{AID}$ depletion[61]. Reductions scaled with CTCF enrichment and were stronger at distal than promoter peaks (Fig. 6b). At promoters, accessibility decreased only in

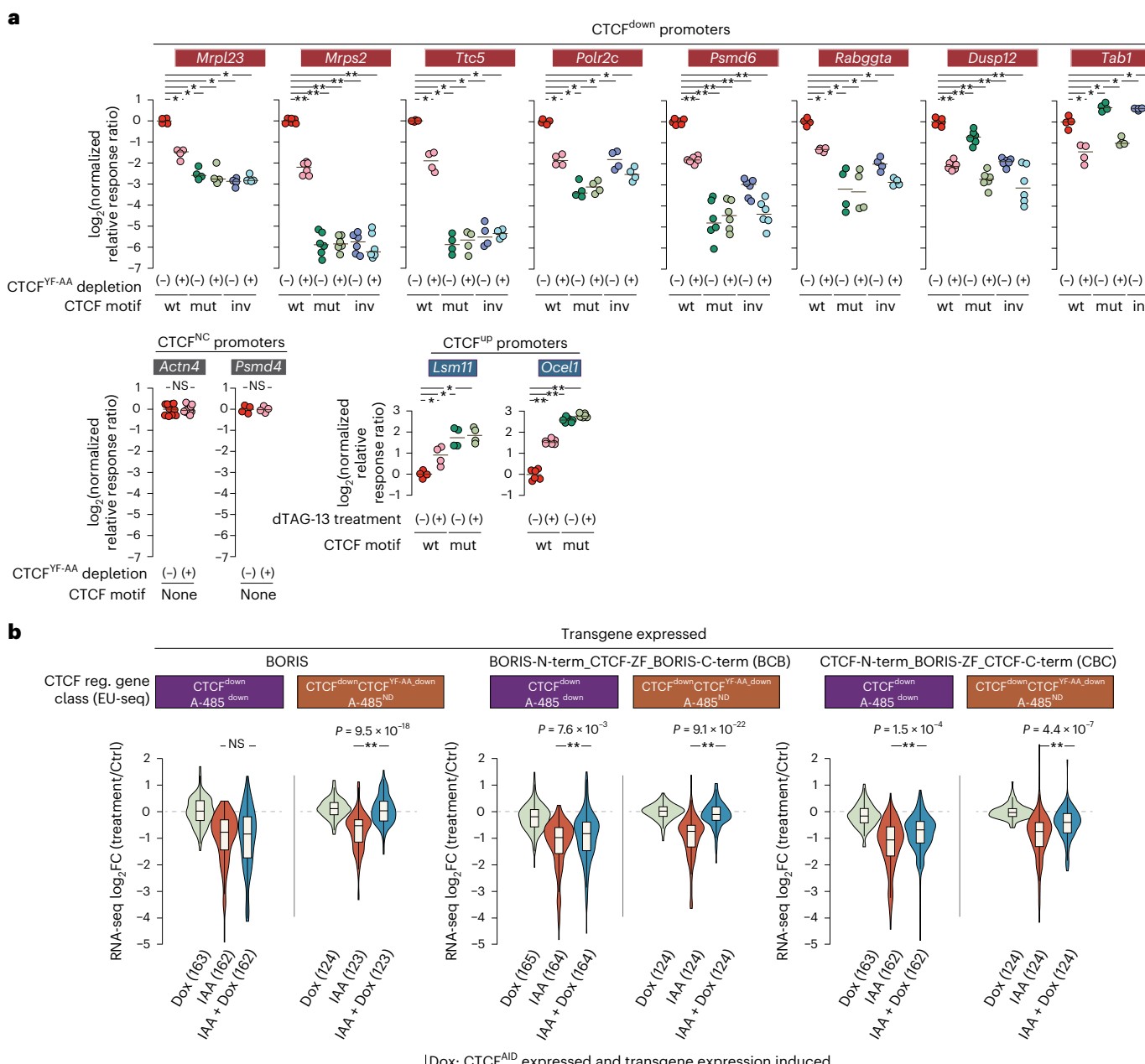

**Fig. 5 | CTCF's functional diversity is shaped by its position- and orientation-specific binding at promoters, and CTCFL can activate a subset of CTCF target genes. a**, Relative luciferase response ratio of the promoters of the indicated genes, selected from the specified CTCF-regulated gene class. Promoter activity is measured for wild-type (wt) promoter sequences, putative CTCF binding mutant (mut) sequences and putative CTCF binding motif inverted (inv) sequences. The reporter activity is measured in mESCs expressing dTAG-tag-fused CTCF$^{YF-AA}$, with (+) or without (−) depleting CTCF$^{YF-AA}$ (6 h) by dTAG-13 treatment. Circles indicate measurements from individual experiments, and a horizontal line indicates the median value. Biological replicates: $n$ = 4 for *Mrpl23*, *Ttc5*, *Polr2c*, *Rabggta*, *Tab1*, *Psmd4*, *Lsm11*; $n$ = 6 for *Mrps2*, *Psmd6*, *Dusp12*, *Ocel1*; $n$ = 10 for *Actn4*. Two-sided Mann–Whitney $U$-test, followed by correction for multiple comparisons with the Benjamini–Hochberg method; \*\*$P_{adj}$ < 0.001, \*$P_{adj}$ < 0.05, NS, not significant. Median values for individual comparisons are provided in the Source data. **b**, BORIS (CTCFL) overexpression rescues

expression of a subset of CTCF-regulated genes. The indicated transgenes were introduced into CTCF$^{AID}$ mESCs, and changes in mRNA expression were quantified by RNA-seq in the following conditions[49]: CTCF$^{AID}$ depletion (24 h) without transgene expression (IAA), transgene expression without CTCF$^{AID}$ depletion (Dox), CTCF$^{AID}$ depletion with simultaneous transgene expression (IAA + Dox). The CTCF-regulated gene class is defined based on the changes in nascent transcription after acute (6 h) CTCF$^{AID}$ depletion in this study. Shown is the FC in RNA-seq in the indicated class of CTCF-regulated genes in cell lines expressing the shown transgenes and treatment conditions ($n$ = 2 biological replicates per condition). Numbers of genes ($n$) per class are shown in brackets on the plot. The box plots display the median and upper and lower quartiles; whiskers show 1.5 × IQR. $P$ values were calculated using two-sided Mann–Whitney $U$-test, followed by correction for multiple comparisons with the Benjamini–Hochberg method. $P_{adj}$ values for individual comparisons are provided in the Source data. Dox, doxycycline.

CTCF[down] genes, with CTCF binding correlating with transcriptional downregulation (Fig. 6c). In contrast, CTCF[NC] and CTCF[uasTrx_up] genes retained promoter accessibility despite having CTCF binding. These results support that CTCF helps maintain nucleosome-depleted regions[62], and affects a small set of promoters where CTCF promotes transcription activation.

### CTCF promoter binding distinctly impacts Pol II recruitment

Next, we examined Pol II recruitment after [dTAG]CTCF depletion. In CTCF[down] genes, Pol II recruitment decreases both in promoters and in transcription end sites, with the magnitude of change corresponding with the extent of transcription downregulation in EU-seq (Fig. 6d). In contrast, in CTCF[uasTrx_up] genes, Pol II binding increases in TSS-proximal areas but not in downstream gene regions (Fig. 6d). CTCF binding also influences Pol II binding position, albeit subtly (Fig. 6e). CTCF depletion causes an upstream shift in Pol II binding in CTCF[uasTrx_up] genes, and to a lesser extent in CTCF[NC] genes. This shift is directly attributable to CTCF, as genes lacking CTCF binding in their promoters show no change in Pol II position.

Examining individual genes further supports these nuances (Fig. 6f and Extended Data Fig. 8a). Most CTCF[down] genes (for example, *Pabpc4*, *Nle1*, *Mrpl23*) show reduced promoter Pol II without positional shifts, while a small subset (*Rbm45*, *Vps36*, *Ttc5*, *Zbtb39*) displays weaker reductions but exhibits altered binding positions. Gratifyingly, Pol II binding is diminished at the *App* promoter (Extended Data Fig. 8a), the first promoter shown to be CTCF-dependent[22], although this gene is weakly expressed in mESCs (TPM < 15).

Among CTCF[up] genes, those with CTCF bound within the core promoter (*Mrps16*, *Kctd6*) exhibit upstream-shifted Pol II peaks upon [dTAG]CTCF depletion, while genes with CTCF bound downstream of TBP (*Ocel1*, *Mlst8*) show increased Pol II binding immediately downstream of the TSS (Fig. 6f and Extended Data Fig. 8b).

In CTCF[uasTrx_up] genes, [dTAG]CTCF depletion increases Pol II binding (Fig. 6f and Extended Data Fig. 8c). When CTCF binds immediately upstream of the TSS (*Gins4*, *Rps3a1*), Pol II shifts upstream but appears as a single broadened peak; when binding occurs further upstream (*Timm22*, *Azi2*), a new upstream Pol II peak appears after CTCF depletion. Notably, in CTCF[up] and CTCF[uasTrx_up] genes, Pol II accumulates precisely at CTCF binding sites, suggesting that CTCF binding may physically occlude Pol II binding or block elongation.

These findings reveal that CTCF functions as a position-specific transcription activator or repressor, promoting or hindering Pol II recruitment and/or elongation depending on promoter context.

### CTCF's non-architectural functions depend on its C terminus

To dissect domains required for CTCF's non-architectural functions, we engineered mESCs with allele-specific degron tags and targeted deletions of the N or C terminus (Extended Data Fig. 9a). From cells carrying biallelic dTAG-GFP-tagged CTCF, we deleted the N-terminal region (residues 1–265) in one allele and fused it with BromoTag (bTAG)[63], yielding [dTAG]*Ctcf*/[bTAG_ΔN]*Ctcf*. In this background, we truncated the C-terminal domain (residues 578–736) on the [dTAG]*Ctcf* allele by inserting an ALFA-tag and stop codon after zinc finger 11, yielding [bTAG_ΔN]*Ctcf*/[dTAG]*Ctcf*[ΔC] genotype.

GFP and ALFA-tag imaging confirmed expression of the engineered variants, and treatment with dTAG-13 or AGB1 selectively degraded the respective dTAG- or bTAG-fused proteins (Extended Data Fig. 9b). Functionally, loss of [bTAG_ΔN]CTCF impaired proliferation, whereas depletion of [dTAG]CTCF[ΔC] did not (Extended Data Fig. 9c), indicating the C terminus is essential for growth while the N terminus is dispensable.

Nascent transcription profiling revealed that [dTAG]CTCF[ΔC] depletion alone did not affect genes regulated by CTCF[AID] depletion, whereas [bTAG_ΔN]CTCF depletion strongly perturbed their expression (Fig. 7a). Genes likely dependent on CTCF's direct activator function (those commonly downregulated by CTCF[AID] and CTCF[YF-AA], but not by A-485) were unaffected by [dTAG]CTCF[ΔC] depletion but were globally downregulated upon [bTAG_ΔN]CTCF depletion or combined depletion of both (Fig. 7b). Similar trends were observed for antisense transcripts and commonly upregulated targets (Extended Data Fig. 10a,b). In contrast, genes uniquely upregulated by CTCF[AID] (but not CTCF[YF-AA]), or those downregulated by both CTCF[AID] and RAD21[AID] depletion, were largely unaffected by depletion of either variant (Extended Data Fig. 10c,d), possibly reflecting dominant-negative effects of truncated proteins on cohesin looping.

Together, these results indicate that CTCF's non-architectural functions—both transcriptional activation and repression—depend primarily on its C-terminal domain.

## Discussion

Cohesin and CTCF are central to chromatin organization, but their impact on global gene regulation has been debated[20,23,25,26,57,64].

---

**Fig. 6 | CTCF binding in promoters regulates chromatin accessibility and the recruitment and positioning of Pol II. a**, Change in global chromatin accessibility after acute (3 h) [dTAG]CTCF depletion. The percentage of ATAC-seq peaks showing a ≥2-FC in accessibility is indicated. Dotted lines indicate log₂FC of −1, 0 and 1. **b**, FC in chromatin accessibility in CTCF peak regions after [dTAG]CTCF depletion. Distal (>200 bp from TSS) and promoter (±200 bp from TSS) ATAC peaks are grouped into quartiles (Q1–Q4) based on overlapping CTCF peak height. Promoters without CTCF binding (No CTCF) were included as a reference. The box plots display the median and upper and lower quartiles; whiskers show 1.5 × IQR. Statistical significance was determined by two-sided Mann–Whitney U-test, followed by correction for multiple comparisons with the Benjamini–Hochberg method; **$P_{adj}$ < 0.001. $P_{adj}$ for the differences of ATAC signal changes at CTCF overlapping versus nonoverlapping (No CTCF) regions are: (Promoter) Q1, $3.0 \times 10^{-73}$; Q2, $1.3 \times 10^{-22}$; Q3, $2.9 \times 10^{-13}$; Q4, $1.1 \times 10^{-5}$. (Distal) Q1, $<1.0 \times 10^{-115}$; Q2, $1.0 \times 10^{-115}$; Q3, $2.3 \times 10^{-109}$; Q4, $1.2 \times 10^{-16}$. Numbers of peaks (n) per group are shown in brackets on the plot. **c**, In the indicated CTCF-regulated gene class, the change in chromatin accessibility after [dTAG]CTCF depletion. The analysis includes only promoters bound by CTCF (within ±200 bp of TSS). In the left and right panels, promoters are classified based on the regulation of sense or antisense transcripts, respectively, after CTCF[AID] depletion. The box plots display the median and upper and lower quartiles; whiskers show 1.5 × IQR. Statistical significance was determined by two-sided Mann–Whitney U-test, followed by correction for multiple comparisons with the Benjamini–Hochberg method; **$P_{adj}$ < 0.001, *$P_{adj}$ < 0.05. $P_{adj}$: (CTCF reg. class) HU + IU versus NC, 0.77; SU versus NC, 0.77; SD versus NC, $2.1 \times 10^{-4}$; ID versus NC, $4.2 \times 10^{-9}$; HD versus NC, $2.2 \times 10^{-14}$; (CTCF[asTrx] reg. class) Up versus NC, 0.57. Numbers of genes (n) per class are shown in brackets on the plot. **d**, Change in Pol II binding in the indicated groups of CTCF-regulated gene class. Pol II enrichment is quantified in the promoter (±400 bp from TSS) and nonpromoter regions (+500 bp from TSS to transcription end site (TES)). Normalization was performed based on the assumption that the Pol II binding at the gene body region of the CTCF nonregulated genes does not change between the control and [dTAG]CTCF-depleted conditions. Change in Pol II binding is analyzed by ChIP-seq after [dTAG]CTCF depletion (4 h). The box plots display the median and upper and lower quartiles; whiskers show 1.5 × IQR. Statistical significance was determined by two-sided Mann–Whitney U-test, followed by correction for multiple comparisons with the Benjamini–Hochberg method; **$P_{adj}$ < 0.001, *$P_{adj}$ < 0.05. $P_{adj}$ values for individual comparisons are provided in the Source data. Numbers of genes (n) per class are shown in brackets on the plot. The dotted line indicates a log₂FC of 0. **e**, Pol II binding in promoters of the indicated groups of CTCF-regulated genes, with (+) or without (−) CTCF binding in their promoters. Pol II binding is analyzed by ChIP-seq in control and [dTAG]CTCF-depleted (4 h) conditions. The dotted line indicates the TSS position. **f**, Upper genome browser tracks show nascent transcript expression of the representative genes exhibiting decreased (*Pabpc4*) or increased (*Kctd6*) transcription or increased upstream antisense transcription (*Gins4*) after CTCF[AID] depletion. Lower panels show the binding of CTCF, TBP and Pol II in the promoters of the indicated genes. Pol II binding is analyzed with or without depletion of [dTAG]CTCF. For additional examples, see Extended Data Fig. 8a–c. ATAC-seq, assay for transposase-accessible chromatin using sequencing.

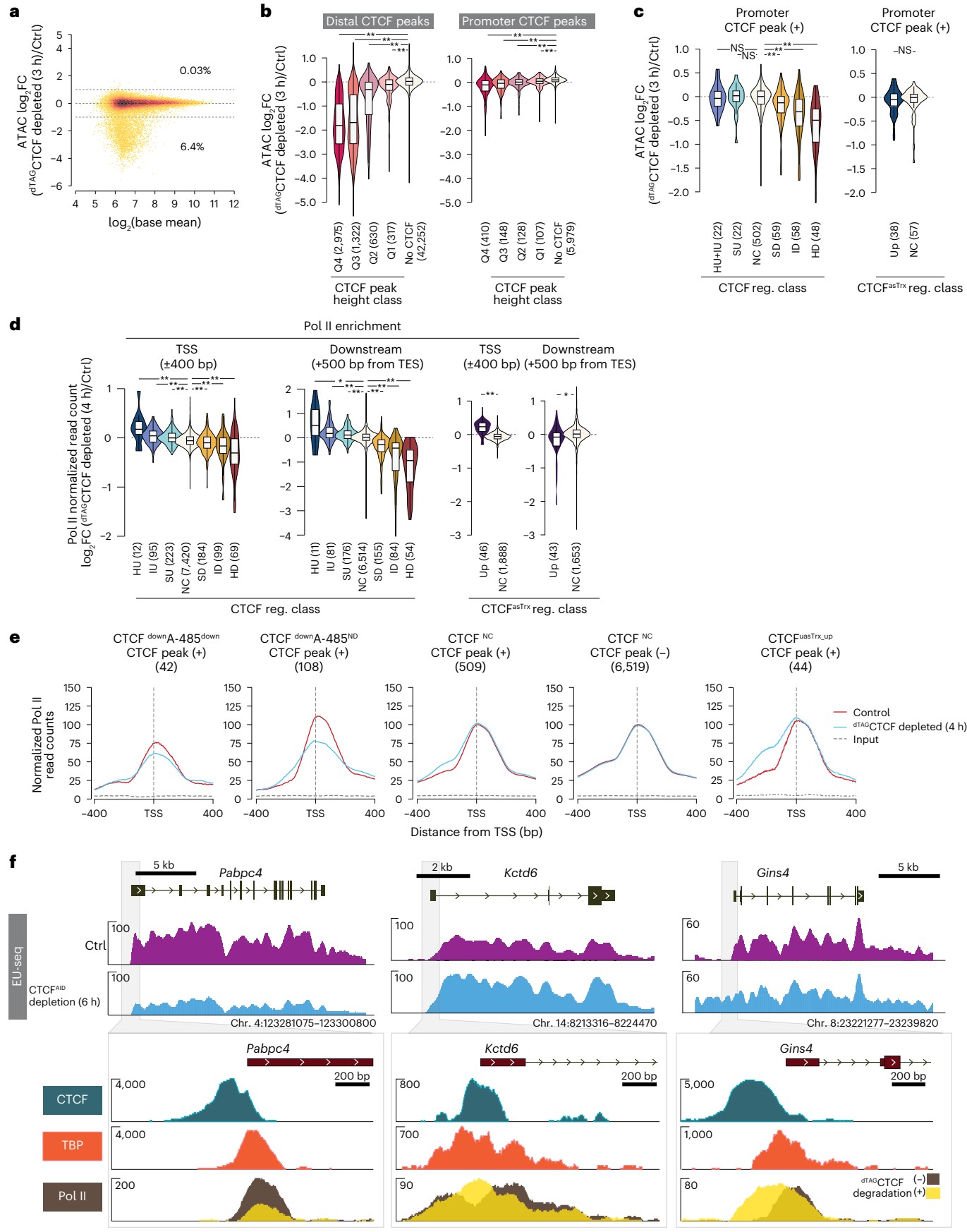

By delineating their architectural and non-architectural roles in enhancer-dependent and enhancer-independent gene regulation, our findings help to reconcile diverging views and provide a unified model of their role in transcription control.

By systematically examining cohesin and CTCF function within the context of CBP/p300-regulated genes, we uncover a broad gene regulatory impact that was largely missed in earlier studies. Acute RAD21 or CTCF depletion dysregulates hundreds of genes, yet the changes are subtle in magnitude, and most changes fall below standard statistical thresholds—leading previous efforts to underestimate their scope[20,25,26,46,65,66]. Crucially, without first defining CBP/p300-regulated genes, our own analyses too would have dismissed these effects as noise. This demonstrates the power of multi-way comparisons to detect weak but genuine regulation. Three factors may explain this subtle regulation. First, degron systems do not fully eliminate RAD21 or CTCF, and residual protein may sustain partial function. Second, some genes are activated by both promoters and enhancers[38], so disrupting E–P looping produces only mild reductions in partially enhancer-dependent genes. Third, cohesin is not universally required: while distal enhancers depend strongly on cohesin, proximal enhancers can function independently[67,68]. This framework reconciles the anticipated broad regulatory scope of cohesin–CTCF looping with the modest transcriptional changes observed after acute cohesin and CTCF removal. Multiple orthogonal assays confirm that these modest effects are reproducible and biologically meaningful (Supplementary Note 1).

Modest transcriptional changes after RAD21 and CTCF depletion led to the 'time-buffering' model[20], which proposed that cohesin is required for establishing E–P contacts. Once E–P contacts are established, an unknown memory mechanism can preserve gene expression for hours without cohesin. We show that over one-third of CBP/p300-regulated genes are downregulated by RAD21$^{AID}$ depletion within 2–4 h in NPCs, and among the most strongly A-485-regulated genes (>4-fold, $n = 515$), over half are downregulated by RAD21$^{AID}$ depletion at 4 h, implying a lack of long-term E–P contact memory in these contexts. Given the diversity of enhancers across cell types, cohesin likely influences thousands of genes at the organismal level. Still, the requirement of cohesin is not universal, suggesting alternative mechanisms of E–P engagement (Supplementary Note 3).

The prevailing view is that cohesin regulates transcription by extruding chromatin loops[1–5], although alternative models suggest roles in recruiting transcription factors[69–71], counteracting Polycomb repression[46] or forming loops through 'loop capture'[72]. We find that RAD21$^{down}$ genes are strongly enriched for CBP/p300 targets but lack H3K27me3, making it unlikely that gene downregulation is caused by enhanced Polycomb repression. Transcription factors generally bind through their DNA-binding domain, rather than cohesin, and a loop capture model would require improbable specificity for selectively capturing CBP/p300-dependent enhancers and their target promoters. Thus, while alternative mechanisms remain possible, our findings are most consistent with the loop extrusion model, whereby cohesin looping helps bridge the distance gap between enhancers and nearby promoters. The loop extrusion model may explain both

the notable relationship between enhancers and CBP/p300 dependency of nearby genes[38], and the apparent scarcity of bona fide active gene-skipping enhancers[39]. Interestingly, the fact that an overwhelming majority of genes strongly affected by RAD21 depletion overlap with CBP/p300-regulated genes indicates that cohesin primarily promotes transcription through CBP/p300-dependent enhancers.

The architectural function of CTCF is extensively studied, but its long-suggested non-architectural transcriptional roles[21,22] have long been largely overlooked (Supplementary Note 4). Some recent studies suggested that promoter-proximal CTCF binding facilitates interaction with distal enhancers[15,16], while others hinted that CTCF may also act as a transcriptional activator[26,66,73]. However, they did not directly demonstrate this activator role or separate it from CTCF's architectural functions. Our genome-scale analyses confirm five ways by which CTCF likely regulates transcription (Fig. 7c,d): (1) facilitating enhancer target activation via cohesin anchoring, (2) blocking enhancer mistargeting, (3) suppressing antisense transcription at bidirectional promoters, (4) acting as a canonical activator and (5) repressing transcription in rare cases.

CTCF's functional diversity appears to be dictated by the position and/or polarity of its DNA binding (Fig. 7c,d). Forward-oriented binding just upstream of the promoter activates transcription, binding further upstream blocks antisense transcription and binding within or just downstream of the promoter represses sense transcription. This suggests CTCF likely represses sense and antisense transcription by the same principle—affecting Pol II binding or elongation—with its impact depending on its binding position relative to the TSS.

Architectural and non-architectural functions map to distinct domains and affect distinct gene sets. The N terminus anchors cohesin and regulates cell-type-specific genes, whereas the C-terminal domain mediates non-architectural functions, primarily controlling conserved housekeeping genes. These dual roles of CTCF in enhancer-dependent and enhancer-independent gene regulation explain the limited overlap between CTCF- and RAD21-regulated genes[20,27].

CTCFL binds many CTCF sites but cannot anchor cohesin loops[34]. By competing with CTCF, it may disrupt loop architecture. Our results indicate that CTCFL can function as a transcription activator. This likely explains its preferential binding to promoters[59,60]. Aberrant CTCFL expression in cancer could thus rewire enhancer-driven gene programs by altering genome architecture[74,75] while maintaining CTCF-dependent housekeeping gene expression, thereby supporting continued proliferation. Together, these findings show that CTCF and CTCFL operate through distinct and overlapping mechanisms.

CTCF is broadly conserved across bilaterians[28] and blocks enhancer activity in both *Drosophila* and mammals[6–8,32]. Yet, its functions have diverged: unlike mammalian CTCF, *Drosophila* CTCF neither anchors cohesin loops nor directly activates transcription[29,31,76,77]. Instead, it acts primarily as an insulator and is dispensable for *Drosophila* embryonic development and cell proliferation.

In contrast, mammalian CTCF is essential for the proliferation of cultured cells, and the cohesin loop-defective mutant (CTCF$^{YF-AA}$) retains the ability to activate hundreds of genes, including essential housekeeping genes. This indicates that CTCF's pan-essentiality in mammalian cells

---

**Fig. 7 | The C-terminal region is required for CTCF's non-architectural functions. a**, Genes regulated after the depletion (2 h) of $^{bTAG-ΔN}$CTCF, $^{dTAG}$CTCF$^{ΔC}$ and their combination are grouped into the indicated classes. Within each class, the fraction of genes down- and upregulated by the depletion of wild-type CTCF$^{AID}$ is shown. In CTCF$^{AID}$ cells, gene regulation was quantified after 6 h of CTCF$^{AID}$ depletion. **b**, The heatmap shows FC in gene expression after depletion of $^{bTAG-ΔN}$CTCF, $^{dTAG}$CTCF$^{ΔC}$ and their combination. The analysis includes genes ($n = 124$) that are commonly downregulated by >1.5-fold after CTCF$^{AID}$ and CTCF$^{YF-AA}$ depletion and are not downregulated by A-485 treatment. **c,d**, Proposed model of gene regulation by cohesin and CTCF. In their architectural roles (**c**), cohesin mediates loop extrusion that facilitates interaction of distal enhancers—particularly CBP/p300-dependent ones—with

their target promoters. Convergently bound CTCF anchors cohesin to halt loop extrusion, thereby facilitating enhancer interaction with their targets and blocking enhancers from activating nontarget promoters. Beyond its structural functions, CTCF regulates gene expression through position- and orientation-specific binding near TSS (**d**). When bound immediately upstream of the TSS, CTCF acts as a transcriptional activator. When positioned further upstream, it represses uasTrx. If bound within the core promoter or just downstream of the TSS, it blocks transcription from the sense strand. CTCF's role as a transcriptional activator depends critically on its binding to motifs oriented in the forward direction relative to the gene. In contrast, motif orientation appears to be less important for its repressive functions on both sense and antisense transcription. NA, not available.

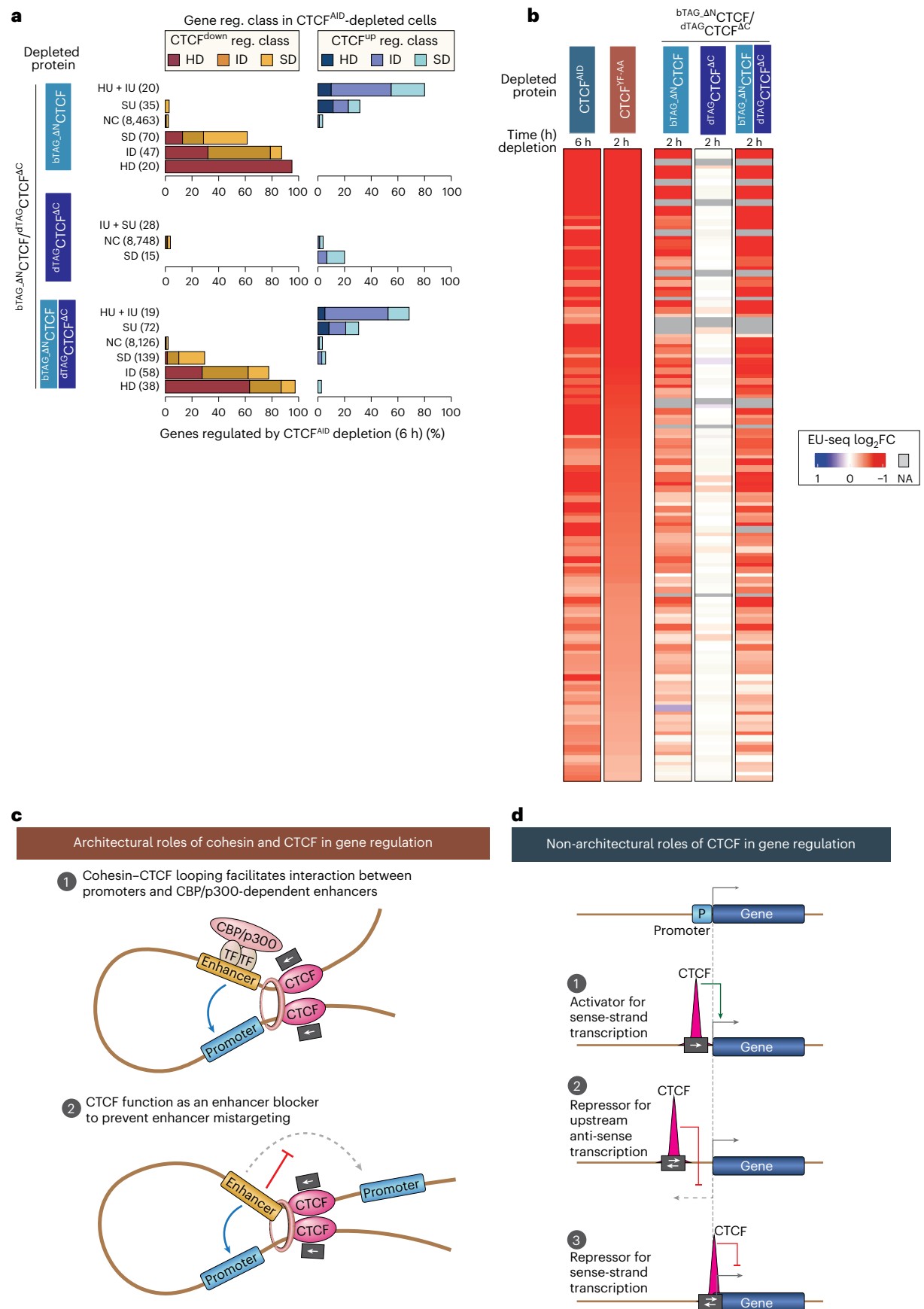

likely arises from its role as a transcription activator rather than as a loop anchor. This functional divergence may explain why CTCF is dispensable in *Drosophila* but indispensable in mammal cell proliferation.

Overall, our findings reveal how cohesin and CTCF integrate architectural and non-architectural functions to regulate genes through both enhancer-dependent and independent mechanisms, providing a unified model of their roles in transcription.

## Online content

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

## Methods

Our research complies with the relevant Danish and European ethical policies and did not require a specific board to approve our study.

### Cell culture

mESCs were purchased from Sigma-Aldrich (E14TG2a; Sigma-Aldrich, cat. no. 08021401). RAD21[AID] mESCs were kindly provided by R. Klose[46], and CTCF[AID] mESCs by E. P. Nora and B. G. Bruneau[26]. Unless indicated otherwise, mESCs were cultured in a custom-made (C.C.Pro) N2B27 medium consisting of a 1:1 mix of DMEM/F12 and neurobasal medium but lacking arginine and lysine. Before use, the medium was supplemented with 1,000 U ml⁻¹ leukemia inhibitory factor (Merck Millipore), 1 μM PD0325901, 3 μM CT-99021 (custom-made by ABCR), 100 μM 2-mercaptoethanol, 0.5 × B27 supplement (Thermo Fisher Scientific), 0.5 × N2 supplement (made in-house or from Thermo Fisher Scientific), L-lysine and L-arginine. Unless indicated otherwise, the cells were treated with the following chemical concentrations: A-485 10 μM, dTAG-13 0.1–0.2 μM, AGB1 0.1–0.2 μM or indole-3-acetic acid (IAA) 500 μM.

### Differentiation of RAD21[AID] mESCs to NPCs

RAD21[AID]-expressing mESCs were first differentiated into epiblast stem cells by culturing them on dishes coated with 10 ng ml⁻¹ fibronectin (Merck Millipore), in N2B27 supplemented with 20 ng ml⁻¹ Activin A and 12 ng ml⁻¹ fibroblast growth factor 2 (FGF2) (PeproTech). Epiblast stem cells were further differentiated into NPCs by seeding 0.5–1 × 10⁵ cells per cm² to fibronectin-coated dishes in N2B27 without growth factors. After 5 d, cells were dissociated with 0.5 × TrypLE (Thermo Fisher Scientific) and seeded to 0.1% gelatinized tissue culture plates in N2B27 supplemented with 10 ng ml⁻¹ FGF2 and 10 ng ml⁻¹ epithelial growth factor (PeproTech). NPCs were cultured in 0.1% gelatin-coated plates in N2B27 supplemented with 10 ng ml⁻¹ FGF2 and 10 ng ml⁻¹ epithelial growth factor.

### CRISPR–Cas9 gene editing

The endogenous CTCF locus was edited through the CRISPR–Cas9 system by co-transfecting Cas9-single guide RNA (sgRNA) plasmids with one of the following donor homology arms: *Puro-P2A-Gfp-Fkbp12^{F36V}-Ctcf* homology arm, *Puro-P2A-Gfp-Fkbp12^{F36V}-Ctcf^{Y226A/F228A}* homology arm, *Hygro-P2A-3XFLAG-BromoTag-Ctcf* homology arm and *Ctcf-ALFA-tag-P2A-Neo-Stop codon-polyA tail* homology arm (full sequences of the homology arms are provided in Supplementary Table 1). Parental mESCs (ES-E14TG2a) were obtained from the European Collection of Authenticated Cell Cultures (Sigma-Aldrich, cat. no. 08021401). To generate the CTCF^{YF-AA} cell line, both alleles of *Ctcf* were mutated to Y226A/F228A tagged with *Gfp-Fkbp12^{V36F}*, generating *Gfp-Fkbp12^{F36V}-Ctcf^{Y22A6/F228A}*. The ^{dTAG}CTCF cell line was generated by targeting both alleles of *Ctcf* with *Puro-P2A-Gfp-Fkbp12^{F36V}-Ctcf*. In this background, we targeted one of the *Ctcf* alleles with *Hygro-P2A-3XFLAG-BromoTag-Ctcf* homology arm, generating the ^{bTAG_ΔN}CTCF/^{dTAG}CTCF cell line. In the ^{bTAG_ΔN}CTCF/^{dTAG}CTCF cell line, the C terminus of the ^{dTAG}*Ctcf* allele was targeted with the *Ctcf-ALFA-tag-P2A-Neo-Stop codon-polyA tail* homology arm, yielding the ^{bTAG_ΔN}CTCF/^{dTAG}CTCF^{ΔC} cell line (Extended Data Fig. 9a).

mESCs were grown in 2i media and 500 ng of donor and 1 μg of sgRNA plasmid (for the generation of ^{dTAG}CTCF and CTCF^{YF-AA} cell lines) or 1.5 μg of donor and 500 ng of sgRNA plasmid (for the generation of ^{bTAG_ΔN}Ctcf/^{dTAG}Ctcf^{ΔC} cell line) were transfected using Lipofectamine 2000 (Thermo Fisher Scientific, cat. no. 11668019), according to the manufacturer's instructions. After overnight incubation, the transfection mix was exchanged to normal 2i media. The next day, all cells were dissociated using TrypLE (Thermo Fisher Scientific), spun down and transferred to a new plate. Antibiotic selection was then applied by using 0.5 μg ml⁻¹ Puromycin (Invitrogen, cat. no. ant-pr-1), 1.75 μl ml⁻¹ hygromycin (Invitrogen, cat. no. ant-hg-1) or 200 μg ml⁻¹ geneticin (Gibco, cat. no. 10131035) according to the selection markers used. After

6 d of selection, the cells were transferred to normal 2i media, and individual colonies were picked and genotyped using KOD Xtreme Hot Start DNA Polymerase (Novagen, cat. no. 71975). The ^{bTAG_ΔN}Ctcf/^{dTAG}Ctcf^{ΔC} cell line was confirmed by immunostaining against GFP and the ALFA-tag, and by treatment with dTAG-13 and AGB1 (Extended Data Fig. 9b). We noted that while cells expressing CTCF^{YF-AA} and ^{bTAG_ΔN}CTCF/^{dTAG}CTCF^{ΔC} proliferated continuously in culture, their growth rate appeared slower than cells expressing CTCF^{AID} and ^{dTAG}CTCF.

### Assay for transposase-accessible chromatin using sequencing

CTCF-dependent chromatin accessibility was analyzed in mESCs expressing ^{dTAG}CTCF. Cells were first washed in pre-warmed PBS and dissociated using TrypLE (Thermo Fisher Scientific). Then, 2 × 10⁵ cells were transferred to a LoBind microfuge tube (Eppendorf) and centrifuged at 500g for 5 min at 4 °C. After centrifugation, the cells were resuspended in buffer A (10 mM HEPES pH 7.9, 10 mM KCl, 1.5 mM MgCl₂, 0.34 M sucrose, 10% glycerol, 10% Triton X), mixed by tapping and incubated on ice for 7 min. Following another centrifugation, nuclei were resuspended in lysis buffer (10 mM Tris-HCl pH 7.4, 10 mM NaCl, 3 mM MgCl₂, 0.3% Igepal). A quarter of nuclei were then transferred to a new LoBind tube, briefly vortexed, incubated on ice for 15 min, vortexed again briefly and pelleted down at 600g for 10 min at 4 °C. After removing the lysis buffer, nuclei were recovered with 2.5 μl of 2 × TD buffer and 2.5 μl of transposase (TDE1) mixture (Illumina). Following mixing by pipetting, the nuclei were incubated for 30 min at 37 °C on a rocking mixer. The transposed DNA was purified using the MinElute PCR Purification kit (QIAGEN) following the manufacturer's protocol. The transposed genomic regions were PCR-amplified using Nextera Primers (https://www.encodeproject.org/documents/860c95a8-fdc4-46bd-b9b5-d44242ff20ef/@@download/attachment/ATAC_garberlab_protocol.pdf) and NEBNext High-Fidelity 2X PCR Master Mix (NEB, cat. no. M0541). The PCR product was purified using Agencourt AMPure XP beads (Beckman Coulter) and sequenced on a NextSeq 500 sequencer (Illumina).

### Chromatin immunoprecipitation followed by sequencing

The cells were crosslinked for 10 min at room temperature with 1% formaldehyde solution with gentle agitation and quenched with 0.125 M glycine. Crosslinked cells were washed once with PBS, added to swelling buffer (10 mM Tris-HCl pH 8.0, 1.5 mM MgCl₂, 10 mM NaCl, 0.5% NP-40) and incubated on ice for 10 min. After centrifugation, pellets were suspended in RIPA buffer (10 mM Tris-HCl pH 8.0, 1 mM EDTA, 140 mM NaCl, 1% Triton X-100, 0.1% SDS, 0.1% sodium deoxycholate) containing protease inhibitor cocktail (Roche), and then sonicated using a BioRuptor sonicator (Diagenode). After centrifugation, solubilized chromatin was recovered and incubated with 5 μg of Pol II antibody (cat. no. 14958, Cell Signaling Technology) bound to Dynabeads M-280 sheep anti-Rabbit IgG (Thermo Fisher Scientific). After overnight incubation at 4 °C, the magnetic beads were washed with RIPA buffer twice and with RIPA high-salt buffer (10 mM Tris-HCl pH 8.0, 1 mM EDTA, 500 mM NaCl, 1% Triton X-100, 0.1% SDS, 0.1% sodium deoxycholate), LiCl buffer (10 mM Tris-HCl pH 8.0, 1 mM EDTA, 250 mM LiCl, 0.5% NP-40, 0.5% sodium deoxycholate) and TE buffer (Invitrogen). The bound materials were eluted with elution buffer (10 mM Tris-HCl pH 8.0, 5 mM EDTA, 300 mM NaCl, 0.1% SDS) overnight at 65 °C and treated with RNase A for 30 min at 37 °C, followed by further incubation with Proteinase K for 1 h at 37 °C. Then, DNA was purified with a PCR purification kit (QIAGEN). Chromatin immunoprecipitation followed by sequencing (ChIP–seq) libraries were prepared by using the NEBNext Ultra II DNA prep kit (NEB E7645) following the manufacturer's instructions and sequenced on a NextSeq 500 sequencer (Illumina).

### EU-seq

To analyze nascent transcription changes under various conditions, cells were treated with DMSO or with the indicated chemicals for the specified time points. Unless indicated otherwise, A-485 at 10 μM,

dTAG-13 at 0.1–0.2 µM and IAA at 500 µM were used for treatment. Cells treated with or without chemicals were pulse-labeled with 0.5 mM 5-EU for 20 min before sample collection. After the 5-EU labeling, cells were quickly washed once with PBS, and immediately lysed in RLT buffer, and RNA was isolated using the RNeasy Mini Kit (QIAGEN) according to the manufacturer's protocol. Labeled RNA was further enriched using the Click-iT Nascent RNA Capture Kit (Thermo Fisher Scientific) according to the manufacturer's instructions. EU-seq libraries were prepared using the NEBNext Ultra II RNA First Strand Synthesis Module (cat. no. E7771) combined with NEBNext Ultra II Non-Directional RNA Second Strand Synthesis Module (cat. no. E6111) or NEBNext Ultra II Directional RNA Library prep kit for Illumina (cat. no. E6111), following the manufacturer's instructions, and subsequently sequenced on either a NextSeq 500 sequencer or a NextSeq 2000 sequencer (Illumina).

### Cloning of CTCF promoter luciferase plasmids

First, 5 µg of pGL421 plasmid (Addgene, cat. no. 118695) was linearized by digestion with XhoI and HindIII and gel-extraction-purified. Genomic sequences encompassing promoters of analyzed genes were amplified by nested PCR using first KOD polymerase and then PrimeStar MAX DNA polymerase to generate homology regions to the linearized pGL421. Pieces were gel-extraction-purified and inserted into the vector by Hot Fusion cloning, followed by transformation, clone picking, plasmid isolation and sequence verification by Sanger sequencing. Where indicated, promoter luciferase plasmids with inverted and mutated CTCF sites were generated by using the plasmids with the native promoters as templates for amplification of two pieces upstream and downstream of the putative CTCF site with primers in between encoding the inversion/mutation as well as homology for homology-directed cloning. The two pieces were combined with linearized pGL421 and cloned by Hot Fusion cloning, followed by clone picking, plasmid isolation and sequence verification by Sanger sequencing. Promoter sequences used for reporter assays are provided in Supplementary Table 2.

### CTCF promoter luciferase assay

Promoter activity was measured using the Dual-Glo Luciferase Assay System. Briefly, 500 ng of pGL474 and 1,000 ng of pGL421 were mixed in 150 µl of OptiMEM with 3 µl of Lipofectamine 2000, incubated at room temperature for 10 min and transfected into $5 \times 10^5$ CTCF$^{YF-AA}$ mESCs for 10 min. Cells were centrifuged, the transfection mix was removed and cells were resuspended in 1.5 ml of 2i/LIF medium. For each transfection, $5 \times 10^4$ cells in 150 µl were seeded into four geltrex-coated wells of a 96-well plate, allowed to settle for 20 min at room temperature and then incubated overnight for attachment. At 18 h after transfection, cells were treated with DMSO (1:500) or dTAG-13 (200 nM) for 6 h. After one PBS wash, 48 µl of PBS was added before lysis. Firefly luciferase was assayed with 48 µl of Dual-Glo Luciferase reagent (10 min at room temperature), followed by luminescence measurement on a Tecan Spark. Stop & Glo reagent (48 µl) was then added to quench Firefly and activate Renilla luciferase, and luminescence was measured again after 10 min. Raw Firefly and Renilla signals were background-corrected using blank wells. Values <30 relative luminescence units after correction were set to 30 to reduce variability at low signals. Relative response ratios were calculated as Firefly/Renilla and normalized to the relative response ratio of the wild-type promoter with DMSO treatment.

### Immunoblotting

The day before sample preparation, $5 \times 10^5$ cells were seeded for each sample in a 6-well plate. The next day, treatment was done with vehicle control (DMSO, 1:500) or dTAG-13 (100 nM) for 2 h. Cells were collected by scraping and lysed in RIPA, followed by $5 \times 10$-s sonication at 40% amplitude and centrifugation for 10 min at 21,000$g$ to collect the sample from the supernatant. The protein concentration was determined with the bicinchoninic acid assay. The 25-µg protein sample mixed with 4 × NuPage LDS to 1× was heated at 70 °C for 5 min, and the sample was loaded into a 4–12% NuPage Bis-Tris 1.0-mm mini protein gel. Seeblue Plus2 pre-stained was used as a ladder. The gel was run with 1 × MOPS buffer for 20 min at 80 V and then for 40 min at 200 V. Proteins on the gel were transferred to a nitrocellulose membrane in a transfer buffer for 90 min at 30 V. The membrane was washed with water, stained with Ponceau for 5 min and washed with water for 5 min, followed by imaging. Ponceau was washed away with 0.1 M NaOH for 2 min, and the membrane was washed with water, followed by overnight blocking with 5% skim milk in 1 × TBST at 4 °C for 14–18 h. Milk was washed away with 1 × TBST, and the blot was incubated with 1:1,000 anti-CTCF antibody (rabbit, cat. no. D31H2, Cell Signaling Technologies) in 1 × TBST with 1% BSA for 2 h followed by 4 × 3-min washes with 1 × TBST. The blot was then incubated with anti-rabbit HRP antibody for 1 h, washed 4 × 3 min with 1 × TBST, developed with Clarity Western ECL substrate and imaged on an ImageQuant LAS 4000 or an Amersham ImageQuant 800 system.

### Immunofluorescence staining

Treatment, fixation and staining for CTCF microscopy and quantitative image-based cytometry were performed as follows. Cells (25,000 per well) were seeded to geltrex-coated wells in a Perkin Elmer Cell Carrier 96w plate and allowed to adhere overnight. Cells were treated with vehicle control (DMSO, 1:500) or dTAG-13 (100 nM) for 2 h. Then, cell media was removed by inversion of the plate, and the cells were fixed with 4% formaldehyde in PBS for 15 min. Cells were then washed with PBS, and the plate was stored in the fridge until staining. For staining, cells were permeabilized with 0.5% Triton X-100 in PBS for 5 min and blocked for 30 min with antibody diluent (DMEM with 10% FCS, 0.02% sodium azide, filtered). Then, cells were incubated with anti-CTCF (1:1,000, Cell Signaling Technology, cat. no. 3418) and anti-ALFA-tag (1:500, SYSY Antibodies, cat. no. N1583) antibodies in an antibody diluent for 2 h, washed twice with PBS and incubated with anti-rabbit Alexa 568 in an antibody diluent with DAPI. Finally, the cells were washed twice with PBS. Images of the stained cells were acquired with the screening microscope described below.

### Microscopy and image-based cytometry

Microscopy images and images for quantitative image-based cytometry were acquired on an automated Perkin Elmer Phenix spinning disk screening microscope. The microscope was equipped with two 16-bit OEM Andor Zyla sCMOS cameras with 2,160 × 2,160 pixels with a 6.5-µm pixel size. The objectives used were a ×20 water immersion numerical aperture 1.0 W Plan apochromat or a ×40 water immersion numerical aperture 1.1 LD C-Apochromat. Harmony 4.8 image analysis software was used for the quantification of images. The analysis pipeline included background correction, nuclear segmentation, filtering of border objects, quantification of the individual signals (mean, sum and median per nuclei) and generation of the data tables. The data tables were further processed in R using ggplot2 for visualization. Representative example image projections from each condition, with set contrast and brightness settings for each staining, are displayed.

### Reference genome annotation

Mouse and human genome annotation and reference genomes were downloaded from GENCODE (mouse: GRCm38.p6 release M25, gencode.vM25.basic.annotation.gtf; human: Grch37 version release 29, gencode.v29lift37.basic.annotation.gtf)[78]. As a reference, the transcript type of 'protein coding' and 'lincRNA' was chosen for representative genes. To build the mESC transcript reference used in the EU-seq, we applied a two-step transcript annotation. First, the expressed transcripts were selected using the criteria of TPM > 2 in RNA-seq (GSE140363) and the presence of H3K4me3 ChIP–seq peak (GSE146328) within 1 kb from TSS. The longest isoform was selected if a single H3K4me3 peak was proximal to multiple transcripts annotated to the same. For the genes where the H3K4me3 peak was not detected,

the longest isoform was chosen as a representative transcript and combined with the reference transcripts.

## Processing of RNA-seq data

Raw RNA-seq data from CTCF[AID]-expressing mESCs (GSE140363), with or without CTCF[AID] depletion and with or without induction of transgenes (BORIS, CBC: CTCF-N–BORIS-ZF–CTCF-C, or BCB: BORIS-N–CTCF-ZF–BORIS-C), were downloaded from GEO and reanalyzed. Adapters were trimmed using Cutadapt (https://doi.org/10.14806/ej.17.1.200). Reads were mapped to the reference genome using STAR (v.2.6.1a)[79] with removal of the noncanonical junctions. Reads mapped to transfer RNA and ribosomal RNA regions were removed using Bedtools (v.2.23)[80]. The number of reads mapped to an exon was counted on a gene basis by using HTseq (v.0.11.1)[81]. The TPM was calculated using R.

## Processing of EU-seq data

Adapter trimming was performed as described in the RNA-seq section. Read sequences were aligned to the mm10 mouse genome using bwa meme (v.1.0.4)[82] with the soft clipping option for supplementary alignments. Low-mapping-quality reads (<q 10) were removed using samtools (v.1.4)[83]. Reads mapped to rRNA and tRNA regions, obtained from the UCSC genome browser, were removed using Bedtools. For the 30-, 60- and 120-min treatment data, based on the Pol II elongation rates, gene body regions up to 30, 90 and 240 kb from the TSS were used for differential gene expression (DGE) analysis. The number of reads mapped to defined regions was counted using HTseq. Transcripts with low mapped reads (average reads ≤20 between replicates in both control and treatment conditions) were excluded from DGE analysis. The $\log_2$ fold-change ($\log_2$FC) was calculated by using the result function of DESeq2 with the default scaling method of DESeq2[84], which uses the median of relative abundance. Asymmetrical DGE profiles resulted in a bias toward the opposite direction of the asymmetry. To adjust for this, the median of DEseq2 $\log_2$FC values from all the expressed genes was subtracted from the DEseq2 $\log_2$FC values and thus used as normalized $\log_2$FC values. On the calculation of log FC of the additional 12 replicates, count data were processed using DESeq2. Scaling factors for each sample were determined with the estimateSizeFactors function, and the median $\log_2$FC across expressed genes was calculated. These scaling factors were then applied to the matched control and treatment count data. For each gene, the normalized treatment count was divided by the normalized control count, the ratio was transformed to a $\log_2$ scale and the median $\log_2$FC calculated from 12 replicates was subtracted to yield the adjusted $\log_2$FC values. Unless described otherwise, FCs from distinct time points were summarized by calculating the averages. The TPM was calculated using the filtered transcripts based on read counts. Further, to keep only transcripts with robust expression, those with average TPM of control and treatment conditions >15 were defined as expressed.

In the calculation of antisense transcript regulation, only the strand-specific EU-seq data were used. Reads mapped to the antisense strand of upstream regions of −1.7 kb to −200 bp from the TSS were counted. The regulation was calculated only in antisense transcripts corresponding to expressed sense transcripts, and those with low-mapping reads (average reads ≤10 between replicates in both conditions) were excluded from DGE analysis. For the upregulated antisense transcripts used in the presented analyses, we manually checked the corresponding gene tracks to confirm that the transcripts are genuine antisense transcripts and not artifacts deriving from overlapping or adjacent transcription. Normalization and asymmetrical bias correction were performed using the scaling factors and median $\log_2$FC values derived from the sense transcripts DGE analysis. For the gene track visualization, we used deeptools and bigWigMerge (UCSC tool)[85] to generate bigwig files using scaling factors calculated from DESeq2.

## Classification of EU-seq regulation class

Unless specified otherwise, transcripts in each experimental condition were classified according to their FCs after treatment, as follows. HU: ≥2-fold upregulation. IU: <2- and ≥1.5-fold upregulation. SU: <1.5- and ≥1.3-fold upregulation. NC: <1.2-fold upregulation and <1.2-fold downregulation. SD: ≤1.5- and >1.3-fold downregulation. ID: ≤2- and >1.5-fold downregulation. HD: > 2-fold downregulation. Others: <1.3- and ≥1.2-fold upregulation, or ≤1.3- and >1.2-fold downregulation. For the A-485 regulation class, the following criteria were applied. ND: ≤1.2-fold downregulation. ID: ≤2- and >1.5-fold downregulation. HD1: ≤4- and >2-fold downregulation. HD2: >4-fold downregulation. For RAD21-regulated genes in NPCs, the FC values at 2-h and 4-h timepoints are averaged and classified as follows. HD: >1.3-fold downregulation on both 2-h and 4-h treatment with an average FC of >2-fold downregulation. ID: >1.3-fold downregulation on both 2-h and 4-h treatment with an average FC of ≤2- and >1.5-fold downregulation. SD: >1.3-fold downregulation at 2-h and 4-h treatment with an average FC of ≤1.5- and >1.3-fold downregulation. If the number of genes within any class is fewer than 10, the class is combined with the adjacent class, and the combined group is shown with a joint name (for example, HU + IU). In classifying antisense transcripts, only those exhibiting ≥2-fold upregulation were selected and were manually reviewed to confirm genuine upregulation, ensuring the changes were not due to the changes of overlapping or neighboring transcripts. Antisense transcripts that were confirmed as genuinely upregulated through this manual verification were used for the presented analyses. Processed EU-seq gene expression data are provided in Supplementary Table 4.

## Statistical analysis and reproducibility

Otherwise mentioned, $P$ values were calculated using the two-sided Mann–Whitney $U$-test and corrected for multiple comparisons using the Benjamini–Hochberg method (R package stats v.3.6.2). We performed nonparametric statistics tests (two-sided Mann–Whitney $U$-test and two-sample Kolmogorov–Smirnov test) that do not assume normality or equal variances, and we did not formally test these assumptions. No statistical method was used to predetermine sample size. No data were excluded from the analyses. The experiments were not randomized. The investigators were not blinded to allocation during experiments and outcome assessment.

## Reporting summary

Further information on research design is available in the Nature Portfolio Reporting Summary linked to this article.

## Data availability

Raw EU-seq, ChIP–seq and ATAC-seq data generated in this study have been deposited in the NCBI Gene Expression Omnibus (GEO) under the SuperSeries accession GSE262521 (https://www.ncbi.nlm.nih.gov/geo/query/acc.cgi?acc=GSE262521). A list of datasets generated in this work is provided in Supplementary Table 3, and processed EU-seq data are provided in Supplementary Table 4. Processed read counts, peak region sets and genome browser tracks are provided as supplementary files under the corresponding SubSeries accessions: ATAC-seq (GSE262516), ChIP–seq (GSE262519) and EU-seq (GSE262520). The following publicly available data were used: mouse and human gene annotations were from GENCODE (https://www.gencodegenes.org); FANTOM5 CAGE dataset was from ArrayExpress (https://www.ebi.ac.uk/arrayexpress/); Promoter Elements were referenced from EPD (https://epd.epfl.ch//index.php); Motif references were downloaded from JASPAR (https://jaspar.genereg.net/). DepMap datasets were downloaded from the DepMap portal (https://depmap.org). The orthologue reference was downloaded from https://www.ensembl.org/info/data/biomart. The following publicly available sequencing datasets were reanalyzed: RNA-seq in CTCF[AID] ESCs with or without depletion of CTCF[AID], and with or without overexpressing BORIS or CTCF-BORIS chimera (GSE140363). EU-seq in ESCs treated without

and with A-485 (GSE146328). ChIP-exo-seq in ESCs on CTCF (GSE98671). ChIP–seq in NPCs on CTCF and RAD21 (GSE262551). ChIP–seq in ESCs on H3K4me3 (GSE146328), TBP (GSE146328), H3K27me3 (GSE186349), Input control (GSE146328), CTCF (GSE178982), Rad21 (GSE178982). Following CTCF ChIP–seq and ChIP-exo-seq, data in a variety of tissues and cell lines were reanalyzed. Mouse: 3134 cells (GSE61236), 3T3-L1 cells (GSE95533), AML12 cells (GSE95116), splenic B cells (GSE44637), bone marrow-derived macrophages (GSE189975), bone marrow (GSE49847), brain (GSE114606), brown preadipocytes (GSE74189), cerebellum (GSE49847), CFUMk cells (GSE156074), CH12 cells (GSE49847), CMP cells (GSE159503), cortex (GSE49847), distal forelimbs (GSE101714), DP cells (GSE141223), ECOMG-derived neutrophils (GSE93127), embryoid bodies (GSE119874), EpiLC cells (GSE183828), erythroid cells (GSE142006), erythroid progenitor cells (GSE150415), female germline stem cells (GSE137771), forebrain (GSE127870), G1E cells (GSE156074), GMP cells (GSE156074), GSC cells (GSE183828), GSCLC cells (GSE183828), heart (GSE91813), Hepa-1c1c7 cells (GSE154387), HPC7 cells (GSE48086), HSPC cells (GSE131583), kidney (GSE49847), liver (GSE91731), MEF cells (GSE99197), MEL cells (GSE181234), mature olfactory sensory neurons (GSE112153), nephron progenitor cells (GSE90016), anterior neural progenitor cells (GSE160654), pancreas (GSE59119), Patski cells (GSE59779), pro-B cells (GSE109909), round spermatids (GSE70764), small intestine (GSE49847), spinal motor neurons (GSE196170), spleen (GSE111772), splenic plasmablasts (GSE44637), stomach (GSE91488), testis (GSE49847), thymus (GSE180937), trophoblast stem cells (GSE110950). Human: A-673 cells (GSE185132), adrenal gland (GSE106071), ascending aorta (GSE143103), astrocytes (GSE30263), B cells (GSE206145), bipolar neuron (GSE96269), 5637 cells (GSE193886), brain (GSE209256), brain microvascular endothelial cells (GSE30263), breast epithelium (GSE105608), CLB-Ga cells (GSE224242), CMP cells (GSE231486), colonic mucosa (GSE209074), coronary artery (GSE127477), DLD-1 cells (GSE142746), esophagus muscularis mucosa (GSE142970), esophagus squamous epithelium (GSE105931), gastrocnemius medialis (GSE143020), gastroesophageal sphincter (GSE127521), GMP cells (GSE231486), Gp5d cells (GSE180150), ESCs (GSE211101), ESC-derived SC-beta cells (GSE211101), HAP1 cells (GSE180691), HCT116 cells (GSE184106), heart right ventricle (GSE175279), HEK293T cells (GSE206145), HeLa S3 cells (GSE32883), HepG2 cells (GSE30226), HT1080 cells (GSE153869), Jurkat cells (GSE130140), K562 cells (GSE180175), kidney (GSE33213), Kuramochi cells (GSE152885), L1207 cells (GSE193886), left lung (GSE188029), left ventricle myocardium inferior (GSE187266), liver (GSE127549), lower leg skin (GSE105391), limbal stem cells (GSE192625), lung (GSE33213), MCF 10A cells (GSE183381), MCF7 cells (GSE181460), MEP cells (GSE231486), mucosa descending colon (GSE208481), Mutu-1 cells (GSE160973), pancreas (GSE174993), Peyers patch (GSE105594), PLC/PRF/5 cells (GSE209849), 22Rv1 cells (GSE200168), psoas muscle (GSE209062), right atrium auricular (GSE127378), RPE cells (GSE196727), RT-112 cells (GSE193886), SD48 cells (GSE193886), SH-SY5Y cells (GSE141278), sigmoid colon (GSE105852), suprapubic skin (GSE139782), testis (GSE105739), thoracic aorta (GSE127422), thyroid gland (GSE105921), tibial artery (GSE105707), tibial nerve (GSE105554), U2OS cells (GSE175731), vagina (GSE105477). The following publicly available processed files were used for data visualization. Micro-C data in RAD21[AID] ESCs with or without IAA treatment (GSE178982) (GSE178982_RAD21-UT_pool.mcool and GSE178982_RAD21-AID_pool.mcool). Source data are provided with this paper.

## Code availability

All analyses were performed with standard, publicly available tools as described in Methods.

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

## Acknowledgements

We thank the members of the Choudhary lab for their helpful discussions. The Novo Nordisk Foundation Center for Protein Research is financially supported by the Novo Nordisk Foundation (grant nos. NNF14CC0001, NNF24SA0098829). C.C. is supported by the Novo Nordisk Foundation Distinguished Investigator Bioscience and Basic Biomedicine grant (no. NNF22OC0074677) and an ERC advanced grant (no. ACT-SIGNAL, 101142708). Y.H. was supported by a Grant-in-Aid for JSPS Overseas Postdoctoral Fellows, the Osamu Hayaishi Memorial Scholarship for Study Abroad and funding from the Institute for Promotion of Tenure Track, University of Miyazaki. We thank the CPR Imaging Platform, the CPR Big Data Management Platform, and the CPR and DanStem Genomics Platform for their assistance. We thank R. Klose for providing mESCs expressing *Rad21-mAid-Gfp* (RAD21[AID]) and E. P. Nora and B. G. Bruneau for providing mESCs expressing *Ctcf-mAid-Gfp* (CTCF[AID]).

## Author contributions

C.C. and T.N. conceived the project. T.N. performed all bioinformatic analyses and prepared all figures. T.N., Y.H., S.K., N.M.S., G.P., E.M. and C.C. designed the research, performed the experiments, analyzed the data and interpreted the results. C.C. supervised the project. T.N. and C.C. wrote the manuscript with input from all coauthors.

## Competing interests

The authors declare no competing interests.

## Additional information

**Extended data** is available for this paper at https://doi.org/10.1038/s41588-025-02404-x.

**Correspondence and requests for materials** should be addressed to Chunaram Choudhary.

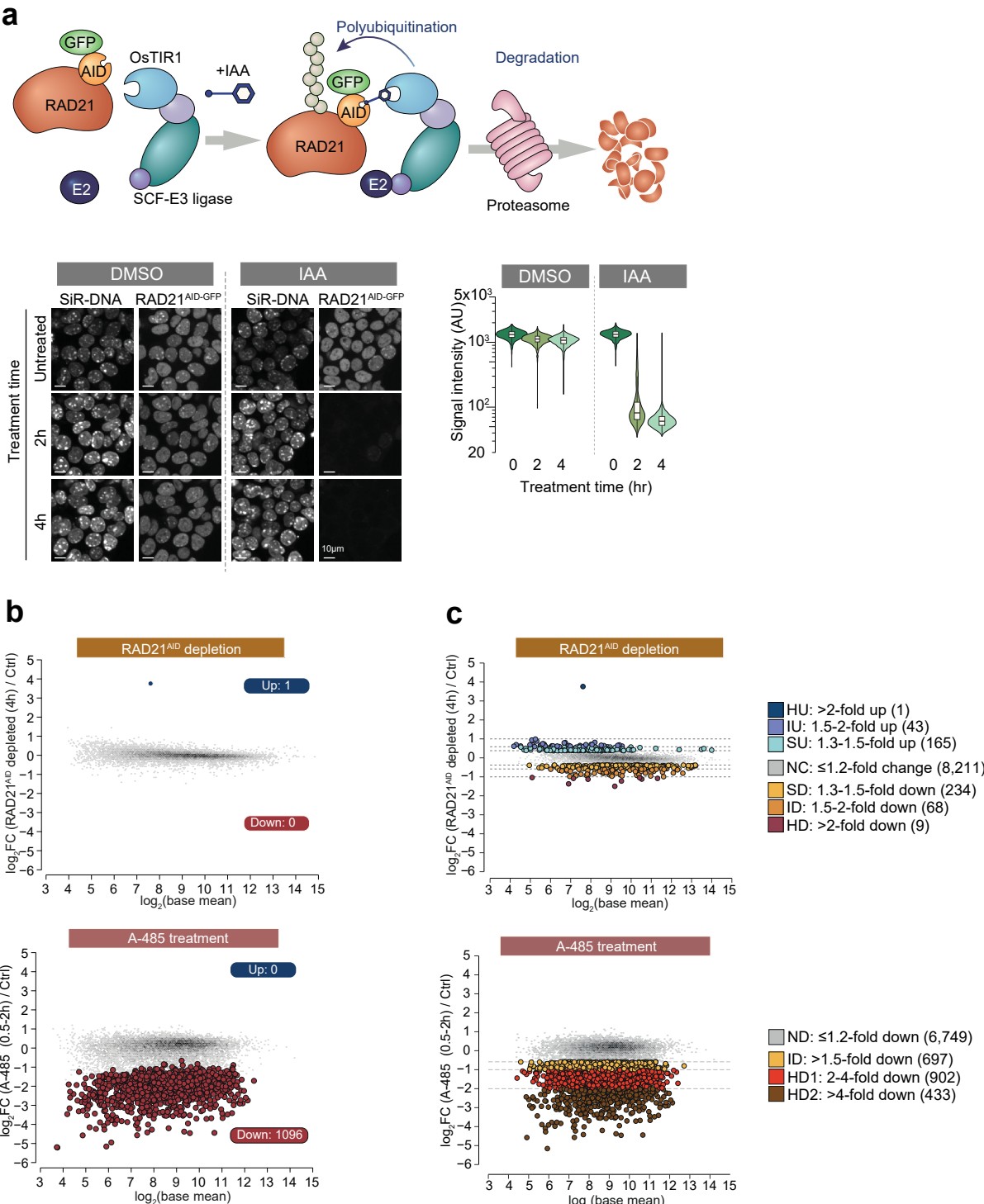

**Extended Data Fig. 1 | The scope of RAD21 and CBP/p300 in gene regulation and classification of regulated genes. a**, Schematic representation of auxin-induced proteasomal degradation of RAD21^AID-GFP (hereafter RAD21^AID) (top panel). Cells were exposed to the indicated treatments for the specified time, and RAD21^AID depletion was analyzed by microscopy-based imaging (n = 1) (bottom panels). Scale bar indicates 10 μm. RAD21^AID expression is quantified by analyzing GFP expression, and nuclei are visualized with SiR-DNA. The violin plots display the mean GFP intensity distribution in each condition (n = 1,000 cells/condition). The box plots show upper and lower quartiles, the line indicates the median, and the whiskers show the 1.5× interquartile range. **b**, The number of significantly (P_adj <0.05) up- and down-regulated genes after acute RAD21^AID depletion (4 h) and A-485 treatment (0.5-2 h). Transcription changes were

quantified in mESCs expressing RAD21^AID using 5-ethylidine uridine nascent RNA labeling and next-generation sequencing (EU-seq) (n = 2). Transcription changes are compared in cells without or with depletion of RAD21^AID by treatment with IAA (500uM). A-485-induced nascent transcription data are from ref.37. This analysis includes all genes expressed at TPM > 5. **c**, Fold-change-based classification of genes regulated after acute RAD21^AID depletion (4 h) and A-485 treatment (0.5-2 h). Regulated genes are classified using the indicated fold-change in gene expression after the specified treatments. The analysis only includes genes that are expressed at TPM > 15. Dotted lines indicate fold-change of 1.3, 1.5, and 2 (up-regulation and down-regulation) for Rad21^AID depletion, and fold-change threshold of 1.5, 2, and 4 (down-regulation) for A-485 treatment.

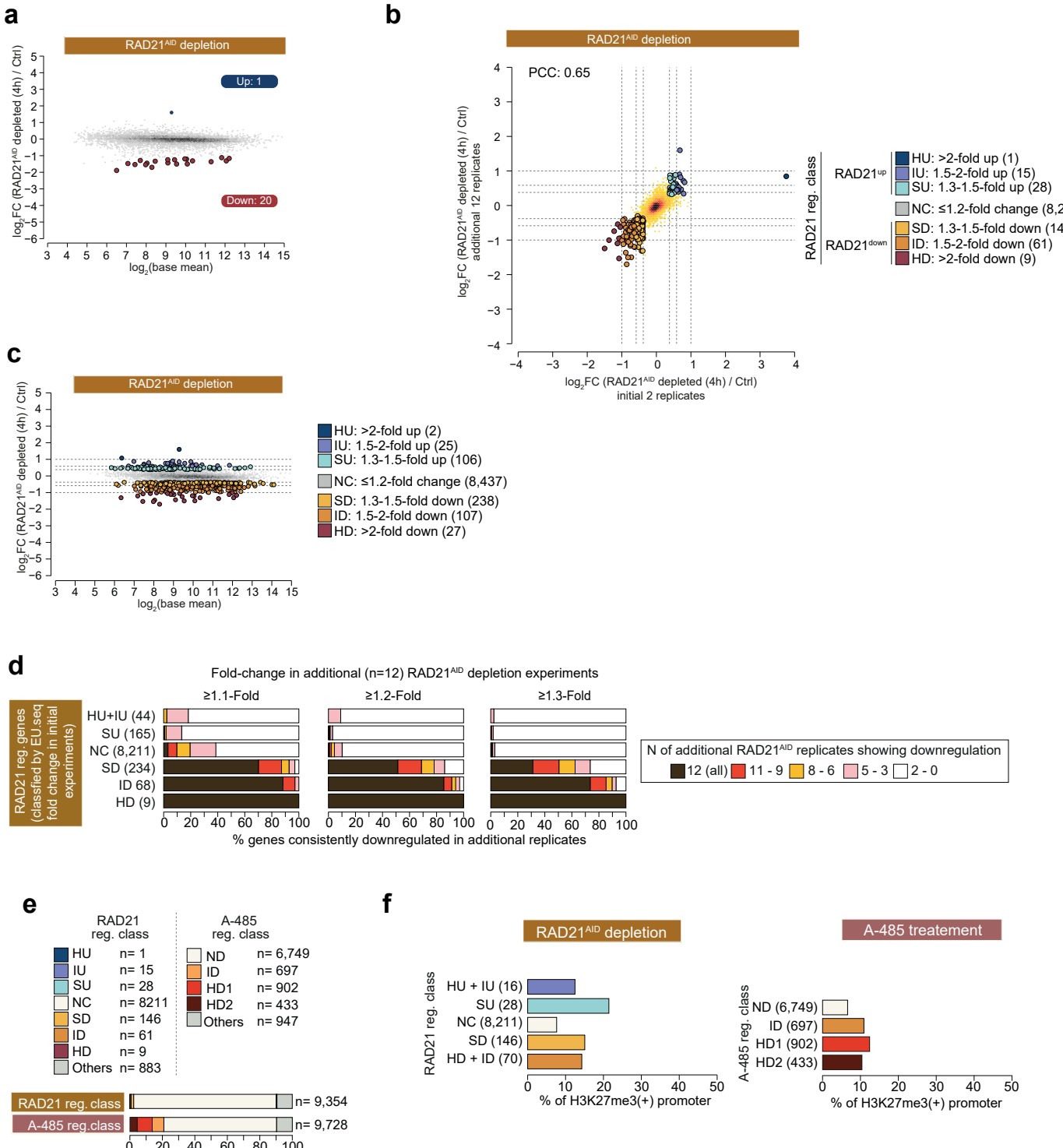

**Extended Data Fig. 2 | See next page for caption.**

**Extended Data Fig. 2 | Genes downregulated in the initial RAD21^AID depletion experiments show reproducible downregulation in additional replicates.**
**a**, The number of significantly ($P_{adj}$ < 0.05) up- and down-regulated genes in mESC after acute (4 h) RAD21^AID depletion. Transcription changes were quantified using EU-seq (n = 12 replicates). The analysis includes all genes expressed at TPM > 5.
**b**, Correlation between gene expression changes after RAD21^AID depletion in the initial experiments (n = 2) and independently performed new experiments (n = 12). PCC: Pearson´s correlation coefficient. Dotted lines indicate fold-change of 1.3, 1.5, and 2 (up- and down-regulations). **c**, Genes up- and down-regulated by RAD21^AID depletion (n = 12 replicates). Up- and down-regulated genes are grouped into indicated gene regulation classes. The analysis only includes genes that are expressed at TPM > 15. Dotted lines indicate fold-change of 1.3, 1.5, and 2 (up- and down-regulations). **d**, Reproducibility of gene regulation by RAD21^AID depletion in the initial experiments and new replicates. RAD21-regulated genes were classified based on fold changes in initial experiments (n = 2). Subsequently, gene expression changes were quantified in 12 additional replicates. Within the RAD21-regulated gene class defined from the initial experiments, shown is the fraction of genes showing ≥1.1, ≥1.2, and ≥1.3 fold downregulation in the indicated number of additional replicates. **e**, Number of genes in the indicated RAD21^AID and A-485 gene regulation class. RAD21-regulated gene class is defined based on the EU-seq fold-change in the initial two biological replicates (Extended Data Fig. 1c), and additionally showing consistent >1.3-fold up- or down-regulation in at least six out of 12 new replicates (Extended Data Fig. 2c). A-485 gene regulation class is defined by the fold-change in EU-seq after 0.5-2 h of A-485 treatment (Extended Data Fig. 1c). **f**, Genes downregulated by RAD21^AID depletion, and A-485 treatment show minimal difference in Polycomb-catalyzed H3K27me3 enrichment. Genes regulated by the specified treatments are grouped into the indicated regulated gene class, as defined in Extended Data Fig. 2e. Within the specified treatment conditions and regulated gene class, genes exhibiting H3K27me3 in promoter regions (+/-2kb from TSS) are shown.

**a**

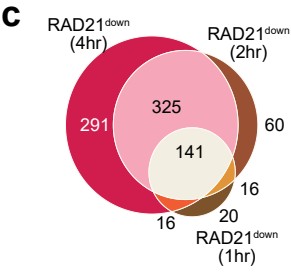

**b**

| RAD21 reg. class (1hr) | | RAD21 reg. class (2hr) | | RAD21 reg. class (4hr) | | A-485 reg. class | |
|---|---|---|---|---|---|---|---|
| HU | n= 16 | HU | n= 11 | HU | n= 11 | ND | n= 7,231 |
| IU | n= 70 | IU | n= 72 | IU | n= 120 | ID | n= 672 |
| SU | n= 165 | SU | n= 314 | SU | n= 373 | HD1 | n= 733 |
| NC | n= 8,997 | NC | n= 8,091 | NC | n= 7,707 | HD2 | n= 515 |
| SD | n= 163 | SD | n= 338 | SD | n= 387 | Others | n= 952 |
| ID | n= 33 | ID | n= 173 | ID | n= 266 | | |
| HD | n= 2 | HD | n= 39 | HD | n= 121 | | |
| Others | n= 613 | Others | n= 1,017 | Others | n= 1,107 | | |

% genes downregulated by A-485

RAD21 reg. class — 1hr n= 10,059
2hr n= 10,055
4hr n= 10,092
A-485 reg. class — n= 10,103

**c**

RAD21down (4hr) — 291
RAD21down (2hr) — 60
325
141
16
16   20
RAD21down (1hr)

**Extended Data Fig. 3 | RAD21^AID depletion in NPC causes downregulation of hundreds of genes. a,** Transcription changes after RAD21^AID depletion in NPC. Transcription changes were quantified in NPC using EU-seq. Transcription changes are compared in NPC without or with depletion RAD21^AID by treatment with IAA (500uM, 1-4 h), and after A-485 treatment (1 h). Regulated genes were categorized into the denoted class based on the indicated fold-change in gene expression. The data are from 2 biological replicates. Dotted lines indicate fold-change of 1.3, 1.5, and 2 (up-regulation and down-regulation). **b,** Number of genes regulated in NPC after 1, 2, and 4 hours of RAD21^AID depletion and 1 hour of A-485 treatment (left panels). Fraction of genes regulated in NPC after 1, 2, and 4 hours of RAD21^AID depletion and 1 hour of A-485 treatment (right panels). **c,** Overlap between genes regulated after 1, 2, and 4 hours of RAD21^AID depletion in NPC. Number of regulated genes is shown.

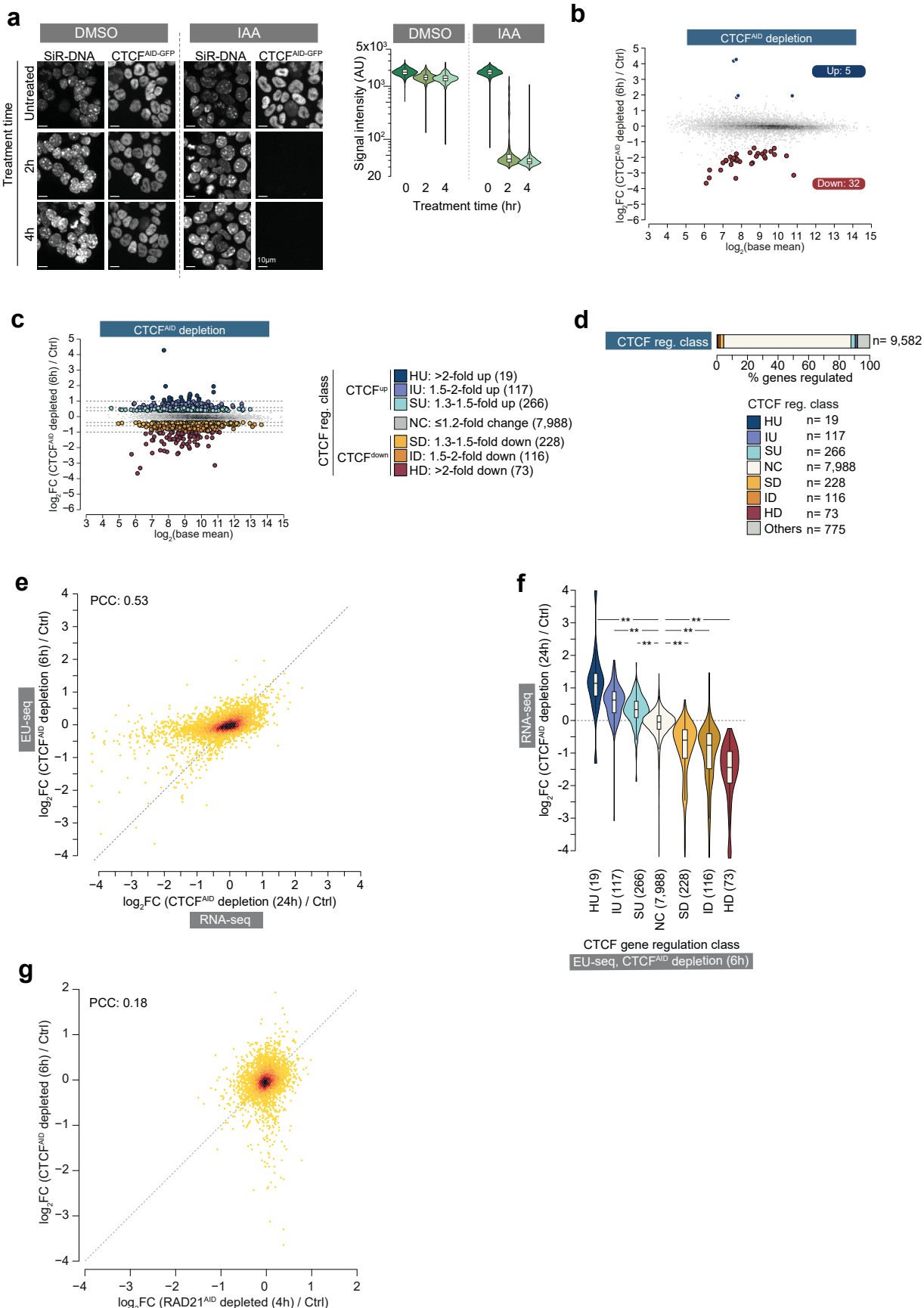

**Extended Data Fig. 4 | See next page for caption.**

**Extended Data Fig. 4 | Many CTCF targets are consistently regulated after short-term and long-term CTCF^AID depletion. a**, Confirmation of the auxin-induced depletion of CTCF^AID–GFP (hereafter CTCF^AID) by microscopy-based imaging (n = 1). Cells were exposed to the indicated treatments for the specified time, and CTCF^AID expression was quantified by analyzing GFP expression, and nuclei were visualized with SiR-DNA. Scale bar indicates 10 μm. The violin plots display the mean GFP intensity distribution in each condition (n = 1,000 cells/ condition). The box plots show upper and lower quartiles, the line indicates the median, and the whiskers show the 1.5× interquartile range. **b**, The number of significantly (P$_{adj}$ <0.05) up- and down-regulated genes after acute CTCF^AID depletion (6 h). Transcription changes were quantified in mESCs expressing CTCF^AID using EU-seq (n = 2). Transcription changes are compared in cells without or with depletion of CTCF^AID by treatment with IAA (500uM, 6 h). The analysis includes all genes expressed at TPM > 5. **c**, Fold-change-based classification of genes regulated after acute (6 h) CTCF^AID depletion. Regulated genes are classified using the indicated fold-change in gene expression after CTCF^AID depletion. The analysis only includes genes that are expressed at TPM > 15. Dotted lines indicate fold-change of 1.3, 1.5, and 2 (up- and down-regulations). **d**, Fraction of genes regulated after CTCF^AID depletion (top panel), and the number of genes in the indicated CTCF gene regulation class (bottom panel). Gene regulation class is defined based on fold-change in gene expression after

CTCF^AID depletion. **e**, Correlation between gene regulation after acute (6 h) and long-term (24 h) depletion of CTCF^AID. Transcription changes were quantified by EU-seq after acute CTCF^AID depletion (this study), and by RNA-seq after long-term CTCF^AID depletion. RNA-seq data are from ref.[49]. Pearson correlation coefficient (PCC) is indicated. **f**, Changes in mRNA expression in the indicated classes of CTCF-regulated genes after long-term (24 h) CTCF^AID depletion. CTCT gene regulation class is defined based on nascent transcription changes after acute CTCF^AID depletion (6 h), as specified in Extended Data Fig. 4c-d. Within each class, the change in mRNA expression after long-term CTCF^AID depletion (24 h, RNA-seq) is shown. RNA-seq data are from ref.[49]. The box plots display the median, upper and lower quartiles; the whiskers show the 1.5× interquartile range. Two-sided Mann–Whitney U-test, followed by correction for multiple comparisons with the Benjamini–Hochberg method; **P$_{adj}$ < 0.001. P$_{adj}$: HU vs NC, 7.1 ×10⁻¹¹; IU vs NC, 2.3 ×10⁻⁴⁰; SU vs NC, 2.8 ×10⁻⁵²; SD vs NC, 9.2 ×10⁻⁶²; ID vs NC, 1.5 ×10⁻⁴⁶; HD vs NC, 8.4 ×10⁻⁴⁴. Numbers of genes (n) per class are shown in brackets. Dotted lines indicate log2 fold-change of 0. **g**, Correlation between nascent transcription changes quantified after RAD21^AID depletion (4 h) and CTCF^AID depletion (6 h). RAD21^AID depletion-induced fold changes are determined from two biological replicates. PCC: Pearson´s correlation coefficient. The dotted line indicates an identical line (y = x).

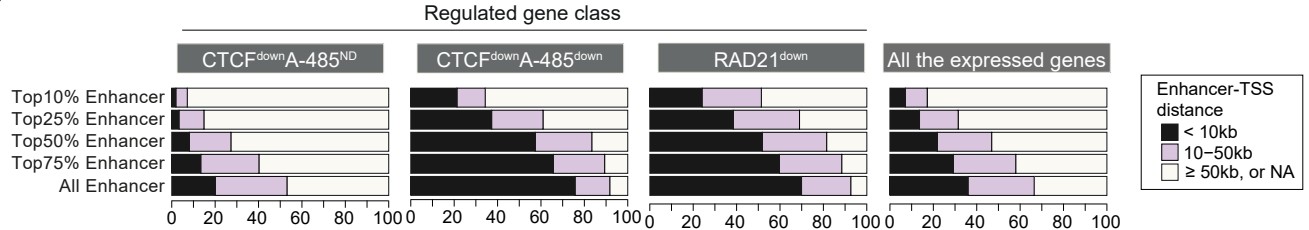

**Extended Data Fig. 5 | CTCF and RAD21 regulated genes differ in their proximity to candidate enhancers. a**, Candidate enhancer enrichment in proximity to the indicated classes of genes regulated after RAD21$^{AID}$ and CTCF$^{AID}$ depletion and randomly selected genes in mESC. CTCF downregulated genes are grouped into two categories: CTCF$^{down}$A-485$^{down}$ and CTCF$^{down}$A-485$^{ND}$. Candidate enhancers are defined by H3K27ac and H2BK20ac overlapping peaks,

excluding region +/-500bp from the TSS, and enhancer strength is determined by H2BK20ac ChIP signal enrichment. **b**, Quantification of enhancer enrichment near RAD21- and CTCF-regulated genes shown in panel **a**. Enhancer enrichment near all expressed genes is used as a reference to assess the relative prevalence of candidate enhancers in CTCF and RAD21-regulated genes.

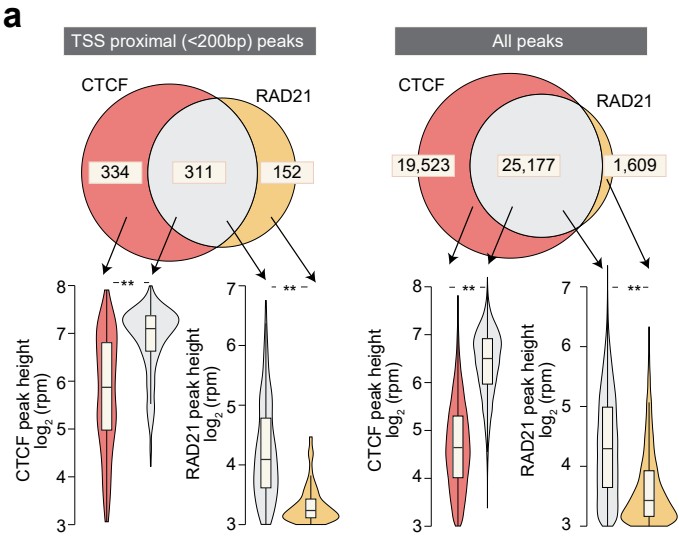

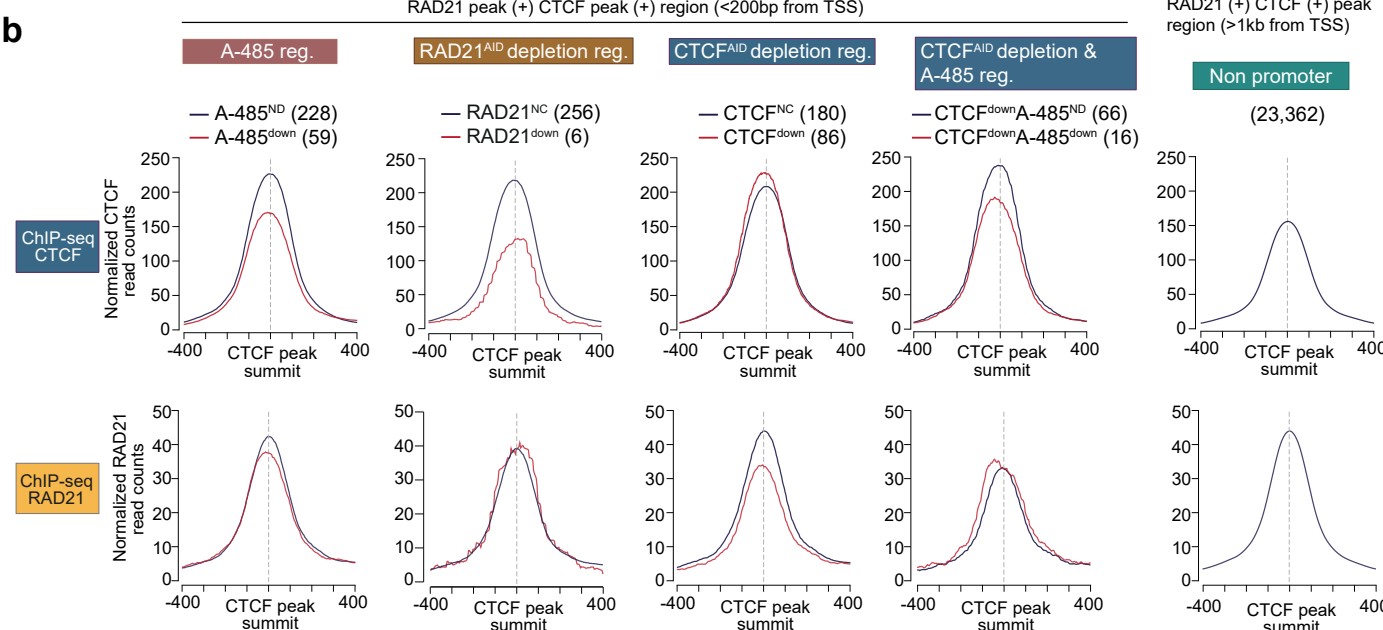

**Extended Data Fig. 6 | Promoters of CTCF^down genes show strong enrichment of CTCF but weak enrichment of RAD21. a**, The top panels show overlap between CTCF and RAD21 ChIP-seq peaks detected in TSS proximal (+/-200bp from TSS) regions and all regions. If RAD21 peak overlapped with more than one CTCF peaks or vice versa, the higher number of overlapping peaks is shown. The bottom panels show comparative enrichment of CTCF and RAD21 in TSS-proximal and distal peaks. The box plots display the median, upper and lower quartiles; the whiskers show the 1.5× interquartile range. Two-sided Mann–Whitney U-test, followed by correction for multiple comparisons with the Benjamini–Hochberg method; **$P_{adj}$ < 0.001. $P_{adj}$: (TSS proximal) CTCF$^{ov\_RAD21}$ vs CTCF$^{non-ov\_RAD21}$, < 1.1 ×10$^{-37}$; RAD21$^{ov\_CTCF}$ vs RAD21$^{non-ov\_CTCF}$,

4.5×10$^{-28}$; (All peaks) CTCF$^{ov\_RAD21}$ vs CTCF$^{non-ov\_RAD21}$, < 1 ×10$^{-200}$; RAD21$^{ov\_CTCF}$ vs RAD21$^{non-ov\_CTCF}$, < 1 ×10$^{-200}$. Of note, CTCF peak height is similar, or even higher in TSS proximal peaks than in distal peaks, but RAD21 shows lower enrichment in TSS proximal regions as compared to distal regions. **b**, Aggregate plots showing CTCF and RAD21 enrichment in the indicated groups of promoters and non-promoter regions. This analysis only includes promoters that are bound (within +/- 200 bp of TSS) by both CTCF and RAD21 in the analyzed ChIP-seq data. Promoters are grouped based on the regulation of corresponding genes after the specified treatments. Peaks are centered using the CTCF peak summit. The dotted line indicates the CTCF peak summit position. Non-promoter peaks are defined as those present >1 kb away from TSS.

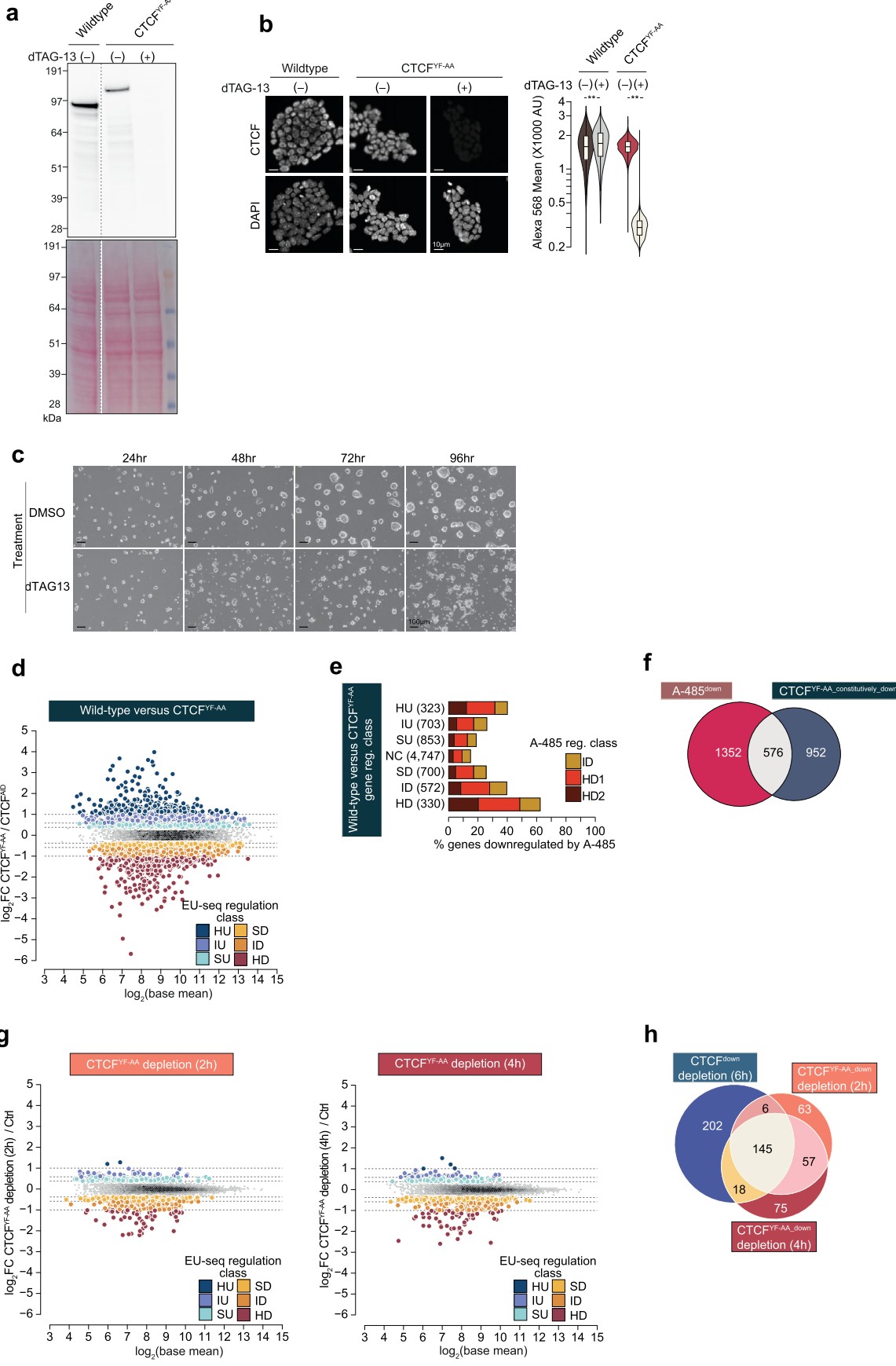

**Extended Data Fig. 7 | See next page for caption.**

**Extended Data Fig. 7 | Acute depletion of CTCF[YF-AA] impairs cell proliferation and downregulates a large portion of genes downregulated by wild-type CTCF[AID] depletion. a-b**, Confirmation of dTAG-13-induced depletion of CTCF[YF-AA] by immunoblotting (**a**) and immunofluorescence (**b**). Ponceau staining serves as a protein loading control. Scale bar indicates 10 μm. CTCF expression in immunofluorescence is quantified by anti-CTCF antibody, and nuclei are stained using DAPI (n = 1). The box plots display the mean CTCF fluorescence intensity distribution in each condition, showing the median, upper and lower quartiles; the whiskers show the 1.5× interquartile range. Two-sided Mann–Whitney U-test, followed by correction for multiple comparisons with the Benjamini–Hochberg method; **$P_{adj}$ < 0.001. $P_{adj}$: (Wildtype) dTAG-13 (+) vs dTAG13 (-), 8.9 ×10$^{-22}$; (CTCF[YF-AA]) dTAG-13 (+) vs dTAG13 (-), <1 ×10$^{-200}$. **c**, Micrographs showing colony growth of CTCF[YF-AA] mESCs, with or without depletion of CTCF[YF-AA]. Cells were imaged at the indicated time points after starting the dTAG-13 or DMSO treatments (n = 2 replicates). Scale bar indicates 100 μm. **d**, Gene expression changes in CTCF[YF-AA] cells. Gene expression is compared between CTCF[YF-AA] and CTCF[AID] cells cultured under standard conditions without any treatments. HU: >2-fold upregulated, IU: 1.5-2-fold upregulated, SU: 1.3-1.5-fold upregulated, NC: <1.2-fold change, SD: 1.3-1.5-fold downregulated, ID: 1.5-2-fold downregulated, HD: >2-fold downregulated. Dotted lines indicate fold-change of 1.3, 1.5, and 2 (up-regulation and down-regulations). **e**, Shown is the fraction of A-485-regulated genes among the genes dysregulated in cells constitutively expressing CTCF[YF-AA]. Classification of regulated genes in constitutively expressing CTCF[YF-AA] cells is defined in panel **d**. **f**, Overlap between A-485 downregulated genes and genes downregulated in constitutively CTCF[YF-AA] expressing cells. **g**, Change in gene expression after depletion of CTCF[YF-AA] for 2 h (left panel) or 4 h (right panel). Based on fold-change in gene expression, regulated genes are grouped into the indicated categories. Dotted lines indicate fold-change of 1.3, 1.5, and 2 (up-regulation and down-regulation). **h**, Overlap between genes downregulated after the acute depletion of CTCF[AID] (6 h), and after CTCF[YF-AA] depletion (2 h, or 4 h).

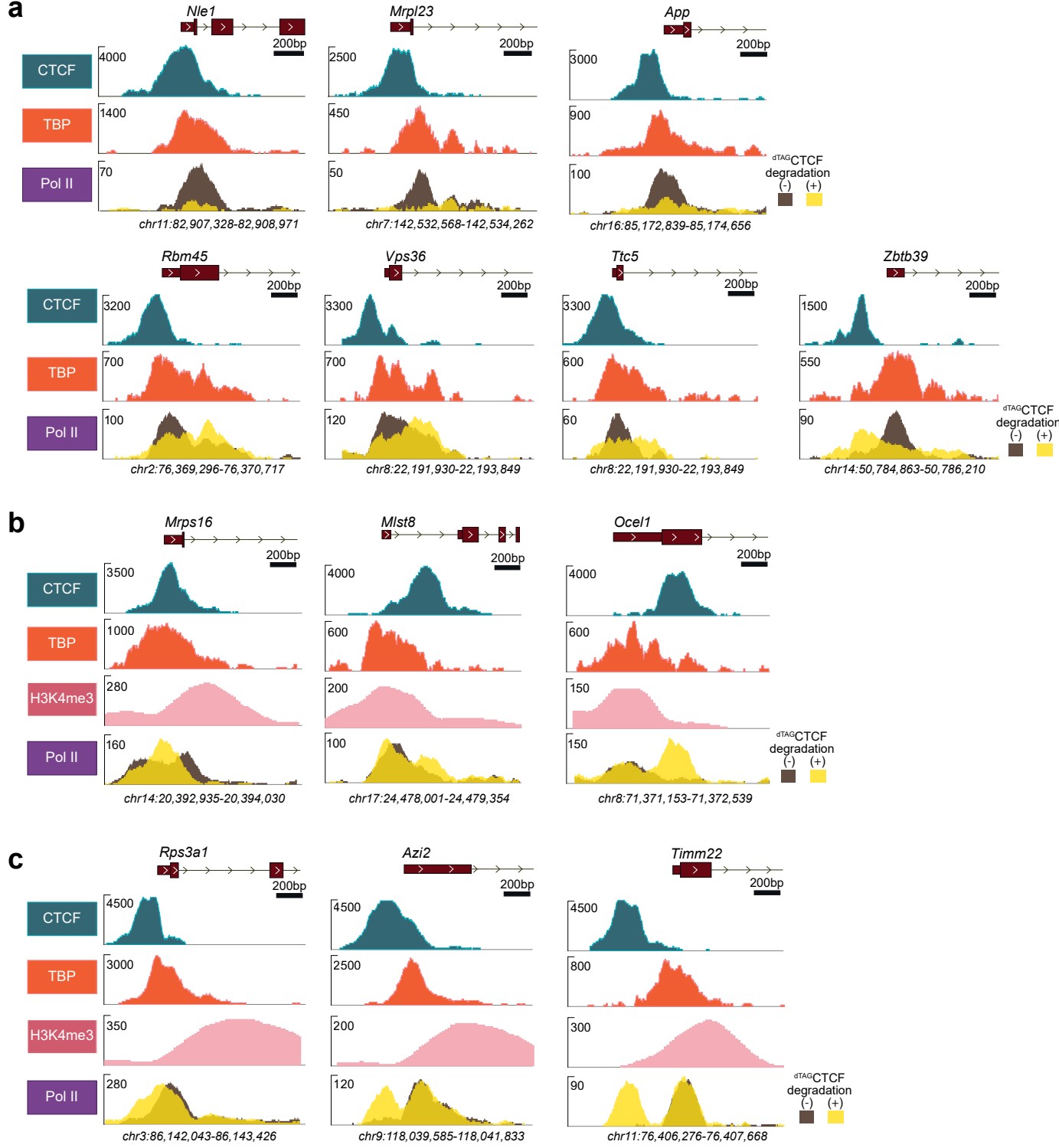

**Extended Data Fig. 8 | CTCF promoter binding affects Pol II recruitment in a position- and orientation-specific manner. a-c,** Shown are genome browser tracks of representative CTCF[down] genes (**a**), CTCF[up] genes (**b**), and CTCF[uasTrx_up] genes (**c**) after acute depletion of CTCF. Pol II enrichment is analyzed with or without depletion (4 h) of [dTAG]CTCF. ChIP-seq profiles of CTCF, TBP, and where specified H3K4me3, are included as references. Among the shown genes, *App* and *Ocel1* are expressed at a low level (EU-seq TPM < 15), and therefore, were not included in global analyses presented throughout the manuscript.

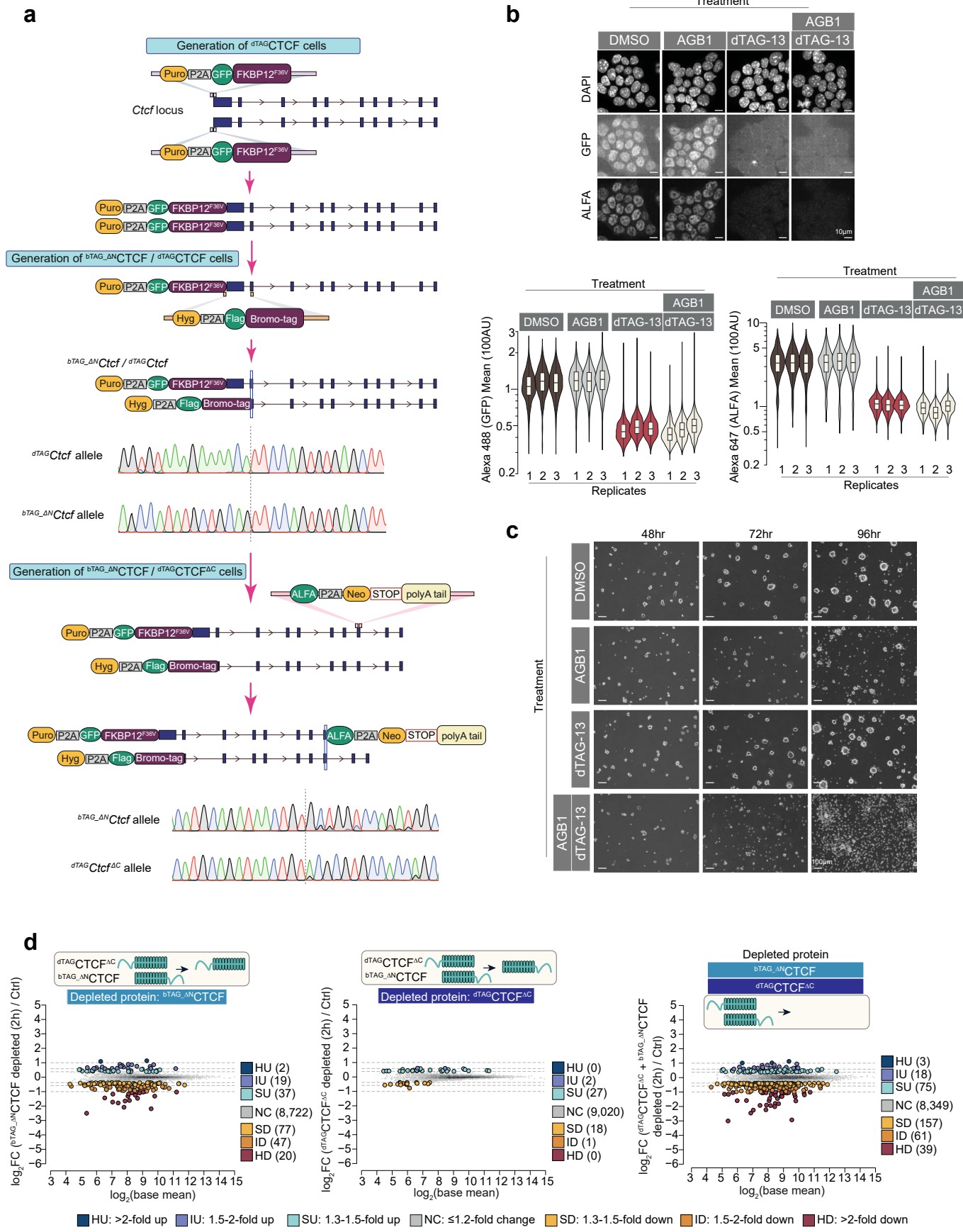

**Extended Data Fig. 9 | See next page for caption.**

**Extended Data Fig. 9 | Generation and characterization of** $^{bTAG\_\Delta N}$**CTCF /** $^{dTAG}$**CTCF$^{\Delta C}$ expressing mESC. a**, Schematic of the stepwise genome editing strategy for creating $^{bTAG\_\Delta N}$CTCF / $^{dTAG}$CTCF$^{\Delta C}$ expressing cells. First, both *Ctcf* alleles were fused with GFP-dTAG using CRISPR and the indicated targeting construct. Next, one allele was edited to fuse BromoTag (bTAG) and delete the N-terminal coding region (amino acids 1–265). Finally, the GFP-dTAG allele was edited to truncate the CTCF C-terminal region (residues 578–736) and insert an ALFA-tag, generating the $^{bTAG\_\Delta N}$CTCF / $^{dTAG}$CTCF$^{\Delta C}$ cell line. **b**, Validation of ALFA-tag and GFP co-expression and their selective depletion by dTAG-13 (n = 3). Cells were treated as indicated and imaged for the expression of GFP and ALFA-tag (top panels). Nuclei were visualized with SiR-DNA. Scale bar indicates

10 μm. The violin plots show GFP (left, bottom panels) and ALFA (right, bottom panels) intensity across specified conditions (n = 1,000 cells per condition). The box plots show upper and lower quartiles, the line indicates the median, and the whiskers show the 1.5× interquartile range. **c**, Micrographs showing colony growth of $^{bTAG\_\Delta N}$CTCF / $^{dTAG}$CTCF$^{\Delta C}$ expressing cells under the indicated treatments. Images were taken at the specified time points (n = 2 replicates). Scale bar indicates 100 μm. **d**, Fold-change in nascent transcription after the depletion of $^{bTAG\_\Delta N}$CTCF, $^{dTAG}$CTCF$^{\Delta C}$, or both (n = 2 replicates). Up- and down-regulated genes are grouped into the indicated gene regulation classes. The analysis only includes genes that are expressed at TPM > 15. Dotted lines indicate fold-change of 1.3, 1.5, and 2 (up-regulation and down-regualtion).

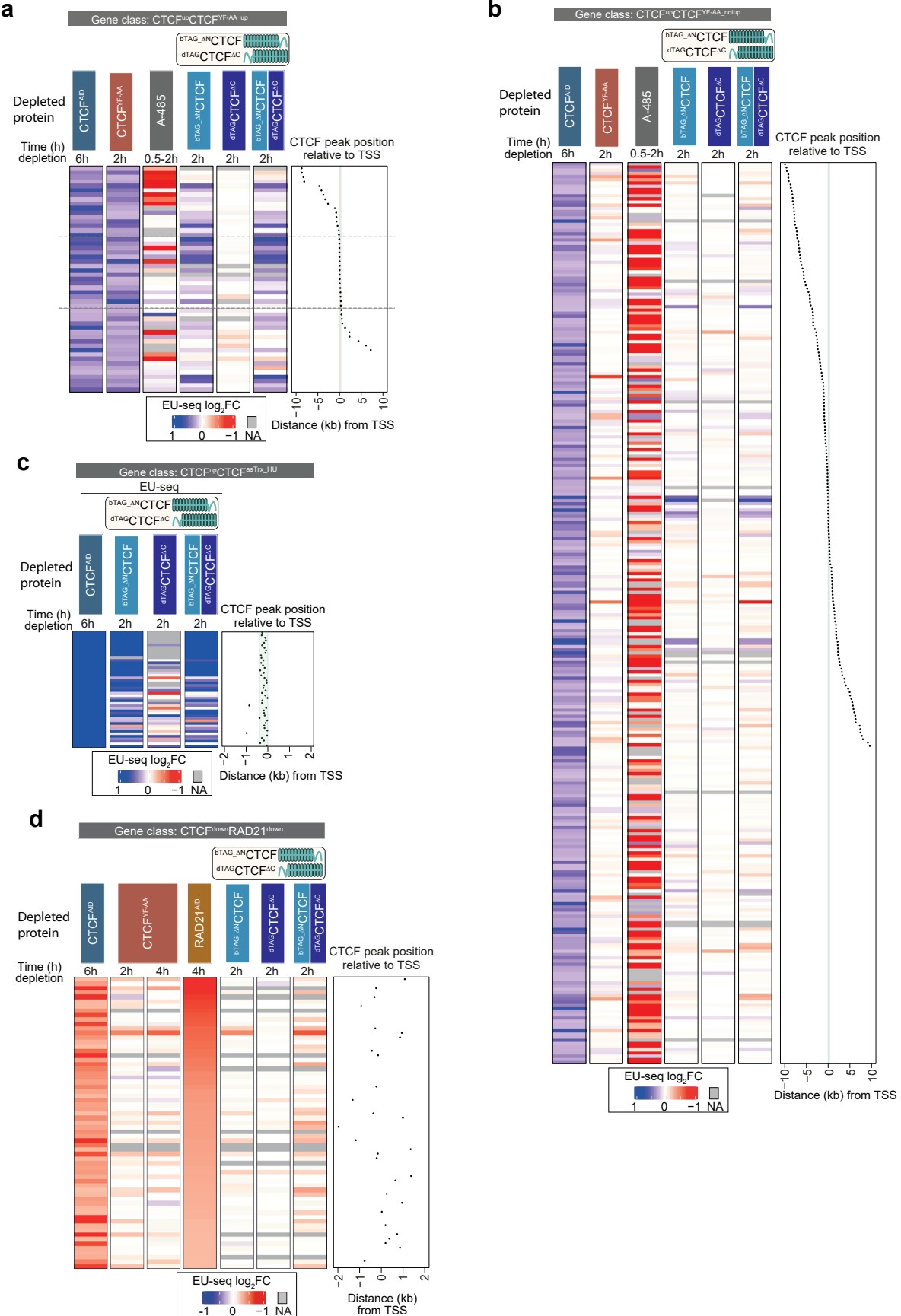

**Extended Data Fig. 10 | See next page for caption.**

**Extended Data Fig. 10 | The CTCF C-terminal region is required for repressing sense and anti-sense transcription. a**, Fold-change in expression of genes commonly upregulated ( > 1.3-fold) after CTCF$^{AID}$ and CTCF$^{YF-AA}$ depletion, and their regulation following depletion of $^{bTAG\_\Delta N}$CTCF, $^{dTAG}$CTCF$^{\Delta C}$, or both. Regions from TSS to +400 bp are highlighted. Dotted lines mark genes with a CTCF peak position located between TSS to +400 bp. **b**, Fold-change in genes specifically upregulated by CTCF$^{AID}$ but not by CTCF$^{YF-AA}$, and their regulation following depletion of $^{bTAG\_\Delta N}$CTCF, $^{dTAG}$CTCF$^{\Delta C}$, or both. Regions from -400bp to TSS are highlighted. **c**, Fold-change in expression of upstream antisense transcripts upregulated ( > 2-fold) after CTCF$^{AID}$ depletion, and their regulation following depletion of $^{bTAG\_\Delta N}$CTCF, $^{dTAG}$CTCF$^{\Delta C}$, or both. Regions from TSS to +400 bp are highlighted. **d**, Fold-change in genes commonly downregulated ( > 1.3-fold) by depletion of CTCF$^{AID}$ and RAD21$^{AID}$, and their regulation after depletion of $^{bTAG\_\Delta N}$CTCF, $^{dTAG}$CTCF$^{\Delta C}$, or their combination.

# Reporting Summary

## Statistics

For all statistical analyses, confirm that the following items are present in the figure legend, table legend, main text, or Methods section.

| n/a | Confirmed | |
|---|---|---|
| ☐ | ☒ | The exact sample size (*n*) for each experimental group/condition, given as a discrete number and unit of measurement |
| ☐ | ☒ | A statement on whether measurements were taken from distinct samples or whether the same sample was measured repeatedly |
| ☐ | ☒ | The statistical test(s) used AND whether they are one- or two-sided *Only common tests should be described solely by name; describe more complex techniques in the Methods section.* |
| ☐ | ☒ | A description of all covariates tested |
| ☐ | ☒ | A description of any assumptions or corrections, such as tests of normality and adjustment for multiple comparisons |
| ☐ | ☒ | A full description of the statistical parameters including central tendency (e.g. means) or other basic estimates (e.g. regression coefficient) AND variation (e.g. standard deviation) or associated estimates of uncertainty (e.g. confidence intervals) |
| ☐ | ☒ | For null hypothesis testing, the test statistic (e.g. *F*, *t*, *r*) with confidence intervals, effect sizes, degrees of freedom and *P* value noted *Give P values as exact values whenever suitable.* |
| ☒ | ☐ | For Bayesian analysis, information on the choice of priors and Markov chain Monte Carlo settings |
| ☒ | ☐ | For hierarchical and complex designs, identification of the appropriate level for tests and full reporting of outcomes |
| ☐ | ☒ | Estimates of effect sizes (e.g. Cohen's *d*, Pearson's *r*), indicating how they were calculated |

*Our web collection on statistics for biologists contains articles on many of the points above.*

## Software and code

Policy information about availability of computer code

| Data collection | No software is used for data collection. |
|---|---|
| Data analysis | Cutadapt, fastqc(0.12.1), STAR(2.6.1a), Bedtools(2.23), HTseq(0.11.1), bwa meme (version 1.0.4),  samtools(1.4), Lanceotron(20210215), R(4.1.1), ggplot2(3.3.5), DESeq(1.32.0), bigWigMerge,  deeptools(3.5.2), IGV(2.16), Picard-tools(2.9.1), epic2(0.0.47), GEM(3.4), STREME(5.5.4), FIMO (5.5.4), higlass-python(v1.2.0) |

For manuscripts utilizing custom algorithms or software that are central to the research but not yet described in published literature, software must be made available to editors and reviewers. We strongly encourage code deposition in a community repository (e.g. GitHub). See the Nature Portfolio guidelines for submitting code & software for further information.

## Data

Policy information about availability of data

All manuscripts must include a data availability statement. This statement should provide the following information, where applicable:

- Accession codes, unique identifiers, or web links for publicly available datasets
- A description of any restrictions on data availability
- For clinical datasets or third party data, please ensure that the statement adheres to our policy

Raw EU-seq, ChIP-seq and ATAC-seq data generated in this study have been deposited in the NCBI Gene Expression Omnibus (GEO) under the SuperSeries accession GSE262521 (https://www.ncbi.nlm.nih.gov/geo/query/acc.cgi?acc=GSE262521). Processed read counts, peak region sets, and genome-browser tracks are provided

## Research involving human participants, their data, or biological material

Policy information about studies with human participants or human data. See also policy information about sex, gender (identity/presentation), and sexual orientation and race, ethnicity and racism.

| | |
|---|---|
| Reporting on sex and gender | Not relevant. |
| Reporting on race, ethnicity, or other socially relevant groupings | Not relevant. |
| Population characteristics | Not relevant. |
| Recruitment | Not relevant. |
| Ethics oversight | Not relevant. |

Note that full information on the approval of the study protocol must also be provided in the manuscript.

# Field-specific reporting

Please select the one below that is the best fit for your research. If you are not sure, read the appropriate sections before making your selection.

☒ Life sciences          ☐ Behavioural & social sciences          ☐ Ecological, evolutionary & environmental sciences

For a reference copy of the document with all sections, see [nature.com/documents/nr-reporting-summary-flat.pdf](http://nature.com/documents/nr-reporting-summary-flat.pdf)

# Life sciences study design

All studies must disclose on these points even when the disclosure is negative.

| | |
|---|---|
| Sample size | No statistical methods were used to pre-determine sample sizes but our sample sizes are similar to those reported in previous publications (PMID 35410381, PMID 36471071, PMID: 34002095). |
| Data exclusions | No data were excluded. |
| Replication | All attempts to replicate the results were successful.  ATAC-seq and ChIP-seq were each performed in two biological replicates. For analyses of expression changes following CTCF depletion, both stranded and non-stranded EU-seq were conducted with two biological replicates. |

Expression changes after RAD21 depletion or CTCF YF-AA mutant were also assessed in two biological replicates. To evaluate the robustness of EU-seq measurements for RAD21 depletion, we analyzed an additional 12 biological replicates. Transcriptome changes after bTAG-ΔN CTCF and/or dTAG CTCF-ΔC depletion were assessed using two biological replicates. Protein depletion of RAD21 and CTCF was confirmed by western blotting and/or quantitative microscopy with the following designs: RAD21 depletion—one biological replicate at two time points; CTCF depletion—one biological replicate at two time points; CTCF YF-AA mutant—one biological replicate at one time point; bTAG-ΔN CTCF and/or dTAG CTCF-ΔC—three biological replicates at one time point.

| | |
|---|---|
| Randomization | The study did not involve animal or human participants. Random allocation did not apply because samples were not subjected to co- or multivariate analysis. |
| Blinding | Control and treatment cells were cultured in parallel under identical conditions, and all procedures other than the intended treatment were performed identically. Library preparation and sequencing followed the same protocols for all samples as described in the method section. |

# Reporting for specific materials, systems and methods

We require information from authors about some types of materials, experimental systems and methods used in many studies. Here, indicate whether each material, system or method listed is relevant to your study. If you are not sure if a list item applies to your research, read the appropriate section before selecting a response.

## Materials & experimental systems

| n/a | Involved in the study |
|---|---|
| ☐ | ☒ Antibodies |
| ☐ | ☒ Eukaryotic cell lines |
| ☒ | ☐ Palaeontology and archaeology |
| ☒ | ☐ Animals and other organisms |
| ☒ | ☐ Clinical data |
| ☒ | ☐ Dual use research of concern |
| ☒ | ☐ Plants |

## Methods

| n/a | Involved in the study |
|---|---|
| ☐ | ☒ ChIP-seq |
| ☒ | ☐ Flow cytometry |
| ☒ | ☐ MRI-based neuroimaging |

## Antibodies

| | |
|---|---|
| Antibodies used | ChIP-seq<br>Rpb1 NTD (D8L4Y) Rabbit mAb #14958, Cell Signaling Technology, 5ug<br><br>Western blotting<br>CTCF (D31H2) XP Rabbit mAb #3418, Cell Signaling Technology, 1:1000<br>Histone H3 (D1H2) XP Rabbit mAb #4499, Cell Signaling Technology<br>Anti-GAPDH Antibody #ABS16, Sigma-Aldrich<br><br>Immunostaining<br>CTCF (D31H2) XP Rabbit mAb #3418, Cell Signaling Technology, 1:1000<br>ALFA Recombinant anti-ALFA Antibody #N1581, NanoTag Biotechnologies, 1:500<br>Goat anti-Mouse IgG (H+L) Cross-Adsorbed Secondary Antibody, Alexa Flur 647, A-21235, Invitrogen |
| Validation | The following quality assurance were provided on the manufacture's websites.<br>Anti-Rpb1 NTD ab was tested by ChIP-seq in HeLa cells. According to the provider, Rpb1 NTD (D8L4Y) Rabbit mAb recognizes endogenous levels of total Rpb1 protein at the amino terminal domain (NTD). (https://www.cellsignal.com/products/primary-antibodies/rpb1-ntd-d8l4y-rabbit-mab/14958)<br><br>Anti-CTCF ab was tested by western blot in HeLa, NIH3T3, C6, and COS cells, and also by immunofluorescent analysis of HCT-116 cells. Acoording to the provider, CTCF (D31H2) XP® Rabbit mAb detects endogenous levels of total CTCF protein. This antibody does not cross-react with BORIS. This antibody nonspecifically labels the lamina propria of small intestine in fixed frozen mouse tissue by immunofluorescence. (https://www.cellsignal.com/products/primary-antibodies/ctcf-d31h2-xp-rabbit-mab/3418)<br><br>Anti-H3 ab was tested by western blot in HeLa, NIH3T3, C6, and COS cells. According to the provider, Histone H3 (D1H2) XP® Rabbit mAb detects endogenous levels of total Histone H3 protein, including isoforms H3.1, H3.2, and H3.3. This antibody also detects the Histone H3 variant CENP-A. This antibody does not cross-react with other core histones. (https://www.cellsignal.com/products/primary-antibodies/histone-h3-d1h2-xp-rabbit-mab/4499)<br><br>Anti-GAPDH antibody was evaluated by Western blot in HEK293 cell lysates. According to the provider, This antibody is supported by peer reviewed publications and reliably detects Glyceraldehyde-3-Phosphate Dehydrogenase (GAPDH) is validated for use in WB. (https://www.sigmaaldrich.com/DK/en/product/mm/abs16) |

## Eukaryotic cell lines

Policy information about cell lines and Sex and Gender in Research

| | |
|---|---|
| Cell line source(s) | Mouse embryonic stem cells (mESC; E14TG2a; Sigma-Aldrich, Cat# 08021401) were purchased from Sigma-Aldrich. RAD21- |

| Cell line source(s) | AID mESCs were provided by Dr Rob Klose. Ctcf-mAID-GFP mESCs were provided by Drs Elphège P. Nora and Benoit G. Bruneau. |
|---|---|
| Authentication | The identity of ESC and NSC were authenticated by the expression of cell-type-specific markers by EU-seq. |
| Mycoplasma contamination | Cell lines were tested for mycoplasma contamination every 2-3 months, and were confirmed mycoplasma negative. |
| Commonly misidentified lines (See ICLAC register) | None of the used cell lines are listed in the commonly misidentified lines. |

## Plants

| Seed stocks | Not relevant. |
|---|---|
| Novel plant genotypes | Not relevant. |
| Authentication | Not relevant. |

## ChIP-seq

### Data deposition

☒ Confirm that both raw and final processed data have been deposited in a public database such as GEO.

☒ Confirm that you have deposited or provided access to graph files (e.g. BED files) for the called peaks.

| Data access links *May remain private before publication.* | https://www.ncbi.nlm.nih.gov/geo/query/acc.cgi?acc=GSE262521 |
|---|---|
| Files in database submission | ATAC-seq<br>ATACSeq.dTAG.CTCF.ESC_Ctrl_TC0_CC30_1_R1.fastq.gz<br>ATACSeq.dTAG.CTCF.ESC_Ctrl_TC0_CC30_1_R2.fastq.gz<br>ATACSeq.dTAG.CTCF.ESC_Ctrl_TC0_CC30_2_R1.fastq.gz<br>ATACSeq.dTAG.CTCF.ESC_Ctrl_TC0_CC30_2_R2.fastq.gz<br>ATACSeq.dTAG.CTCF.ESC_dTAG13_100nM_TC180_CC30_1_R1.fastq.gz<br>ATACSeq.dTAG.CTCF.ESC_dTAG13_100nM_TC180_CC30_1_R2.fastq.gz<br>ATACSeq.dTAG.CTCF.ESC_dTAG13_100nM_TC180_CC30_2_R1.fastq.gz<br>ATACSeq.dTAG.CTCF.ESC_dTAG13_100nM_TC180_CC30_2_R2.fastq.gz<br><br>ChIP-seq<br>GFP.dTAG.CTCF.ESC_Ctrl_TC0_Pol2.D8L4Y_CC23_1.fastq.gz<br>GFP.dTAG.CTCF.ESC_Ctrl_TC0_Pol2.D8L4Y_CC25_1.fastq.gz<br>GFP.dTAG.CTCF.ESC_dtag13_100nM_TC240_Pol2.D8L4Y_CC23_1.fastq.gz<br>GFP.dTAG.CTCF.ESC_dtag13_100nM_TC240_Pol2.D8L4Y_CC25_1.fastq.gz<br><br>EU-seq<br>EUSeq.GFP.dTAG.CTCF.Y226A.F228A.ESC_Ctrl_TC0_spike.dm_CC37_1.fastq.gz<br>EUSeq.GFP.dTAG.CTCF.Y226A.F228A.ESC_Ctrl_TC0_spike.dm_CC37_2.fastq.gz<br>EUSeq.GFP.dTAG.CTCF.Y226A.F228A.ESC_dTAG13_200nM_TC120_spike.dm_CC37_1.fastq.gz<br>EUSeq.GFP.dTAG.CTCF.Y226A.F228A.ESC_dTAG13_200nM_TC120_spike.dm_CC37_2.fastq.gz<br>EUSeq.GFP.dTAG.CTCF.Y226A.F228A.ESC_dTAG13_200nM_TC240_spike.dm_CC37_1.fastq.gz<br>EUSeq.GFP.dTAG.CTCF.Y226A.F228A.ESC_dTAG13_200nM_TC240_spike.dm_CC37_2.fastq.gz<br><br>EUSeq.GFP.AID.CTCF.ESC_Ctrl_TC0_EU.CC5_1.fastq.gz<br>EUSeq.GFP.AID.CTCF.ESC_Ctrl_TC0_EU.CC5_2.fastq.gz<br>EUSeq.GFP.AID.CTCF.ESC_Ctrl_TC0_strand_CC26_1.fastq.gz<br>EUSeq.GFP.AID.CTCF.ESC_Ctrl_TC0_strand_CC29_1.fastq.gz<br>EUSeq.GFP.AID.CTCF.ESC_IAA_500uM_TC360_EU.CC5_1.fastq.gz<br>EUSeq.GFP.AID.CTCF.ESC_IAA_500uM_TC360_EU.CC5_2.fastq.gz<br>EUSeq.GFP.AID.CTCF.ESC_IAA_500uM_TC360_strand_CC26_1.fastq.gz<br>EUSeq.GFP.AID.CTCF.ESC_IAA_500uM_TC360_strand_CC29_1.fastq.gz<br><br>EUSeq.GFP.AID.Rad21.ESC_Ctrl_TC0_CC24_1.fastq.gz<br>EUSeq.GFP.AID.Rad21.ESC_Ctrl_TC0_CC24_2.fastq.gz<br>EUSeq.GFP.AID.Rad21.ESC_IAA_500uM_TC240_CC24_1.fastq.gz<br>EUSeq.GFP.AID.Rad21.ESC_IAA_500uM_TC240_CC24_2.fastq.gz |

EUSeq.GFP.AID.Rad21.NSC_Ctrl_TC0_CC56_1.fastq.gz
EUSeq.GFP.AID.Rad21.NSC_Ctrl_TC0_CC56_2.fastq.gz
EUSeq.GFP.AID.Rad21.NSC_IAA_500uM_TC60_CC56_1.fastq.gz
EUSeq.GFP.AID.Rad21.NSC_IAA_500uM_TC60_CC56_2.fastq.gz
EUSeq.GFP.AID.Rad21.NSC_IAA_500uM_TC120_CC56_1.fastq.gz
EUSeq.GFP.AID.Rad21.NSC_IAA_500uM_TC120_CC56_2.fastq.gz
EUSeq.GFP.AID.Rad21.NSC_IAA_500uM_TC240_CC56_1.fastq.gz
EUSeq.GFP.AID.Rad21.NSC_IAA_500uM_TC240_CC56_2.fastq.gz
EUSeq.GFP.AID.Rad21.NSC_A485_10uM_TC60_CC56_1.fastq.gz
EUSeq.GFP.AID.Rad21.NSC_A485_10uM_TC60_CC56_2.fastq.gz

EUSeq.GFP.AID.Rad21.ESC_Ctrl_TC0_CC52_1.fastq.gz
EUSeq.GFP.AID.Rad21.ESC_Ctrl_TC0_CC53_1.fastq.gz
EUSeq.GFP.AID.Rad21.ESC_Ctrl_TC0_CC53_2.fastq.gz
EUSeq.GFP.AID.Rad21.ESC_Ctrl_TC0_CC53_3.fastq.gz
EUSeq.GFP.AID.Rad21.ESC_Ctrl_TC0_CC53_4.fastq.gz
EUSeq.GFP.AID.Rad21.ESC_Ctrl_TC0_CC53_5.fastq.gz
EUSeq.GFP.AID.Rad21.ESC_Ctrl_TC0_CC53_6.fastq.gz
EUSeq.GFP.AID.Rad21.ESC_Ctrl_TC0_CC53_7.fastq.gz
EUSeq.GFP.AID.Rad21.ESC_Ctrl_TC0_CC53_8.fastq.gz
EUSeq.GFP.AID.Rad21.ESC_Ctrl_TC0_CC53_9.fastq.gz
EUSeq.GFP.AID.Rad21.ESC_Ctrl_TC0_CC53_10.fastq.gz
EUSeq.GFP.AID.Rad21.ESC_Ctrl_TC0_CC53_11.fastq.gz

EUSeq.GFP.AID.Rad21.ESC_IAA_500uM_TC240_CC52_1.fastq.gz
EUSeq.GFP.AID.Rad21.ESC_IAA_500uM_TC240_CC53_1.fastq.gz
EUSeq.GFP.AID.Rad21.ESC_IAA_500uM_TC240_CC53_2.fastq.gz
EUSeq.GFP.AID.Rad21.ESC_IAA_500uM_TC240_CC53_3.fastq.gz
EUSeq.GFP.AID.Rad21.ESC_IAA_500uM_TC240_CC53_4.fastq.gz
EUSeq.GFP.AID.Rad21.ESC_IAA_500uM_TC240_CC53_5.fastq.gz
EUSeq.GFP.AID.Rad21.ESC_IAA_500uM_TC240_CC53_6.fastq.gz
EUSeq.GFP.AID.Rad21.ESC_IAA_500uM_TC240_CC53_7.fastq.gz
EUSeq.GFP.AID.Rad21.ESC_IAA_500uM_TC240_CC53_8.fastq.gz
EUSeq.GFP.AID.Rad21.ESC_IAA_500uM_TC240_CC53_9.fastq.gz
EUSeq.GFP.AID.Rad21.ESC_IAA_500uM_TC240_CC53_10.fastq.gz
EUSeq.GFP.AID.Rad21.ESC_IAA_500uM_TC240_CC53_11.fastq.gz

EUSeq.GFP.dTAG.1.577aa.CTCF_bTAG.Alfa.266.736aa.CTCF.ESC_Ctrl_TC0_strand_CC71_1.fastq.gz
EUSeq.GFP.dTAG.1.577aa.CTCF_bTAG.Alfa.266.736aa.CTCF.ESC_Ctrl_TC0_strand_CC71_2.fastq.gz
EUSeq.GFP.dTAG.1.577aa.CTCF_bTAG.Alfa.266.736aa.CTCF.ESC_AGB1_100nM_TC120_strand_CC71_1.fastq.gz
EUSeq.GFP.dTAG.1.577aa.CTCF_bTAG.Alfa.266.736aa.CTCF.ESC_AGB1_100nM_TC120_strand_CC71_2.fastq.gz
EUSeq.GFP.dTAG.1.577aa.CTCF_bTAG.Alfa.266.736aa.CTCF.ESC_dTAG13_200nM_TC120_strand_CC71_1.fastq.gz
EUSeq.GFP.dTAG.1.577aa.CTCF_bTAG.Alfa.266.736aa.CTCF.ESC_dTAG13_200nM_TC120_strand_CC71_2.fastq.gz
EUSeq.GFP.dTAG.1.577aa.CTCF_bTAG.Alfa.266.736aa.CTCF.ESC_dTAG13_200nM_AGB1_100nM_TC120_strand_CC71_1.fastq.gz
EUSeq.GFP.dTAG.1.577aa.CTCF_bTAG.Alfa.266.736aa.CTCF.ESC_dTAG13_200nM_AGB1_100nM_TC120_strand_CC71_2.fastq.gz

| Genome browser session (e.g. UCSC) | NA |
| --- | --- |

## Methodology

| Replicates | Two biological replicates were performed for both ATAC-seq and ChIP-seq experiments. For the analysis of expression changes following CTCF depletion, two biological replicates were conducted in both stranded EU-seq and non-stranded EU-seq experiments. For the analysis of expression changes after RAD21 depletion or the introduction of the YF-AA CTCF mutant, two biological replicates were performed. To assess the robustness of EU-seq experiments, expression changes after RAD21 depletion analyzed using 12 additional biological replicates. Transcriptome changes after bTAG_ΔNCtcf and/or dTAGCtcfΔC depletion were performed using two biological replicates. |
| --- | --- |
| Sequencing depth | ATAC-seq<br>ATACSeq.dTAG.CTCF.ESC_Ctrl_TC0_CC30_1_R1.fastq.gz, ATACSeq.dTAG.CTCF.ESC_Ctrl_TC0_CC30_1_R2.fastq.gz, total:95504203, unique:48863451, 41bp, paired-end<br>ATACSeq.dTAG.CTCF.ESC_Ctrl_TC0_CC30_2_R1.fastq.gz, ATACSeq.dTAG.CTCF.ESC_Ctrl_TC0_CC30_2_R2.fastq.gz, total:91042591, unique:45519571, 41bp, paired-end<br>ATACSeq.dTAG.CTCF.ESC_dTAG13_100nM_TC180_CC30_1_R1.fastq.gz, ATACSeq.dTAG.CTCF.ESC_dTAG13_100nM_TC180_CC30_1_R2.fastq.gz, total:91098130, unique:44343561, 41bp, paired-end<br>ATACSeq.dTAG.CTCF.ESC_dTAG13_100nM_TC180_CC30_2_R1.fastq.gz, ATACSeq.dTAG.CTCF.ESC_dTAG13_100nM_TC180_CC30_2_R2.fastq.gz, total:85262149, unique:43015138, 41bp, paired-end<br><br>ChIP-seq<br>GFP.dTAG.CTCF.ESC_Ctrl_TC0_Pol2.D8L4Y_CC23_1.fastq.gz, total:37738410, unique:26519883, 101bp, single-end<br>GFP.dTAG.CTCF.ESC_Ctrl_TC0_Pol2.D8L4Y_CC25_1.fastq.gz, total:34739334, unique:25402293, 101bp, single-end |

GFP.dTAG.CTCF.ESC_dtag13_100nM_TC240_Pol2.D8L4Y_CC23_1.fastq.gz, total:37554806, unique:26866053, 101bp, single-end
GFP.dTAG.CTCF.ESC_dtag13_100nM_TC240_Pol2.D8L4Y_CC25_1.fastq.gz, total:33599053, unique:24715268, 101bp, single-end

EU-seq
EUSeq.GFP.dTAG.CTCF.Y226A.F228A.ESC_Ctrl_TC0_spike.dm_CC37_1.fastq.gz, total:14419123, mapped to the mouse genome (excluding rRNA, tRNA):6738367, 82bp, single-end
EUSeq.GFP.dTAG.CTCF.Y226A.F228A.ESC_Ctrl_TC0_spike.dm_CC37_2.fastq.gz, total:38526836, mapped to the mouse genome (excluding rRNA, tRNA):17389892, 82bp, single-end
EUSeq.GFP.dTAG.CTCF.Y226A.F228A.ESC_dTAG13_200nM_TC120_spike.dm_CC37_1.fastq.gz, total:15651533, mapped to the mouse genome  (excluding rRNA, tRNA):7158643, 82bp, single-end
EUSeq.GFP.dTAG.CTCF.Y226A.F228A.ESC_dTAG13_200nM_TC120_spike.dm_CC37_2.fastq.gz, total:34666193, mapped to the mouse genome  (excluding rRNA, tRNA):15634422, 82bp, single-end
EUSeq.GFP.dTAG.CTCF.Y226A.F228A.ESC_dTAG13_200nM_TC240_spike.dm_CC37_1.fastq.gz, total:26202524, mapped to the mouse genome  (excluding rRNA, tRNA):12041967, 82bp, single-end
EUSeq.GFP.dTAG.CTCF.Y226A.F228A.ESC_dTAG13_200nM_TC240_spike.dm_CC37_2.fastq.gz, total:36482931, mapped to the mouse genome  (excluding rRNA, tRNA):16347731, 82bp, single-end

EUSeq.GFP.AID.CTCF.ESC_Ctrl_TC0_EU.CC5_1.fastq.gz, total:30248062, mapped (excluding rRNA, tRNA):23027586, 75bp, single-end
EUSeq.GFP.AID.CTCF.ESC_Ctrl_TC0_EU.CC5_2.fastq.gz, total:31174220, mapped (excluding rRNA, tRNA):24086800, 75bp, single-end
EUSeq.GFP.AID.CTCF.ESC_Ctrl_TC0_strand_CC26_1.fastq.gz, total:35113121, mapped (excluding rRNA, tRNA):18546823, 101bp, single-end
EUSeq.GFP.AID.CTCF.ESC_Ctrl_TC0_strand_CC29_1.fastq.gz, total:62355181, mapped (excluding rRNA, tRNA):35663422, 101bp, single-end
EUSeq.GFP.AID.CTCF.ESC_IAA_500uM_TC360_EU.CC5_1.fastq.gz, total:32874432, mapped (excluding rRNA, tRNA):24797490, 75bp, single-end
EUSeq.GFP.AID.CTCF.ESC_IAA_500uM_TC360_EU.CC5_2.fastq.gz, total:30707213, mapped (excluding rRNA, tRNA):21445919, 75bp, single-end
EUSeq.GFP.AID.CTCF.ESC_IAA_500uM_TC360_strand_CC26_1.fastq.gz, total:31187672, mapped (excluding rRNA, tRNA):15302948, 101bp, single-end
EUSeq.GFP.AID.CTCF.ESC_IAA_500uM_TC360_strand_CC29_1.fastq.gz, total:57686132, mapped (excluding rRNA, tRNA):31351029, 101bp, single-end

EUSeq.GFP.AID.Rad21.ESC_Ctrl_TC0_CC24_1.fastq.gz, total:27205836, mapped (excluding rRNA, tRNA):15222440, 101bp, single-end
EUSeq.GFP.AID.Rad21.ESC_Ctrl_TC0_CC24_2.fastq.gz, total:29298620, mapped (excluding rRNA, tRNA):17367617, 101bp, single-end
EUSeq.GFP.AID.Rad21.ESC_IAA_500uM_TC240_CC24_1.fastq.gz, total:28119926, mapped (excluding rRNA, tRNA):15702675, 101bp, single-end
EUSeq.GFP.AID.Rad21.ESC_IAA_500uM_TC240_CC24_2.fastq.gz, total:32429251, mapped (excluding rRNA, tRNA):17425766, 101bp, single-end

EUSeq.GFP.AID.Rad21.NSC_Ctrl_TC0_CC56_1.fastq.gz, total: 48987777, mapped (excluding rRNA, tRNA):25478206, 132bp, single-end
EUSeq.GFP.AID.Rad21.NSC_Ctrl_TC0_CC56_2.fastq.gz, total: 54143947, mapped (excluding rRNA, tRNA):28565765, 132bp, single-end
EUSeq.GFP.AID.Rad21.NSC_IAA_500uM_TC60_CC56_1.fastq.gz, total: 50431987, mapped (excluding rRNA, tRNA):23210749, 132bp, single-end
EUSeq.GFP.AID.Rad21.NSC_IAA_500uM_TC60_CC56_2.fastq.gz, total: 50035877, mapped (excluding rRNA, tRNA):26043886, 132bp, single-end
EUSeq.GFP.AID.Rad21.NSC_IAA_500uM_TC120_CC56_1.fastq.gz, total: 57933246, mapped (excluding rRNA, tRNA):26530518, 132bp, single-end
EUSeq.GFP.AID.Rad21.NSC_IAA_500uM_TC120_CC56_2.fastq.gz, total: 47171992, mapped (excluding rRNA, tRNA):14714775, 132bp, single-end
EUSeq.GFP.AID.Rad21.NSC_IAA_500uM_TC240_CC56_1.fastq.gz, total: 53498665, mapped (excluding rRNA, tRNA):26173221, 132bp, single-end
EUSeq.GFP.AID.Rad21.NSC_IAA_500uM_TC240_CC56_2.fastq.gz, total: 50727296, mapped (excluding rRNA, tRNA):23934216, 132bp, single-end
EUSeq.GFP.AID.Rad21.NSC_A485_10uM_TC60_CC56_1.fastq.gz, total: 48556419, mapped (excluding rRNA, tRNA):23455010, 132bp, single-end
EUSeq.GFP.AID.Rad21.NSC_A485_10uM_TC60_CC56_2.fastq.gz, total: 57198713, mapped (excluding rRNA, tRNA):24723013, 132bp, single-end

EUSeq.GFP.AID.Rad21.ESC_Ctrl_TC0_CC52_1.fastq.gz, total: 28219652, mapped (excluding rRNA, tRNA):13541043, 132bp, single-end
EUSeq.GFP.AID.Rad21.ESC_Ctrl_TC0_CC53_1.fastq.gz, total: 58460033, mapped (excluding rRNA, tRNA):27382270, 82bp, single-end
EUSeq.GFP.AID.Rad21.ESC_Ctrl_TC0_CC53_2.fastq.gz, total: 53798231, mapped (excluding rRNA, tRNA):24170540, 82bp, single-end
EUSeq.GFP.AID.Rad21.ESC_Ctrl_TC0_CC53_3.fastq.gz, total: 61775977, mapped (excluding rRNA, tRNA):30367238, 82bp, single-end
EUSeq.GFP.AID.Rad21.ESC_Ctrl_TC0_CC53_4.fastq.gz, total: 56607381, mapped (excluding rRNA, tRNA):26599994, 82bp, single-end
EUSeq.GFP.AID.Rad21.ESC_Ctrl_TC0_CC53_5.fastq.gz, total: 71895791, mapped (excluding rRNA, tRNA):33218201, 82bp, single-end
EUSeq.GFP.AID.Rad21.ESC_Ctrl_TC0_CC53_6.fastq.gz, total: 59612965, mapped (excluding rRNA, tRNA):27150564, 82bp, single-end
EUSeq.GFP.AID.Rad21.ESC_Ctrl_TC0_CC53_7.fastq.gz, total: 70230572, mapped (excluding rRNA, tRNA):31681225, 82bp, single-end
EUSeq.GFP.AID.Rad21.ESC_Ctrl_TC0_CC53_8.fastq.gz, total: 54155638, mapped (excluding rRNA, tRNA):25266713, 82bp, single-end
EUSeq.GFP.AID.Rad21.ESC_Ctrl_TC0_CC53_9.fastq.gz, total: 54791813, mapped (excluding rRNA, tRNA):25840812, 82bp, single-end
EUSeq.GFP.AID.Rad21.ESC_Ctrl_TC0_CC53_10.fastq.gz, total: 57915265, mapped (excluding rRNA, tRNA):27646378, 82bp, single-end
EUSeq.GFP.AID.Rad21.ESC_Ctrl_TC0_CC53_11.fastq.gz, total: 58831257, mapped (excluding rRNA, tRNA):27144694, 82bp, single-end

EUSeq.GFP.AID.Rad21.ESC_IAA_500uM_TC240_CC52_1.fastq.gz, total: 30286974, mapped (excluding rRNA, tRNA):14545398, 132bp, single-end

EUSeq.GFP.AID.Rad21.ESC_IAA_500uM_TC240_CC53_1.fastq.gz, total: 57175734, mapped (excluding rRNA, tRNA):27015488, 82bp, single-end
EUSeq.GFP.AID.Rad21.ESC_IAA_500uM_TC240_CC53_2.fastq.gz, total: 56839342, mapped (excluding rRNA, tRNA):26485244, 82bp, single-end
EUSeq.GFP.AID.Rad21.ESC_IAA_500uM_TC240_CC53_3.fastq.gz, total: 55248304, mapped (excluding rRNA, tRNA):25509358, 82bp, single-end
EUSeq.GFP.AID.Rad21.ESC_IAA_500uM_TC240_CC53_4.fastq.gz, total: 57091660, mapped (excluding rRNA, tRNA):26336268, 82bp, single-end
EUSeq.GFP.AID.Rad21.ESC_IAA_500uM_TC240_CC53_5.fastq.gz, total: 80134100, mapped (excluding rRNA, tRNA):37153615, 82bp, single-end
EUSeq.GFP.AID.Rad21.ESC_IAA_500uM_TC240_CC53_6.fastq.gz, total: 61141076, mapped (excluding rRNA, tRNA):28136519, 82bp, single-end
EUSeq.GFP.AID.Rad21.ESC_IAA_500uM_TC240_CC53_7.fastq.gz, total: 53578200, mapped (excluding rRNA, tRNA):23732150, 82bp, single-end
EUSeq.GFP.AID.Rad21.ESC_IAA_500uM_TC240_CC53_8.fastq.gz, total: 61845713, mapped (excluding rRNA, tRNA):28202924, 82bp, single-end
EUSeq.GFP.AID.Rad21.ESC_IAA_500uM_TC240_CC53_9.fastq.gz, total: 59599989, mapped (excluding rRNA, tRNA):27559979, 82bp, single-end
EUSeq.GFP.AID.Rad21.ESC_IAA_500uM_TC240_CC53_10.fastq.gz, total: 55666119, mapped (excluding rRNA, tRNA):26191806, 82bp, single-end
EUSeq.GFP.AID.Rad21.ESC_IAA_500uM_TC240_CC53_11.fastq.gz, total: 54016766, mapped (excluding rRNA, tRNA):24467447, 82bp, single-end

EUSeq.GFP.dTAG.1.577aa.CTCF_bTAG.Alfa.266.736aa.CTCF.ESC_Ctrl_TC0_strand_CC71_1.fastq.gz, total:  30853200, mapped (excluding rRNA, tRNA):12928287, 132bp, single-end
EUSeq.GFP.dTAG.1.577aa.CTCF_bTAG.Alfa.266.736aa.CTCF.ESC_Ctrl_TC0_strand_CC71_2.fastq.gz, total: 30750153, mapped (excluding rRNA, tRNA):13868505, 132bp, single-end
EUSeq.GFP.dTAG.1.577aa.CTCF_bTAG.Alfa.266.736aa.CTCF.ESC_AGB1_100nM_TC120_strand_CC71_1.fastq.gz, total: 27530797, mapped (excluding rRNA, tRNA):13226384, 132bp, single-end
EUSeq.GFP.dTAG.1.577aa.CTCF_bTAG.Alfa.266.736aa.CTCF.ESC_AGB1_100nM_TC120_strand_CC71_2.fastq.gz, total: 29982928, mapped (excluding rRNA, tRNA):15474652, 132bp, single-end
EUSeq.GFP.dTAG.1.577aa.CTCF_bTAG.Alfa.266.736aa.CTCF.ESC_dTAG13_200nM_TC120_strand_CC71_1.fastq.gz, total: 31059148, mapped (excluding rRNA, tRNA):11848765, 132bp, single-end
EUSeq.GFP.dTAG.1.577aa.CTCF_bTAG.Alfa.266.736aa.CTCF.ESC_dTAG13_200nM_TC120_strand_CC71_2.fastq.gz, total: 29079519, mapped (excluding rRNA, tRNA):15908468, 132bp, single-end
EUSeq.GFP.dTAG.1.577aa.CTCF_bTAG.Alfa.266.736aa.CTCF.ESC_dTAG13_200nM_AGB1_100nM_TC120_strand_CC71_1.fastq.gz, total: 37541169, mapped (excluding rRNA, tRNA):17875249, 132bp, single-end
EUSeq.GFP.dTAG.1.577aa.CTCF_bTAG.Alfa.266.736aa.CTCF.ESC_dTAG13_200nM_AGB1_100nM_TC120_strand_CC71_2.fastq.gz, total: 27654266, mapped (excluding rRNA, tRNA):14149272, 132bp, single-end

| | |
|---|---|
| Antibodies | The following antibodies were used for ChIP-seq.<br>Rpb1 NTD (D8L4Y) Rabbit mAb #14958, Cell Signaling Technology |
| Peak calling parameters | Reads were mapped to the reference genome by using bwa aln with default parameters (BWA version 0.7.10). Multi-mapped reads, duplicated reads, or reads with more than three mismatches were removed by samtools. Reads mapped to the DAC Blacklisted Regions (https://www.encodeproject.org/annotations/ENCSR636HFF/) were omitted from the downstream analysis. Peak regions were called using LanceOtron with default model (wide-and-deep_jan-2021) (doi: https://doi.org/10.1101/2021.01.25.428108 ). The poorly enriched peaks of maximum peak height < 8 reads mapped per million (rpm) were omitted. |
| Data quality | FASTQC was used for quality check of sequencing reads. |
| Software | Cutadapt, fastqc(0.12.1), STAR(2.6.1a), Bedtools(2.23), HTseq(0.11.1), bwa meme (version 1.0.4), samtools(1.4), Lanceotron(20210215), R(4.1.1), ggplot2(3.3.5), DESeq(1.32.0), bigWigMerge,  deeptools(3.5.2), IGV(2.16), Picard-tools(2.9.1), epic2(0.0.47), GEM(3.4), STREME(5.5.4), FIMO (5.5.4), higlass-python(v1.2.0) |

