## [Peer Review File · Nature Genetics]

Disentangling the architectural and non-architectural functions of CTCF and cohesin in gene regulation

Corresponding Author: Professor Chunaram Choudhary

A version of this paper was originally rejected for publication by Nature Genetics, however that decision was reconsidered after appeal by the authors.

Version 0:

Decision Letter:

11th Dec 2024

Dear Professor Choudhary,

Thank you for submitting your manuscript entitled "A unified model of gene expression control by cohesin and CTCF", for consideration. I regret that we are unable to publish it in Nature Genetics.

As you may know, we decline a substantial proportion of manuscripts without sending them to referees, so that they may be sent elsewhere without delay. Our editorial judgments are based on such considerations as the degree of advance provided, the breadth of potential interest to researchers and timeliness.

In this case, your manuscript has not matched our criteria for further consideration at Nature Genetics, and we think it would find a more suitable outlet in another journal, such as Nature Communications. To discover more about our other journals and, should you wish, have your paper considered by the editors, please use the link to the manuscript transfer service provided in the footnote below. Please note that we have not consulted with our editorial colleagues at other journals and that they will make their own independent editorial decision.

Please be assured that this editorial decision does not represent a criticism of the quality of your work, nor are we questioning its value to others working in this area. We hope that you will rapidly receive a more favorable response elsewhere.

I am sorry that we cannot respond more positively on this occasion.

Sincerely,
Chiara

Chiara Anania, PhD
Associate Editor
Nature Genetics
<https://orcid.org/0000-0003-1549-4157>

Although we cannot offer to publish your manuscript, my colleagues at Nature Communications will send your manuscript out for external review. To transfer your manuscript please use our manuscript transfer portal. You will not have to re-supply manuscript metadata and files, unless you wish to make modifications. For more information, please see our [manuscript transfer FAQ](http://www.nature.com/authors/author_resources/transfer_manuscripts.html?WT.mc_id=EMI_NPG_1511_AUTHORTRANSF&WT.ec_id=AUTHOR) page.

Version 1:

Decision Letter:

IMPORTANT: Please note the reference number: NG-A67394R-Z Choudhary. This number must be quoted whenever you communicate with us regarding this paper.

3rd Feb 2025

Dear Dr. Choudhary,

Thank you for your message of asking us to reconsider our decision on your manuscript "A unified model of gene expression control by cohesin and CTCF". I have now discussed the points of your letter with my colleagues, and we think that you have some valid points. We therefore invite you to revise your manuscript along the lines that you propose.

When preparing a revision, please ensure that it fully complies with our editorial requirements for format and style; details can be found in the Guide to Authors on our website (<http://www.nature.com/ng/>).

Please be sure that your manuscript is accompanied by a separate letter detailing the changes you have made and your response to the points raised. At this stage we will need you to upload:

1) a copy of the manuscript in MS Word .docx format.

2) The Editorial Policy Checklist:

<https://www.nature.com/documents/nr-editorial-policy-checklist.pdf>

3) The Reporting Summary:

(Here you can read about the role of the Reporting Summary in reproducible science:

<https://www.nature.com/news/announcement-towards-greater-reproducibility-for-life-sciences-research-in-nature-1.22062>)

Please use the link below to be taken directly to the site and view and revise your manuscript:

Link Redacted

With kind wishes,
Chiara

Chiara Anania, PhD

Associate Editor

Nature Genetics

<https://orcid.org/0000-0003-1549-4157>

Version 2:

Decision Letter:

7th Apr 2025

Dear Professor Choudhary,

Your Article, "A unified model of gene expression control by cohesin and CTCF" has now been seen by 2 referees. You will see from their comments copied below that while they find your work of considerable potential interest, they have raised quite substantial concerns that must be addressed. In light of these comments, we cannot accept the manuscript for publication, but would be interested in considering a revised version that addresses these serious concerns.

We hope you will find the referees' comments useful as you decide how to proceed. If you wish to submit a substantially revised manuscript, please bear in mind that we will be reluctant to approach the referees again in the absence of major revisions.

To guide the scope of the revisions, the editors discuss the referee reports in detail within the team, including with the chief editor, with a view to identifying key priorities that should be addressed in revision and sometimes overruling referee

requests that are deemed beyond the scope of the current study. In this case, we ask you to address reviewers' comments in full. We hope that you will find the prioritised set of referee points to be useful when revising your study. Please do not hesitate to get in touch if you would like to discuss these issues further.

If you choose to revise your manuscript taking into account all reviewer and editor comments, please highlight all changes in the manuscript text file. At this stage we will need you to upload a copy of the manuscript in MS Word .docx or similar editable format.

*2) If you have not done so already please begin to revise your manuscript so that it conforms to our Article format instructions, available here. Refer also to any guidelines provided in this letter.

*3) Include a revised version of any required Reporting Summary: <https://www.nature.com/documents/nr-reporting-summary.pdf>

Please be aware of our guidelines on digital image standards.

EXTENDED DATA FIGURES

Link Redacted

If you wish to submit a suitably revised manuscript we would hope to receive it within 6 months. If you cannot send it within this time, please let us know. We will be happy to consider your revision so long as nothing similar has been accepted for publication at Nature Genetics or published elsewhere. Should your manuscript be substantially delayed without notifying us in advance and your article is eventually published, the received date would be that of the revised, not the original, version.

Nature Genetics is committed to improving transparency in authorship. As part of our efforts in this direction, we are now requesting that all authors identified as 'corresponding author' on published papers create and link their Open Researcher and Contributor Identifier (ORCID) with their account on the Manuscript Tracking System (MTS), prior to acceptance. ORCID helps the scientific community achieve unambiguous attribution of all scholarly contributions. You can create and link your ORCID from the home page of the MTS by clicking on 'Modify my Springer Nature account'. For more information please visit please visit www.springernature.com/orcid.

Thank you for the opportunity to review your work.

Sincerely,
Chiara

Chiara Anania, PhD
Associate Editor
Nature Genetics
<https://orcid.org/0000-0003-1549-4157>

Referee expertise:

Referee #1: gene regulation

Referee #2: genomics, bioinformatics

Referee #3:

Reviewers' Comments:

Reviewer #1 (Remarks to the Author):

In this manuscript, Narita and colleagues use several CTCF and cohesin degron approaches combined with CBP/P300-dependant enhancer inhibition to decipher the roles of CTCF and cohesin in enhancer-promoter communication, transcription, and CTCF's function as a transcription factor. In contrast to previous studies, the authors show that the loss of cohesin and CTCF induces widespread, yet subtle, changes in transcription, reinforcing the predominant role of these architectural proteins in gene regulation. Additionally, they distinguish the roles of CTCF as both an architectural protein and a transcription factor binding directly at promoters. Together, this manuscript provides a comprehensive description of CTCF and cohesin function.

The approaches used in this manuscript are elegant, powerful, and well-controlled. It presents a novel and balanced perspective on the different functions of CTCF and, importantly, demonstrates how the 3D genome organization mediated by cohesin and CTCF is crucial to gene regulation. However, the manuscript in its current form is somewhat difficult to follow, and the figures are not very intuitive. Provided the authors address these concerns, as well as a few additional points raised by this reviewer, I support the publication of this manuscript in Nature Genetics.

Major Points

1. The introduction does not sufficiently explain previous work on cohesin, CBP/P300 and H2BNTac performed by the authors. For a broader audience, it is not immediately clear that CBP/P300 might not directly affect promoter function. Also, a more comprehensive introduction to degron approaches used so far, along with their respective limitations, would be important.
2. The authors state that their results indicate that cohesin specifically controls CBP/P300 enhancers (See for instance line 866). However, this is not clearly supported by the data, as other long-range enhancer categories could also depend on cohesin. The authors should clarify whether they view CBP/P300 enhancers as a general model for enhancers or as a unique class that specifically depends on cohesin.
3. The authors classify certain regions as enhancer-rich, yet the manuscript does not measure enhancer density or population. These aspects should be quantified.
4. In the discussion on Drosophila, it is unclear why enhancer blocking and cohesin loop formation would be considered different processes, given that both rely on the same loop extrusion mechanism.
5. The manuscript's writing and figures should be extensively improved, as it contains a substantial amount of data and key points but is currently difficult to read and understand.

Minor Points

The CTCFY-YY-AA degron approach should be better described in the text and displayed in the main figure. The setup is not immediately clear and the possibility that cells can bear such an allele and divide normally is not obvious for non-specialists. Some CTCF-AID signals significantly decrease or disappear in the FC approach, despite having a base mean of around 7, suggesting they are far enough from noise to be considered reliable. It is unclear whether TPM thresholding is the best approach, especially since requiring TPM > 15 in the treatment may exclude potentially true downregulated genes.

S1D (or B): numbers of genes in each category would be an improvement

To show how the genes deregulated in the CTCF degron are affected in the RAD21 degron, a plot with the log₂fc of CTCF in x and RAD21 in y (or the contrary) and color the dots by ID/HD1/HD2 could be produced

S3B, in the example of Klf4, the loops in the Ctrl are difficult to visualize, could they be shown with arrows?

Line 275, a definition of proximal enhancer should be provided by the authors

In Figure 2A it is mentioned that enhancers are defined by H3K27ac and H2BK20ac, yet in the main text this is not explained. The same also happens for Supplemental Figure 6A, 6B. The use of both histone marks should be referred to in the main text.

Line 294: a word is missing in the sentence.

Line 309: authors should clarify what they mean by 'co-enrichment of cohesin'.

Line 311: authors should state that by distal CTCF peaks, they are referring to all the peaks that don't fall in their definition of CTCF peaks in gene promoters (+/- 200bp)

Line 312: The ratio does not appear to have been calculated by the authors.

S4B: the numbers of peaks used to build the profile should be indicated (at least in the legend).

Figure 3: panels are labeled with lowercase while it is uppercase everywhere else

The luciferase experiment uses as a baseline control the gene promoter, but it is not clear if it is more or less active than without promoter. Because of this: it is unclear if Psm4 and Actn4 promoters are unaffected because they are never active in first place? It would also be important to know if the chosen genes are also affected by the CTCF mutant degron.

Line 488: The authors conclude that promoters with reverse motif orientation exhibit no specific bias for downregulation. The data shows in fact that there is a bias toward less downregulation. Could you rephrase it?

Figure 5B: The effect of BCB and CBC on the rescue is not easy to catch from the graph, maybe the median log2FC could be written in the legend.

Figure 6A: a horizontal dot line is missing at -1.

Line 656: There is probably a typo: TSS downstream regions should be TES downstream regions.

Figure 7A: the 3 panels are not described. They are probably 1h, 2h, 4h of RAD21 depletion. This should be in the legend.

Figure 7B: the legend for RAD21 up is missing.

Line 1245: gencode human release 29 is GRCh38.p12 and release 19 is GRCh37.p13, can this be fixed? Also specify which version of Gencode you used (comprehensive/basic etc...).

Line 1249: A threshold of TPM > 2 is used but the RNA-seq data used for this filter is not specified.

Line 1256: The data provenance should be specified (publication GEO etc...).

Line 1271: It is not clear how the log2FC were calculated. Are they coming from 'results' DESeq2 or where they manually computed? Both for the initial 2 replicates and for the 12 supplemental replicates. Are the control and the treated matched, if so would the introduction of the replicate as covariate in DESeq2 analysis help to better estimate the log2FC? To assess the reproducibility between replicates, it would be good to have a global correlation clustering between the initial 2 replicates and the 12 new replicates.

Line 1400: The title of the section is wrong.

A supplementary table listing all the data generated in this study would be useful.

On GEO, there are 'Stranded EU-seq'. Were these samples used in this study? Table of counts should be added to the GEO submission for the EU-seq.

A supplementary table listing all genes tested and their category in each analysis would be useful (CTCF-AID, RAD21-AID, A-485, CTCF mutant - AID vs CTCF-AID, CTCF mutant -AID).

Accessibility changes at promoters might be independent of CTCF and could result from transcriptional changes.

Reviewer #2 (Remarks to the Author):

Narita and colleagues address the role of cohesin and CTCF in gene regulation. Using mES degron lines, they investigate the effect of different cohesin, CTCF, and enhancer-activity perturbations, and thereby provide insight into the looping-dependent and -independent roles of cohesin and CTCF in gene regulation.

The manuscript contains a lot of data and is not very easy to read/follow. Some of the data provide interesting new insights. However, there are also several findings that are not novel even though they are presented as if they are.

As explained in more detail below, the part of the manuscript about the role of cohesin in gene regulation is not novel. The analyses are still of interest, but should be presented as confirmatory and previous literature on this topic should be properly acknowledged.

The findings concerning the looping/enhancer-independent role of CTCF in gene regulation are more novel and certainly of interest to the field; we would therefore recommend that the authors focus their manuscript on this part and adapt their abstract and title accordingly, as the current versions contain heavily overstated claims.

We leave it up to the judgment of the Editor whether a re-written manuscript with a direct activator role of CTCF as the main finding is significant enough to justify publication in Nature Genetics.

Major comments

1. The justification for the first part of the manuscript "Several studies have examined transcriptional changes following acute RAD21 and CTCF depletion^{20,25-29}, yet they have not addressed why these perturbations affect only a limited set of genes or whether the affected genes are enhancer-regulated." is not correct. There are several papers that have addressed this (although they have not looked at this in a genome-wide manner). First papers that come to mind (although we are sure that there are more) are: Thiecke et al Cell Reports 2020, Aljahani et al Nature Communications 2022, Kane et al NSMB 2022, Rinzema et al NSMB 2022, Goel et al Nature Genetics 2023. Furthermore, in several recent review papers on the topic of loop extrusion and enhancers and/or gene regulation, notions that are presented in the paper as novel findings are already discussed.

2. The statement in the abstract and similar statements throughout the manuscript "cohesin appears to exclusively support gene activation via CBP/p300-dependent enhancers" should be removed. The authors have only investigated CBP/p300-

dependent enhancers and can therefore not make any claims that other types are not affected. It would be better not to overinterpret the data and more precisely discuss the data without overstatements.

3. We suggest that the authors re-structure their manuscript and separate the part about cohesin and CTCF. Now they start with cohesin, then focus on CTCF, and switch back to cohesin at the end, which makes the manuscript unnecessarily difficult to follow.

4. The statement that the effects of cohesin and CTCF perturbations on transcription are bigger than previously appreciated is somewhat unfair, as the authors simply decide to take non-significant changes into consideration, which is highly controversial. With this approach, it would be good to only include datapoints for which the fold-change is at least in the same direction for the independent replicates, but it is unclear whether this is the case. If that would be changed/clarified, we have no specific issues with this strategy (as the validation is relatively convincing), but it was known before that there are many changes that are not statistically significant. The authors should interpret these data with more caution and less overstatements, especially because they are not backed up by any statistical analysis. In this regard, it is of interest that previous work has showed increased variability of expression upon cohesin perturbation (Hafner et al *Molecular Cell* 2023). This could also explain the lack of statistically significant changes and should be acknowledged.

5. It is unfortunate that the study relies on only two biological replicates for the most interesting part about CTCF. More replicates would be valuable and contribute to the robustness of the findings.

6. Figure 4F: The orientation of CTCF should be considered in relation to the direction of transcription of the gene, but this does not seem to be the case.

7. Page 34: "Depending on the position of CTCF binding relative to the TSS, CTCF can act as a repressor of either anti-sense or sense transcript." This statement is only supported by a few examples in Figure 6 and Supplemental Figure 7. It would be more convincing to show a quantitative analysis of this effect across all relevant genes.

8. Is it possible that the effects of CTCF perturbation on RNAPII recruitment are secondary to the observed changes in chromatin accessibility? If so, this should be acknowledged.

9. Page 42: "By analyzing the architectural and non-architectural roles of cohesin and CTCF (Figure 7D), this work introduces four major conceptual advances to these ongoing discussions: (1) Following acute cohesin and CTCF depletion, the extent of gene dysregulation is much broader than previously appreciated. (2) Among diverse enhancer types posited, there is likely only one enhancer type – the CBP/p300-dependent type – that use cohesin-dependent looping for gene activation. (3) Beyond its architectural roles, CTCF functions as a position and orientation-specific transcription repressor and activator, controlling the expression of housekeeping genes, including ones that are pan-essential for mammalian cell proliferation. (4) Despite their differences in anchoring cohesin loops, CTCF and CTCFL share transcription activation ability."

As discussed above, the first two conclusions are not novel.

Minor comments

1. There are a few sentences that do not make sense:

Page 2: "Using plasmid-based reporter assay, early reports suggested that CTCF can function as transcription activator activator21 and activator 22, but contemporary models raise doubts about in-vivo relevance of these observations and interpret CTCF's role in gene regulation almost exclusively in the context of its architectural function in genome folding 23,24."

Page 29: "CTCF has a vertebrate-specific, called BORIS (CTCFL), is usually only expressed in the testis, but aberrant expression is found in cancers 56. CTCF and BORIS share high sequence identity in their CTCF in DNA binding zinc fingers (ZFs) but exhibit minimal conservation in non-DNA binding N- and C-termini 57."

2. Validation of the depletion efficiency is not consistently included. Even though some of the used cell lines have been characterized previously, it is still important to validate the depletion efficiency, as the authors use shorter depletion times than previously described. As incomplete depletion may also explain the limited effects in some cases, it would be important to include this – ideally using ChIP-seq data, so that the degree of depletion can be directly linked to effect sizes at individual genes.

3. It seems that the authors did not make use of spike-ins for normalization. This may however be useful after genome-wide perturbations and may help the authors in finding more robust changes. It would be good if the authors can consider including this or at least comment on why they may think it is not useful or appropriate in their study.

4. Fig. 1B: It would be nice to add the numbers of genes included in each category.

5. Page 11: Some of the examples described here (Prdm14/Slco5a1 locus in context of cohesin and CTCF depletion) have been discussed previously in Aljahani et al *Nature Communications* 2022. This should be acknowledged.

6. The notion that enhancer-promoter interactions are not only dependent on loop extrusion is well established in the field. The authors mention LDB1 and YY1 in this context. Recently, an important role for Mediator and RNAPII has become clear as well and this would be good to discuss.

Version 3:

Decision Letter:

Our ref: NG-A67394R2

11th Jun 2025

Dear Dr. Choudhary,

Thank you for submitting your revised manuscript "A unified model of gene expression control by cohesin and CTCF" (NG-A67394R2). It has now been seen by the original referees and their comments are below. The reviewers find that the paper has improved in revision, and therefore we'll be happy in principle to publish it in Nature Genetics, pending minor revisions to satisfy the referees' final requests and to comply with our editorial and formatting guidelines.

Sincerely,
Chiara

Chiara Anania, PhD
Associate Editor
Nature Genetics
<https://orcid.org/0000-0003-1549-4157>

Reviewer #1 (Remarks to the Author):

The authors have satisfactorily addressed most of our points. However, some clarifications on the types of enhancers existing is needed.

In the rebuttal, the authors write that there is not much evidence supporting different enhancer types in vivo: "However, it should be noted that the idea of multiple enhancer types mainly rests on coactivator dependencies of enhancers in plasmid-based reporter assays (Nature 2015, PMID: 25517091; Mol Cell 2022, PMID: 35594855; Nat Genet. 2021, PMID: 34183853; Nature 2022, PMID: 35650434).", "Currently, the evidence supporting multiple enhancer types in vivo is limited.", and "We are unaware of examples of distal enhancers in native chromatin that activate gene expression independently of CBP/p300" Yet, in the manuscript introduction, they write, "Metazoans have diverse enhancer types(27-31) » and "Among the functional enhancers identified through CRISPR interference, those linked to proximal genes mostly function using CBP/p300 but those skipping active genes mostly function without CBP/p300"

To us these points in the manuscript are in contradiction with the rebuttal. Moreover, this last point "Among the functional enhancers..." is also controversial as it is based on a non-peer-reviewed paper (the one that was previously co-submitted but rejected). Moreover, it suggests that there is another class of enhancers (that can skip active genes) while the results suggest that if there are, they do not use cohesin. This confuses the message.

In this perspective, this reviewer wonders why there is so much emphasis on enhancer types/classes in the introduction? Could this be removed and acknowledge that most enhancers are dependant on CBP/P300 without specifying what remains as it confuses the message.

Reviewer #2 (Remarks to the Author):

We would like to thank the authors for addressing most of our concerns. We think that the revised manuscript has improved a lot. However, as raised in our initial review, we are not very comfortable with the current title. It is a very "strong" title and in our view it does not reflect the content of the paper well, as it overstates the contribution of this manuscript to our understanding of cohesin/CTCF in gene regulation (as we already pointed out, the work on cohesin is mostly confirmatory) and understates the fact that there are still many open questions about the exact molecular functions of these proteins.

Version 4:

Decision Letter:

In reply please quote: NG-A67394R3 Choudhary

11th Oct 2025

Dear Dr. Choudhary,

I am delighted to say that your manuscript "Disentangling the architectural and non-architectural functions of CTCF and cohesin in gene regulation" has been accepted for publication in an upcoming issue of Nature Genetics.

Your paper will be published online after we receive your corrections and will appear in print in the next available issue. You can find out your date of online publication by contacting the Nature Press Office (press@nature.com) after sending your e-proof corrections.

Authors may need to take specific actions to achieve compliance with funder and institutional open access mandates. If your research is supported by a funder that requires immediate open access (e.g. according to [Plan S principles](https://www.springernature.com/gp/open-science/plan-s-compliance) or the [NIH public access policy](https://www.springernature.com/gp/open-science/us-federal-agency-compliance)) then you should select the gold OA route, and we will direct you to the compliant route where possible. Because authors warrant under our subscription licensing terms that they haven't committed to licensing any version of their article under a licence inconsistent with the terms of our agreement – including the applicable embargo period – publication under the subscription model isn't suitable for authors whose funders require no embargo.

If you have not already done so, we strongly recommend that you upload the step-by-step protocols used in this manuscript to protocols.io. protocols.io is an open online resource that allows researchers to share their detailed experimental know-how. All uploaded protocols are made freely available and are assigned DOIs for ease of citation. Protocols can be linked to any publications in which they are used and will be linked to from your article. You can also establish a dedicated workspace to collect all your lab Protocols. By uploading your Protocols to protocols.io, you are enabling researchers to more readily reproduce or adapt the methodology you use, as well as increasing the visibility of your protocols and papers. Upload your Protocols at <https://protocols.io>. Further information can be found at <https://www.protocols.io/help/publish-articles>.

Sincerely,

Chiara Anania, PhD
Associate Editor
Nature Genetics
<https://orcid.org/0000-0003-1549-4157>

Click here if you would like to recommend Nature Genetics to your librarian
<http://www.nature.com/subscriptions/recommend.html#forms>

** Visit the Springer Nature Editorial and Publishing website at http://editorial-jobs.springernature.com?utm_source=ejp_NGen_email&utm_medium=ejp_NGen_email&utm_campaign=ejp_NGen for more information about our career opportunities. If you have any questions please click [here](mailto:editorial.publishing.jobs@springernature.com).

A point-by-point response to reviewers' comments

We sincerely thank both reviewers for their thorough evaluation of the manuscript and insightful feedback. We are pleased that they found our work interesting. We greatly appreciate their constructive suggestions.

In response to the reviewers' comments, we have restructured the manuscript to enhance its focus. The revised version now presents data on cohesin in the first part and CTCF in the latter. Additionally, we have extensively revised the text for greater clarity, incorporated the suggested references, and adjusted our claims to ensure a balanced presentation.

In addition to addressing the reviewers' comments, we have included new data on CTCF generated since the original submission (**Fig. 7a–b, Supplementary Figs. 13–14**). These results demonstrate that CTCF's non-architectural functions depend on its C-terminus, further reinforcing our original conclusions. If acceptable to the reviewers and editors, we would like to incorporate these findings into the current manuscript. Otherwise, we are prepared to remove them and present them in a separate publication.

Once again, we deeply appreciate the reviewers' invaluable feedback, which has greatly helped in improving the manuscript. Below, we provide point-by-point responses to their comments.

Reviewer #1

In this manuscript, Narita and colleagues use several CTCF and cohesin degron approaches combined with CBP/P300-dependant enhancer inhibition to decipher the roles of CTCF and cohesin in enhancer-promoter communication, transcription, and CTCF's function as a transcription factor. In contrast to previous studies, the authors show that the loss of cohesin and CTCF induces widespread, yet subtle, changes in transcription, reinforcing the predominant role of these architectural proteins in gene regulation. Additionally, they distinguish the roles of CTCF as both an architectural protein and a transcription factor binding directly at promoters. Together, this manuscript provides a comprehensive description of CTCF and cohesin function.

The approaches used in this manuscript are elegant, powerful, and well-controlled. It presents a novel and balanced perspective on the different functions of CTCF and, importantly, demonstrates how the 3D genome organization mediated by cohesin and CTCF is crucial to gene regulation. However, the manuscript in its current form is somewhat difficult to follow, and the figures are not very intuitive. Provided the authors address these concerns, as well as a few additional points raised by this reviewer, I support the publication of this manuscript in Nature Genetics.

We thank the reviewer for very helpful suggestions. We sincerely appreciate their positive remarks and highly constructive suggestions. Below, we provide detailed responses to their comments.

Major Points

1. The introduction does not sufficiently explain previous work on cohesin, CBP/P300 and H2BNTac performed by the authors. For a broader audience, it is not immediately clear that CBP/P300 might not directly affect promoter function. Also, a more comprehensive introduction to degron approaches used so far, along with their respective limitations, would be important.

We have expanded the introduction to include a mention of our previous work on CBP/p300 and H2BNTac. A cartoon diagram showing degron-based protein depletion is now included in the revised **Supplementary Fig. 1a**. Also, in the discussion section we included the following sentence to acknowledge its limitation. “...*although degron systems enable rapid protein depletion, they do not eliminate the target protein entirely; residual RAD21 and CTCF are likely to retain partial functionality.*” Of note, this is a general limitation of degron-based methods and not unique to our study.

2. The authors state that their results indicate that cohesin specifically controls CBP/P300 enhancers (See for instance line 866). However, this is not clearly supported by the data, as other long-range enhancer categories could also depend on cohesin. The authors should clarify whether they view CBP/P300 enhancers as a general model for enhancers or as a unique class that specifically depends on cohesin.

As the reviewer notes, various types of enhancers have been described in both *Drosophila* and human cells. However, it should be noted that the idea of multiple enhancer types mainly rests on coactivator dependencies of enhancers in plasmid-based reporter assays (Nature 2015, PMID: 25517091; Mol Cell 2022, PMID: 35594855; Nat Genet. 2021, PMID: 34183853; Nature 2022, PMID: 35650434).

Currently, the evidence supporting multiple enhancer types in vivo is limited. While it has been shown that p53 can activate genes without Mediator (PMID: 35650434), it is not clear whether those p53-regulated genes are activated by enhancers and if this is independent of CBP/p300. We are unaware of examples of distal enhancers in native chromatin that activate gene expression independently of CBP/p300. Nearly all well-characterized enhancers—including those regulating *Nanog*, *Oct4*, *Sox2*, *Klf4*, *Klf2*, *Myc*, *Sox9*, *Hox*, and hemoglobin genes—appear to operate via CBP/p300.

In our data, genes downregulated by RAD21 depletion in mESC are globally downregulated by CBP/p300 inhibition (**Fig. 1b**). Also, in NPC, more than 100 genes are downregulated by >2-fold after 2-4h of RAD21^{AID} depletion, and 97-99% of them are regulated by CBP/p300 (**Fig. 1f**). If one considers ~5% false positives in the regulated genes, this overlap approaches the theoretical maximum. Conversely, among the A-485-regulated gene class, RAD21-downregulated are specifically enriched in the A-485-downregulated genes (**Fig. 1d, g**). If cohesin broadly activated genes via other enhancer types, we would expect that many of the strongly downregulated genes after RAD21 depletion will remain unaffected by CBP/p300 inhibition. However, this is not the case.

Based on this observation, we originally concluded that cohesin “almost exclusively” promotes gene activation via CBP/p300. However, as the reviewer correctly notes—and as

also emphasized by Reviewer #2—we cannot fully exclude the possibility that cohesin also operates through other enhancer types. We appreciate and respect the reviewers' perspectives. To acknowledge this, we mention that cohesin-regulated genes are biased for their dependency on CBP/p300, but have removed the claim that cohesin “almost exclusively” functions with CBP/p300.

3. The authors classify certain regions as enhancer-rich, yet the manuscript does not measure enhancer density or population. These aspects should be quantified.

Thank you for the suggestion. Due to the restructuring of the figures, the original Fig. 2a is now moved to **Supplemental Fig. 6**, and quantification data are included.

4. In the discussion on *Drosophila*, it is unclear why enhancer blocking and cohesin loop formation would be considered different processes, given that both rely on the same loop extrusion mechanism.

While functionally CTCF acts as an enhancer blocker for both *Drosophila* and vertebrates, the mechanisms seem different. In vertebrates, CTCF appears to block enhancer action by halting cohesin loops. In contrast, *Drosophila* lacks evidence for CTCF-mediated loop anchoring (PMID: 31264253). Instead, *Drosophila* CTCF and a multitude of other insulator-binding proteins commonly interact with Cp190, and insulators appear to function as boundary elements (PMID: 36735780).

5. The manuscript's writing and figures should be extensively improved, as it contains a substantial amount of data and key points but is currently difficult to read and understand.

We have done our very best to improve the clarity of the text and figures. We revised the introduction, reordered the results section, and streamlined the discussion. If the reviewer has further suggestions, we would be happy to work on them.

Minor Points

The CTCF^{YY-AA} degon approach should be better described in the text and displayed in the main figure. The setup is not immediately clear and the possibility that cells can bear such an allele and divide normally is not obvious for non-specialists.

Thank you for the suggestion. The panel depicting the strategy for generating CTCF^{YY-AA} cells is now moved to the revised main figures (**Fig. 3a**).

Some CTCF-AID signals significantly decrease or disappear in the FC approach, despite having a base mean of around 7, suggesting they are far enough from noise to be considered reliable. It is unclear whether TPM thresholding is the best approach, especially since requiring TPM > 15 in the treatment may exclude potentially true downregulated genes.

In our analyses, the TPM threshold (TPM >15) is defined based on the average TPM values in control and treatment conditions. We agree TPM >15 is a relatively strict threshold, and the

exclusion of low-expressed genes likely leads to underestimation of cohesin and CTCF's regulatory scope. However, because most of the regulated genes do not pass statistical significance our approach relies on fold-change-based quantification. As Reviewer #2 points out, the use of a fold-change-based approach is contentious and some may argue that weakly expressed genes prone to false quantification. In short, yes, the exclusion of low-expressed genes likely excludes some potentially true targets, it is a compromise between obtaining reliable quantification and capturing the number of regulated genes.

S1D (or B): numbers of genes in each category would be an improvement.

This is included in the revised figure.

To show how the genes deregulated in the CTCF degnon are affected in the RAD21 degnon, a plot with the log₂fc of CTCF in x and RAD21 in y (or the contrary) and color the dots by ID/HD1/HD2 could be produced.

The correlation between gene expression changes following CTCF and RAD21 depletion is shown in **Supplementary Fig. 5g**. As expected, the correlation is low (PCC = 0.18), reflecting that each factor regulates relatively few genes, with minimal overlap between the sets and modest fold-changes in expression. Due to the small magnitude of change and limited overlap, we feel that annotating regulated gene categories made the plot very crowded and difficult to interpret. Therefore, we show a correlation without labeling of regulated gene class. However, if the Reviewer suggests, we will include the figure with labeling. For the reviewer's reference, the comparison is shown in the figure below.

S3B, in the example of Klf4, the loops in the Ctrl are difficult to visualize, could they be shown with arrows?

We revised the figure and marked the contacts with arrows.

Line 275, a definition of proximal enhancer should be provided by the authors.

This is clarified in the revised text.

In Figure 2A it is mentioned that enhancers are defined by H3K27ac and H2BK20ac, yet in the main text this is not explained. The same also happens for Supplemental Figure 6A, 6B. The use of both histone marks should be referred to in the main text.

This is now mentioned both in the text and figure legends.

Line 294: a word is missing in the sentence.

Apologies for the typo; it is fixed.

Line 309: authors should clarify what they mean by 'co-enrichment of cohesin'.

Here, co-enrichment mean overlap of CTCF and cohesin peak in ChIP-seq data. This is clarified in the revised text.

Line 311: authors should state that by distal CTCF peaks, they are referring to all the peaks that don't fall in their definition of CTCF peaks in gene promoters (+/- 200bp).

This is specified in the revised text.

Line 312: The ratio does not appear to have been calculated by the authors.

The enrichment ratio in RAD21 and CTCF-regulated genes was shown in two panels, placed side-by-side. Specific panels referring to RAD21 and CTCF are now indicated in the text.

S4B: the numbers of peaks used to build the profile should be indicated (at least in the legend).

This is specified in the revised legend.

Figure 3: panels are labeled with lowercase while it is uppercase everywhere else

Panels are labeled in uppercase, except where panel text starts with the “%” sign.

The luciferase experiment uses as a baseline control the gene promoter, but it is not clear if it is more or less active than without promoter. Because of this: it is unclear if Psm4 and Actn4 promoters are unaffected because they are never active in first place? It would also be important to know if the chosen genes are also affected by the CTCF mutant degenon.

The basal activity of the tested promoters under untreated conditions is shown in **Supplementary Fig. 11b**. Notably, the *Psmc4* and *Acnt4* promoters exhibit strong activity, indicating that their lack of CTCF dependency is not due to weak intrinsic promoter activity.

Line 488: The authors conclude that promoters with reverse motif orientation exhibit no specific bias for downregulation. The data shows in fact that there is a bias toward less downregulation. Could you rephrase it?

We rephrased the text to mention a downregulation tendency in this class.

Figure 5B: The effect of BCB and CBC on the rescue is not easy to catch from the graph, maybe the median log₂FC could be written in the legend.

This is specified in the revised figure.

Figure 6A: a horizontal dot line is missing at -1.

This is fixed.

Line 656: There is probably a typo: TSS downstream regions should be TES downstream regions.

This is fixed.

Figure 7A: the 3 panels are not described. They are probably 1h, 2h, 4h of RAD21 depletion. This should be in the legend.

Thank you for pointing this out. This information is included in the revised figure itself (revised **Fig. 1e**), as well as in the legend.

Figure 7B: the legend for RAD21 up is missing.

This information is included in the revised figure legend (revised **Fig. 1g**).

Line 1245: gencode human release 29 is GRCh38.p12 and release 19 is GRCh37.p13, can this be fixed? Also specify which version of Gencode you used (comprehensive/basic etc...).

This information is included in the revised methods section.

Line 1249: A threshold of TPM > 2 is used but the RNA-seq data used for this filter is not specified.

The used RNA-seq data are from GSE140363. This is specified in the revised methods.

Line 1256: The data provenance should be specified (publication GEO etc...).

GEO dataset references (GSE140363, GSE146328) are included in the revised methods.

Line 1271: It is not clear how the log2FC were calculated. Are they coming from 'results' DESeq2 or where they manually computed? Both for the initial 2 replicates and for the 12 supplemental replicates. Are the control and the treated matched, if so would the introduction of the replicate as covariate in DESeq2 analysis help to better estimate the log2FC? To assess the reproducibility between replicates, it would be good to have a global correlation clustering between the initial 2 replicates and the 12 new replicates.

In our original analyses, fold-changes were calculated using 'results' function of DESeq2, without the introduction of replicate as a covariate in DESeq2. Following the reviewer's comment, we repeated this analysis in RAD21 EU-seq data, with or without introducing a replicate covariance model. Introducing the covariance model reduced the number of SD genes by 1 and SU by 3, while the number of HD, ID, HU, and IU remained unchanged (see the comparison in the figure below). Given the minimal difference, we decided to keep the original analyses.

Line 1400: The title of the section is wrong.

We apologize for this error. This is corrected in the revised text.

A supplementary table listing all the data generated in this study would be useful.

A list of data generated in this study is provided in Supplementary Table 3.

On GEO, there are 'Stranded EU-seq'. Were these samples used in this study? Table of counts should be added to the GEO submission for the EU-seq.

Stranded EU-seq data were used for analyzing anti-sense transcripts in CTCF^{AID} mESC. The reason for this is that most anti-sense transcripts are short and low abundant. To avoid miscalculating the EU-seq signal derived from ambiguous TSS positions, we used stranded EU-seq data. This is clarified in the methods section. The table of counts for EU-seq is submitted to GEO.

A supplementary table listing all genes tested and their category in each analysis would be useful (CTCF-AID, RAD21-AID, A-485, CTCF mutant - AID vs CTCF-AID, CTCF mutant –AID).

This information is provided in Supplementary Tables 4a-b.

Accessibility changes at promoters might be independent of CTCF and could result from transcriptional changes.

This is plausible, but not very likely. The reason for this assumption is that accessibility is also globally reduced in distal CTCF peaks, many of which function as TAD boundaries and are not strongly transcribed. Also, CBP/p300 inhibition reduces transcription even more strongly than depletion of CTCF, but accessibility at the regulated genes is minimally reduced (Narita et al. Mol Cell 2021. PMID: 33765415; Hogg et al., Mol Cell 2021. PMID: 34019788).

Reviewer #2

Narita and colleagues address the role of cohesin and CTCF in gene regulation. Using mES degenon lines, they investigate the effect of different cohesin, CTCF, and enhancer-activity perturbations, and thereby provide insight into the looping-dependent and -independent roles of cohesin and CTCF in gene regulation.

The manuscript contains a lot of data and is not very easy to read/follow. Some of the data provide interesting new insights. However, there are also several findings that are not novel even though they are presented as if they are.

As explained in more detail below, the part of the manuscript about the role of cohesin in gene regulation is not novel. The analyses are still of interest, but should be presented as confirmatory and previous literature on this topic should be properly acknowledged.

The findings concerning the looping/enhancer-independent role of CTCF in gene regulation are more novel and certainly of interest to the field; we would therefore recommend that the authors focus their manuscript on this part and adapt their abstract and title accordingly, as the current versions contain heavily overstated claims.

We leave it up to the judgment of the Editor whether a re-written manuscript with a direct activator role of CTCF as the main finding is significant enough to justify publication in Nature Genetics.

We thank the reviewer for evaluating the manuscript and providing valuable feedback.

We are encouraged by the referee's comment that "*looping/enhancer-independent roles of CTCF in gene regulation are more novel and certainly of interest to the field.*" At the same time, we note the reviewer's observation that "*the part of the manuscript about the role of cohesin in gene regulation is not novel. The analyses are still of interest, but should be presented as confirmatory and previous literature on this topic should be properly acknowledged*"

We would like to take this opportunity to clarify how our findings on cohesin also provide important new insights into current models.

We fully acknowledge that the role of cohesin in global chromatin looping and in regulating individual enhancer targets is well established. Also, several studies show that cohesin looping promotes some enhancer-promoter interactions, but not all. In this aspect, we agree that our data confirm and support current models and make no novelty claims on it.

However, we sincerely believe that our work offers notable new insights into cohesin's role in global gene regulation—particularly in its preferential use of specific enhancer types. As the reviewer correctly points out (comment #2), recent studies have described distinct types of enhancers in mammalian cells (Nature 2015, PMID: 25517091; Mol Cell 2022, PMID: 35594855; Nat Genet. 2021, PMID: 34183853; Nature 2022, PMID: 35650434). In light of these findings, it becomes increasingly important to determine which enhancer types cohesin utilizes for gene activation.

Our results show that nearly all strongly (>2-fold change) regulated genes by cohesin depend on CBP/p300, both in mESC and NPC. This finding suggests that cohesin either preferentially engages CBP/p300-dependent enhancers or that other enhancer types are relatively rare. We regard this as a key novel contribution of our study.

Indeed, this discovery was essential in uncovering the enhancer-independent role of CTCF in regulating housekeeping genes. Without identifying the strong bias of cohesin for CBP/p300-dependent enhancers, we would not have been able to confidently distinguish whether CTCF-regulated genes were being influenced through enhancer-independent mechanisms. The clear dependency of cohesin-regulated genes on CBP/p300 allowed us to delineate the separate functions of CTCF in both enhancer-dependent and -independent gene regulation.

After the submission of this manuscript, it has been proposed that cohesin may promote looping by "loop capture" rather than "loop extrusion" (Uhlmann F, Mol Cell 2025. PMID: 40118039). The strong bias of cohesin-regulated genes for CBP/p300 dependency suggests that if cohesin functioned by "loop capture," it would imply that the "loop capture" mechanisms preferentially capture loops between CBP/p300-dependent enhancers and their

promoters. We expanded the discussion to consider the implications of our findings for assessing different models of how cohesin may affect transcription. In short, our data seem most consistent with the prevailing loop extrusion model, but we do not exclude other possibilities.

In summary, we fully agree with the reviewer that demonstrating CTCF's non-architectural function in gene regulation is both novel and significant. At the same time, we believe that our findings regarding cohesin's enhancer-type specificity and its broader role in gene regulation provide a meaningful advancement to the current understanding in the field.

Major comments

1. The justification for the first part of the manuscript "Several studies have examined transcriptional changes following acute RAD21 and CTCF depletion^{20,25-29}, yet they have not addressed why these perturbations affect only a limited set of genes or whether the affected genes are enhancer-regulated." is not correct. There are several papers that have addressed this (although they have not looked at this in a genome-wide manner). First papers that come to mind (although we are sure that there are more) are: Thiecke et al Cell Reports 2020, Aljahani et al Nature Communications 2022, Kane et al NSMB 2022, Rinzema et al NSMB 2022, Goel et al Nature Genetics 2023. Furthermore, in several recent review papers on the topic of loop extrusion and enhancers and/or gene regulation, notions that are presented in the paper as novel findings are already discussed.

We recognize that the original sentence was unclear and confusing. In response to the reviewer's comment, we also revised the quoted sentence as follows: *"Previous studies examined chromatin looping and global transcriptional changes after acute RAD21 and CTCF depletion^{20,25,26,44-46}, but they did not investigate the nature of enhancers involved in the regulation of the affected genes. Addressing this question is important as metazoans possess diverse types of enhancers functioning with distinct coactivators²⁷⁻³¹."*

To check whether questions addressed in our work were addressed in the mentioned references, we carefully reviewed all five references cited by the referee. These studies consistently focus on the role of cohesin in chromatin looping, specifically addressing whether cohesin is universally required for E–P interactions, or whether such interactions can occur independently of cohesin. Their shared conclusion is that while some enhancers depend on cohesin to engage target promoters, many can do so without it.

We feel that the questions addressed in the mentioned references and our work are different. We do not examine chromatin loops directly, nor do we claim novelty regarding cohesin's role in looping or the existence of cohesin-independent E–P mechanisms.

To better clarify the focus of our work, we mention the following five questions in the manuscript introduction:

1. Metazoans have diverse enhancer types—does cohesin have any preference for a specific enhancer type?
Conclusion: Cohesin preferentially supports gene activation by CBP/p300-dependent enhancers.

2. Why is there minimal overlap between genes downregulated after cohesin vs. CTCF depletion?
Conclusion: Beyond its known architectural function, CTCF regulates transcription independently of cohesin looping.
3. Does CTCF regulate transcription through non-architectural mechanisms? If so, what is the nature of genes are regulated by its architectural and non-architectural roles?
Conclusion: CTCF's architectural role regulates enhancer-dependent, cell-type-specific genes, while its non-architectural role regulates enhancer-independent housekeeping genes.
4. Why is CTCF essential for mammalian but not *Drosophila* cell proliferation, despite its evolutionary conservation?
Conclusion: In mammals, CTCF functions as a transcriptional activator of essential housekeeping genes, explaining its essentiality.
5. What functional similarities exist between CTCF and its vertebrate-specific paralog CTCFL, which cannot anchor cohesin loops but binds many of the same promoter sites as CTCF?
Conclusion: Both CTCF and CTCFL can bind to the same promoters and function as transcriptional activators, explaining CTCFL's bias for binding in promoters.

To the best of our knowledge, these specific questions have not been addressed in prior studies. Although previous work has examined global gene regulation after cohesin/CTCF depletion, it has remained challenging to distinguish between enhancer-dependent and enhancer-independent mechanisms. This distinction was essential for uncovering CTCF's non-architectural role in gene activation.

To clarify, we do not claim that enhancers require cohesin universally, nor that we are the first to show that enhancers can act independently of cohesin. Rather, our work complements previous findings by providing a genome-wide estimate of cohesin-dependent and -independent roles of cohesin and CTCF in gene regulation in studied cell models. It further delineates CTCF's dual roles in transcriptional regulation.

All five references suggested by the referee have also been included and appropriately cited in the revised manuscript.

2. The statement in the abstract and similar statements throughout the manuscript "cohesin appears to exclusively support gene activation via CBP/p300-dependent enhancers" should be removed. The authors have only investigated CBP/p300-dependent enhancers and can therefore not make any claims that other types are not affected. It would be better not to overinterpret the data and more precisely discuss the data without overstatements.

For a detailed response, please see our reply to point #2 from referee #1.

Briefly, we do not intend to overstate cohesin's role at CBP/p300-dependent enhancers or downplay its involvement with other enhancer types—we aim only to reflect the data accurately. Because the genes most strongly downregulated by RAD21 depletion are also sensitive to CBP/p300 inhibition, we initially concluded that cohesin functions almost exclusively with CBP/p300.

While we believe this interpretation is supported by the data, we respect the referee's perspective and have revised the language accordingly. We removed the word "exclusively" and now state that cohesin preferentially activates genes through CBP/p300-dependent enhancers, leaving the possibility open that cohesin may also function with other enhancer types.

If the referee is aware of specific references demonstrating cohesin's role in other enhancer types, we would be happy to cite and highlight them.

3. We suggest that the authors re-structure their manuscript and separate the part about cohesin and CTCF. Now they start with cohesin, then focus on CTCF, and switch back to cohesin at the end, which makes the manuscript unnecessarily difficult to follow.

We are grateful to the reviewer for this excellent suggestion. We have now restructured the manuscript – we start with cohesin and then the rest of the manuscript focuses specifically on CTCF.

4. The statement that the effects of cohesin and CTCF perturbations on transcription are bigger than previously appreciated is somewhat unfair, as the authors simply decide to take non-significant changes into consideration, which is highly controversial. With this approach, it would be good to only include datapoints for which the fold-change is at least in the same direction for the independent replicates, but it is unclear whether this is the case. If that would be changed/clarified, we have no specific issues with this strategy (as the validation is relatively convincing), but it was known before that there are many changes that are not statistically significant. The authors should interpret these data with more caution and less overstatements, especially because they are not backed up by any statistical analysis. In this regard, it is of interest that previous work has showed increased variability of expression upon cohesin perturbation (Hafner et al Molecular Cell 2023). This could also explain the lack of statistically significant changes and should be acknowledged.

We thank the reviewer for their thoughtful feedback. This comment raises two concerns:

1. That the comparison of transcriptional changes with previous studies is unfair.
2. That the classification of regulated genes based on fold-change, rather than statistical significance, is controversial.

We address both issues separately below.

On the comparison of gene expression changes with previous studies

We agree with the reviewer that comparing the number of statistically significant regulated genes with those showing non-significant fold changes would be inappropriate. To ensure a fair comparison, we compared only statistically significant genes from our dataset to those reported in a previous study conducted in the same cell type (Hsieh et al., Nat Genet, 2022; PMID: 36471071). This comparison showed that the number of statistically significantly regulated genes in our study is highly consistent with previously published results. For clarity, we now explicitly state the number of genes regulated by RAD21^{AID} and CTCF^{AID} depletion in both our study and the work by Hsieh et al.

We want to emphasize that our intention is not to downplay or undermine earlier work. Rather, we aim to highlight the inherent difficulty in identifying and validating regulated genes in such analyses and demonstrating their biological relevance.

In response to the reviewer's comment, we have revised the relevant text in the discussion section as follows:

*“Aligning with previous findings^{20,25,26,44-46}, only a limited number of genes show statistically significant regulation after acute RAD21^{AID} and CTCF^{AID} depletion. Fold-change-based analyses suggest that cohesin and CTCF regulate hundreds of genes, albeit with modest effects. We propose three possible explanations for this subtle regulation. First, although degron systems enable rapid protein depletion, they do not eliminate the target protein entirely; residual RAD21 and CTCF are likely to retain partial functionality. Second, many are only partially enhancer-regulated⁴², so disruption of E-P looping is expected to cause only mild transcriptional reductions. Third, cohesin is not equally essential for all enhancers: while distal enhancers may critically depend on cohesin, proximal enhancers can function without cohesin^{69,70}. Although fold-change-based approaches may carry a higher false-positive rate than significance-based methods, the robustness of our global analyses is supported by multiple orthogonal assays (see **Supplemental Note 1**).”*

This revision acknowledges the limitations of our fold-change-based approach while citing the relevant literature. We hope the reviewer finds this revised text satisfactory.

Regarding the reference to Hafner et al., we note that their study examined transcriptional variability at the single-cell level using super-resolution microscopy. Unfortunately, our bulk RNA-seq analysis does not allow us to make conclusions about cell-to-cell variability.

Finally, we would like to point out that weak transcriptional responses to cohesin depletion are not unexpected. Even well-characterized targets of enhancers, such as *Nanog*, *Sox2*, and *Slc2a3*, show modest downregulation following cohesin loss (Aljahani et al., Nat Commun, PMID: 35440598).

On the use of fold-change for classifying regulated genes

We agree that statistical significance is important for classifying gene regulation. However, in cases such as RAD21 or CTCF depletion, where changes in expression are modest, relying

solely on statistical thresholds can result in missing biologically relevant regulation. This prompted our use of a fold-change-based approach.

We recognize that when using fold changes, it is important to show that modest, non-significant gene expression changes are not random. To address this, we pursued two strategies: (1) increasing the number of replicates, and (2) using orthogonal assays and multi-way comparisons.

For example, in RAD21^{AID} depleted mESC, we performed 12 additional replicates. Although additional replicates helped recover more statistically significant genes, it was not feasible to scale this for every perturbation due to cost. Nonetheless, the results showed that genes with a mean downregulation >1.3-fold in the initial two replicates, virtually all showed downregulation trend (99% downregulated >1.1-fold, 93% by >1.2-fold, and 90% by >1.3-fold) in the majority of the additional 12 replicates (Supplementary Fig. 2c). These analyses showed that although genes showing modest fold-change do not pass statistical cut-off, the trend of regulation is highly consistent across replicates. Based on this observation, we classified genes using a fold-change of >1.3 and having an EU-seq expression value of >15. For RAD21, we used >1.3-fold in the initial two replicates and consistent >1.3-fold regulation in at least half (6/12) of new replicates.

To further validate the non-random nature of these genes, we performed multiple cross-comparisons using independent data types. We believe that the consistent regulation observed across orthogonal datasets provides the strongest support for the robustness of our conclusions (see Supplemental Note 1).

In summary, we fully acknowledge the limitations of a fold-change-based approach, and we have clearly outlined its rationale and constraints in the manuscript to ensure transparency. Nonetheless, the strong cross-validation across independent datasets reinforces our confidence in the findings.

We hope this explanation adequately addresses the reviewer's concern.

5. It is unfortunate that the study relies on only two biological replicates for the most interesting part about CTCF. More replicates would be valuable and contribute to the robustness of the findings.

Because of the expensive nature of these experiments, two biological replicates are standard in the field for genome-scale nascent transcriptome and ChIP-seq analyses (Wang H, et al. Nature. 2023. PMID: 36859550; Hsieh TS, et al. Nat Genet. 2022. PMID: 36471071; Gressel et al. Nat Commun. 2019. PMID: 31399571). Among all perturbations used in our work, RAD21^{AID} depletion in mESC caused the weakest transcription changes. For this reason, we performed 14 biological replicates for RAD21^{AID}-dependent transcription changes in mESC.

We would also like to point out that although RAD21^{AID}-regulated transcriptomes in NPCs are measured in two replicates, each replicate includes three different time points. Also, nascent transcriptomes in CTCF^{YF-AA} cells include two replicates, but each replicate includes two different time points.

Genes downregulated by CTCF^{AID} depletion show stronger regulation than RAD21^{AID} depletion. The strength of our conclusions on CTCF is most strongly supported by orthogonal

evidence from RNAPII ChIP-seq, and ATAC-seq. Furthermore, CTCF^{YF-AA} mutant cells independently confirm gene regulation by promoter-bound CTCF. With multiple lines of independent evidence, we believe that our conclusions on CTCF-regulated genes are robust.

6. Figure 4F: The orientation of CTCF should be considered in relation to the direction of transcription of the gene, but this does not seem to be the case.

The orientation of CTCF is indeed considered with respect to the gene. This is clarified in the revised text.

7. Page 34: "Depending on the position of CTCF binding relative to the TSS, CTCF can act as a repressor of either anti-sense or sense transcript." This statement is only supported by a few examples in Figure 6 and Supplemental Figure 7. It would be more convincing to show a quantitative analysis of this effect across all relevant genes.

The position and orientation differences in different classes of CTCF-regulated genes is shown in Fig 4e and g.

8. Is it possible that the effects of CTCF perturbation on RNAPII recruitment are secondary to the observed changes in chromatin accessibility? If so, this should be acknowledged.

Yes, we believe that reduction in RNAPII at strongly regulated CTCF genes is secondary to changes in chromatin accessibility. To acknowledge this, we included this sentence. *"This reduction in Pol II binding is likely a consequence of reduced chromatin accessibility."*

9. Page 42: "By analyzing the architectural and non-architectural roles of cohesin and CTCF (Figure 7D), this work introduces four major conceptual advances to these ongoing discussions: (1) Following acute cohesin and CTCF depletion, the extent of gene dysregulation is much broader than previously appreciated. (2) Among diverse enhancer types posited, there is likely only one enhancer type – the CBP/p300-dependent type – that use cohesin-dependent looping for gene activation. (3) Beyond its architectural roles, CTCF functions as a position and orientation-specific transcription repressor and activator, controlling the expression of housekeeping genes, including ones that are pan-essential for mammalian cell proliferation. (4) Despite their differences in anchoring cohesin loops, CTCF and CTCFL share transcription activation ability."

As discussed above, the first two conclusions are not novel.

To avoid repetition, regarding the novelty of the first two original claims, please see our detailed responses in our response to general comments from this reviewer, as well as point #1 from this reviewer, and point # 2 from referee #1.

Based on the reviewer's comment, we revised the text as follows:

“We show that acute RAD21^{AID} depletion downregulates hundreds of genes. Beyond anchoring cohesin loops, CTCF acts as a position- and orientation-specific transcriptional regulator, controlling the expression of housekeeping genes essential for mammalian cell proliferation. These findings offer important insights into their functions in global transcriptional regulation.”

The revised text simply states the results included in the manuscript, without undermining previous work.

Minor comments

1. There are a few sentences that do not make sense:

Page 2: "Using plasmid-based reporter assay, early reports suggested that CTCF can function as transcription activator activator21 and activator 22, but contemporary models raise doubts about in-vivo relevance of these observations and interpret CTCF's role in gene regulation almost exclusively in the context of its architectural function in genome folding 23,24."

Page 29: "CTCF has a vertebrate-specific, called BORIS (CTCFL), is usually only expressed in the testis, but aberrant expression is found in cancers 56. CTCF and BORIS share high sequence identity in their CTCF in DNA binding zinc fingers (ZFs) but exhibit minimal conservation in non-DNA binding N- and C-termini 57."

We apologize for these errors. This is fixed in the revised version.

2. Validation of the depletion efficiency is not consistently included. Even though some of the used cell lines have been characterized previously, it is still important to validate the depletion efficiency, as the authors use shorter depletion times than previously described. As incomplete depletion may also explain the limited effects in some cases, it would be important to include this – ideally using ChIP-seq data, so that the degree of depletion can be directly linked to effect sizes at individual genes.

Following the reviewer's recommendation, the depletion efficiencies of RAD21^{AID} and CTCF^{AID} are shown in revised Supplementary Figs. 1a and 5a. In our experience, microscopy-based imaging provides a more robust approach for assessing protein depletion, including cell-to-cell variability, Therefore, we used this approach for checking depletion efficiencies.

3. It seems that the authors did not make use of spike-ins for normalization. This may however be useful after genome-wide perturbations and may help the authors in finding more robust changes. It would be good if the authors can consider including this or at least comment on why they may think it is not useful or appropriate in their study.

The reason for using DESeq2-based normalization is that used perturbations regulate only a small fraction of expressed genes. In this case, DESeq2-based normalization is robust and is a standard approach in the field for quantifying differentially regulated genes. The use of spike-in becomes important in conditions where gene expression is altered globally.

4. Fig. 1B: It would be nice to add the numbers of genes included in each category.

This information is included in the revised figure.

5. Page 11: Some of the examples described here (Prdm14/ Slco5a1 locus in context of cohesin and CTCF depletion) have been discussed previously in Aljahani et al Nature Communications 2022. This should be acknowledged.

In addition to citing the original references reporting the role of cohesin/CTCF in Prdm14/ Slco5a1 locus, we now included the suggested reference from the Oudelaar group that independently confirmed it.

6. The notion that enhancer-promoter interactions are not only dependent on loop extrusion is well established in the field. The authors mention LDB1 and YY1 in this context. Recently, an important role for Mediator and RNAPII has become clear as well and this would be good to discuss.

We agree – the idea that E-P interactions can occur without cohesin is not new, and we acknowledge this. We included recent references from the Oudelaar and Danko labs showing the role of Mediator and RNAPII in promoting E-P interactions.

Point-by-point response to reviewers' comments

We thank both referees for evaluating the revised manuscript. We are pleased that they found our responses satisfactory to most of their comments. Below, we provide our responses to their remaining concerns.

Reviewer #1 (Remarks to the Author):

The authors have satisfactorily addressed most of our points. However, some clarifications on the types of enhancers existing is needed.

In the rebuttal, the authors write that there is not much evidence supporting different enhancer types in vivo: “However, it should be noted that the idea of multiple enhancer types mainly rests on coactivator dependencies of enhancers in plasmid-based reporter assays (Nature 2015, PMID: 25517091; Mol Cell 2022, PMID: 35594855; Nat Genet. 2021, PMID: 34183853; Nature 2022, PMID: 35650434).”, “Currently, the evidence supporting multiple enhancer types in vivo is limited.”, and “We are unaware of examples of distal enhancers in native chromatin that activate gene expression independently of CBP/p300” Yet, in the manuscript introduction, they write, “Metazoans have diverse enhancer types(27-31) » and “Among the functional enhancers identified through CRISPR interference, those linked to proximal genes mostly function using CBP/p300 but those skipping active genes mostly function without CBP/p300” To us these points in the manuscript are in contradiction with the rebuttal. Moreover, this last point “Among the functional enhancers...” is also controversial as it is based on a non-peer-reviewed paper (the one that was previously co-submitted but rejected). Moreover, it suggests that there is another class of enhancers (that can skip active genes) while the results suggest that if there are, they do not use cohesin. This confuses the message.

In this perspective, this reviewer wonders why there is so much emphasis on enhancer types/classes in the introduction? Could this be removed and acknowledge that most enhancers are dependant on CBP/P300 without specifying what remains as it confuses the message.

Following the reviewer's comment, we revised the text as follows:

“Multiple enhancer types are reported in metazoans⁴⁰⁻⁴⁴. Prior studies assessed transcription changes after acute RAD21 or CTCF depletion^{20,25,26,44-4}, but did not resolve what enhancer type(s) drives the observed changes. A close association between

CBP/p300 and enhancers³⁵⁻³⁹, motivated us to examine cohesin and CTCF's involvement in activating genes through CBP/p300-dependent enhancer type."

This acknowledges reports of multiple enhancer types, while explaining the rationale for our focus on one specific enhancer type.

Reviewer #2 (Remarks to the Author):

We would like to thank the authors for addressing most of our concerns. We think that the revised manuscript has improved a lot. However, as raised in our initial review, we are not very comfortable with the current title. It is a very "strong" title and in our view it does not reflect the content of the paper well, as it overstates the contribution of this manuscript to our understanding of cohesin/CTCF in gene regulation (as we already pointed out, the work on cohesin is mostly confirmatory) and understates the fact that there are still many open questions about the exact molecular functions of these proteins.

We revised the title as: "Delineating the architectural and non-architectural functions of CTCF and cohesin in global gene regulation"

We believe this title accurately reflects the findings presented in the manuscript.

We respectfully disagree with the characterization that our findings on cohesin are mostly confirmatory. In fact, the scope of cohesin's role in global gene regulation has been underestimated. For example, only three genes were reported as downregulated after acute cohesin depletion (Nat Genet. 2022, PMID: 36471071, Fig. 3h)

This also clearly implied in the abstract of PMID: 36471071 "It remains unclear why acute depletion of CTCF (CCCTC-binding factor) and cohesin only marginally affects expression of most genes despite substantially perturbing three-dimensional (3D) genome folding at the level of domains and structural loops.Thus, although CTCF, cohesin, WAPL or YY1 is not required for the short-term maintenance of most E-P interactions and gene expression...."

To our knowledge, no prior study has demonstrated that cohesin preferentially activates CBP/p300-dependent genes, nor shown that it regulates nearly one-third (i.e, many hundreds) of CBP/p300-regulated genes in differentiated cells. Thus, cohesin's scope in gene regulation is more broader than previously appreciated, and we discuss the reasons why its scope has been underestimated. In short, our work examine cohesin's scope in gene regulation from a different perspective, providing valuable contribution to existing knowledge. We therefore feel that is would be inaccurate to describe our work on cohesin as mostly confirmatory.